# The Panoptes system uses decoy cyclic nucleotides to defend against phage

Ashley E. Sullivan[1], Ali Nabhani[2], Daniel S. Izrailevsky[2], Kate Schinkel[1], Charlotte R. K. Hoffman[1], Laurel K. Robbins[1], Toni A. Nagy[1], Melissa L. Duncan[1], Hannah E. Ledvina[1], Annette H. Erbse[1], Emily M. Kibby[1], Uday Tak[1], David M. Dinh[2], Eirene Marie Q. Ednacot[2,3], Christy M. Nguyen[4,5], A. Maxwell Burroughs[6], L. Aravind[6], Aaron T. Whiteley[1✉] & Benjamin R. Morehouse[2,3,7,8✉]

Bacteria combat phage infection using antiphage systems and many systems generate nucleotide-derived second messengers upon infection that activate effector proteins to mediate immunity[1]. Phages respond with counter-defences that deplete these second messengers, leading to an escalating arms race with the host. Here we outline an antiphage system we call Panoptes that indirectly detects phage infection when phage proteins antagonize the nucleotide-derived second-messenger pool. Panoptes is a two-gene operon, *optSE*, wherein OptS is predicted to synthesize a nucleotide-derived second messenger and OptE is predicted to bind that signal and drive effector-mediated defence. Crystal structures show that OptS is a minimal CRISPR polymerase (mCpol) domain, a version of the polymerase domain found in type III CRISPR systems (Cas10). OptS orthologues from two distinct Panoptes systems generated cyclic dinucleotide products, including 2′,3′-cyclic diadenosine monophosphate (2′,3′-c-di-AMP), which we showed were able to bind the soluble domain of the OptE transmembrane effector. Panoptes potently restricted phage replication, but phages that had loss-of-function mutations in anti-cyclic oligonucleotide-based antiphage signalling system (CBASS) protein 2 (Acb2) escaped defence. These findings were unexpected because Acb2 is a nucleotide 'sponge' that antagonizes second-messenger signalling. Our data support the idea that cyclic nucleotide sequestration by Acb2 releases OptE toxicity, thereby initiating inner membrane disruption, leading to phage defence. These data demonstrate a sophisticated immune strategy that bacteria use to guard their second-messenger pool and turn immune evasion against the virus.

Bacteria are under constant assault from their viruses, bacteriophages (phages), and have evolved sophisticated mechanisms to protect themselves[2,3]. The immune system of any bacterium is a combination of individual pathways called antiphage systems that limit phage replication. Bacteria encode an average of about six antiphage systems per strain[4]. However, each bacterial strain in a species dynamically exchanges antiphage systems through horizontal gene transfer, giving each a different suite of defences[2].

Antiphage systems must rapidly sense phage infection and transduce an activation signal to an effector protein to stop virion production. Recent progress cataloguing antiphage systems has identified that many systems transduce their activation signal through nucleotide-derived second messengers[1]. The advantage of nucleotide-derived second messengers is that they can be synthesized from abundant precursors and amplify a rare phage-sensing event into stoichiometrically more activated effector proteins. CBASS synthesizes cyclic di- and trinucleotides[5–7], the pyrimidine cyclase system for antiphage resistance (Pycsar) synthesizes cyclic mononucleotides[8] and the Thoeris system synthesizes cyclic adenosine diphosphate (ADP)-ribose derivatives in response to phage[9]. The success of these strategies is further underscored by their remarkable spread through lateral transfer: homologues of CBASS components can be found in the cyclic guanosine monophosphate (GMP)-adenosine diphosphate (AMP) synthase–stimulator of interferon genes (cGAS–STING) pathway of metazoans, and homologues of Thoeris components can be found in the immune system of plants and metazoans[10,11].

Nucleotide-derived second messengers are also a vulnerability because viruses use phosphodiesterases and 'sponge proteins' to interrupt signalling[12–19]. Just as the nucleotide second-messenger strategy is pervasive from bacteria to eukaryotes, so too are some of the immune evasion strategies from viruses that infect bacteria and metazoans[20,21]. Bacterial antiphage systems synthesize a wide variety

[1]Department of Biochemistry, University of Colorado Boulder, Boulder, CO, USA. [2]Department of Molecular Biology and Biochemistry, University of California, Irvine, Irvine, CA, USA. [3]Department of Pharmaceutical Sciences, University of California, Irvine, Irvine, CA, USA. [4]Department of Biological Chemistry, University of California, Irvine, Irvine, CA, USA. [5]Center for Epigenetics and Metabolism, University of California, Irvine, Irvine, CA, USA. [6]Computational Biology Branch, Division of Intramural Research, National Library of Medicine, National Institutes of Health, Bethesda, MD, USA. [7]Institute for Immunology, University of California, Irvine, Irvine, CA, USA. [8]Center for Virus Research, University of California, Irvine, Irvine, CA, USA. ✉e-mail: aaron.whiteley@colorado.edu; b.morehouse@uci.edu

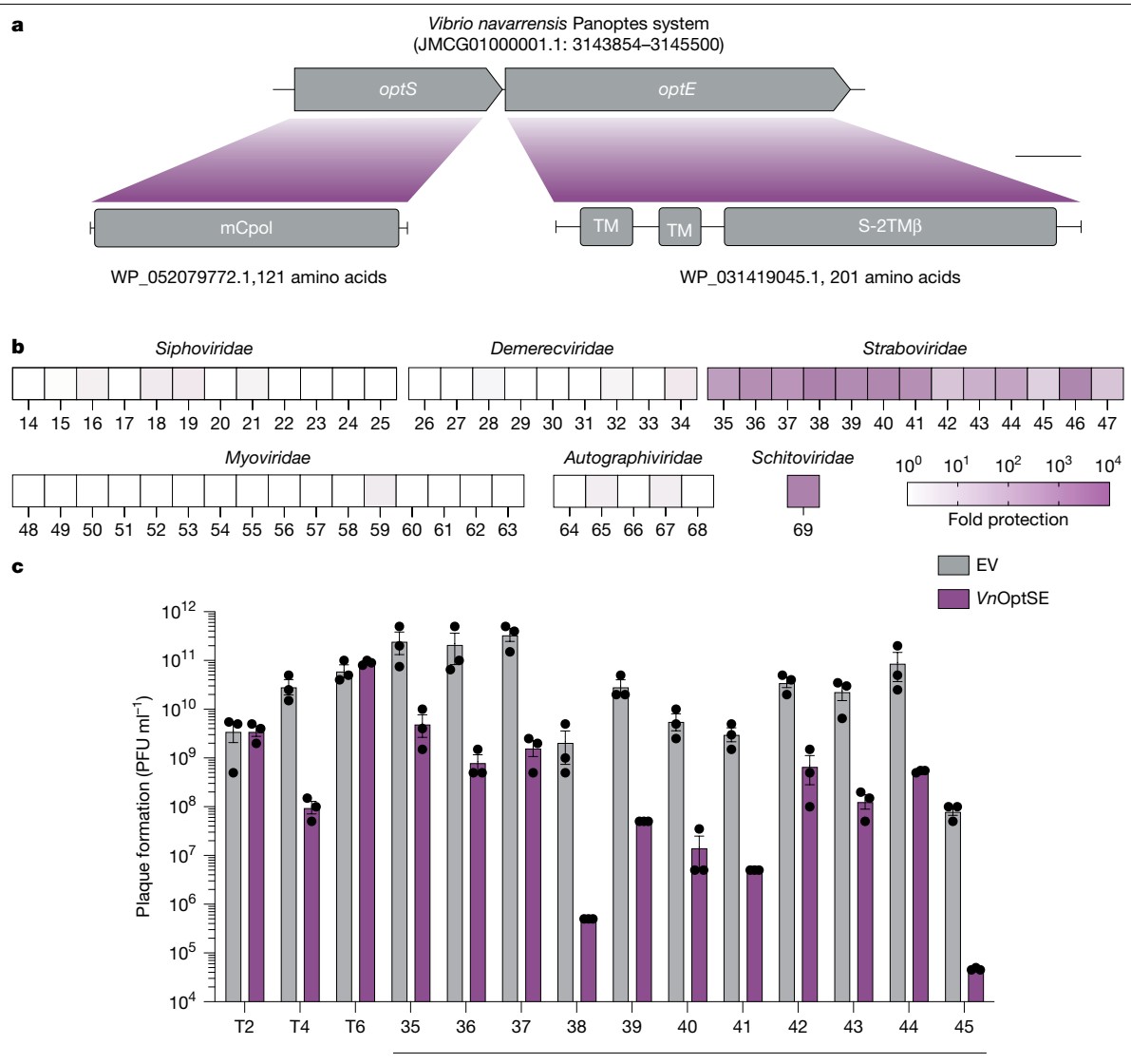

**Fig. 1 | The Panoptes system restricts phage. a**, Accession number and genome coordinates for *V. navarrensis* Panoptes operon genes, *optS* and *optE*. Domain architectures and accession numbers for encoded proteins are shown. **b**, Heat map of fold defence provided by *Vn*OptSE for a panel of diverse phages from the BASEL collection. *Escherichia coli* expressing *Vn*OptSE were challenged with phages and fold defence was calculated for each phage by dividing the efficiency of plating (in PFU per millilitre) on empty vector by the efficiency of

plating on *Vn*OptSE-expressing bacteria. Family names are above each indicated set of phages; numbers indicate BASEL phage numbers. **c**, Efficiency of plating of indicated phages infecting *E. coli* expressing a plasmid with either *Vn*OptSE or an empty vector (EV). Data represent the mean ± s.e.m. of *n* = 3 biological replicates, shown as individual points. TM, transmembrane. Scale bar, 25 amino acids.

of chemical variations of their class of second messenger, often using different nucleotide bases, forming different phosphodiester linkages or even incorporating amino acids[7,8,22]. This variability is likely to be the result of the arms race between host and pathogen. Nevertheless, most phage proteins that antagonize second messengers can interact with a wide range of second-messenger variants, suggesting that changing the second messenger is not sufficient to outpace the virus[12–15,18,19,23–25].

Here we investigated a two-gene operon (*optSE*) that initially seemed to be similar to the CBASS, Pycsar and Thoeris systems, with a predicted nucleotide second-messenger synthase (OptS) and a nucleotide-binding effector protein (OptE). Unlike systems that synthesize signalling nucleotides in response to phage, we found that OptS constitutively produced signalling nucleotides to repress OptE-mediated growth inhibition. In this way, OptSE guards the cyclic dinucleotide pool of the cell by detecting phage anti-defence proteins

that deplete cyclic nucleotides to circumvent CBASS signalling. We named OptSE the Panoptes antiphage system for Argus Panoptes, the all-seeing, many-eyed giant in Greek mythology who was a faithful watchman to Hera.

## Panoptes is an antiphage system

We investigated a candidate two-gene Panoptes operon from *Vibrio navarrensis*. The first gene (*optS*, S = synthase) encoded an mCpol domain that is predicted to synthesize a nucleotide second messenger[26]. The second gene (*optE*, E = effector) encoded two predicted transmembrane (TM) domains and a SMODS (second messenger oligonucleotide or dinucleotide synthetase)-associating 2TM, β-strand rich (S-2TMβ) domain[26] (Fig. 1a). OptE is homologous to Cap15 proteins in CBASS that disrupt membrane integrity in response to cognate cyclic nucleotide binding[26,27].

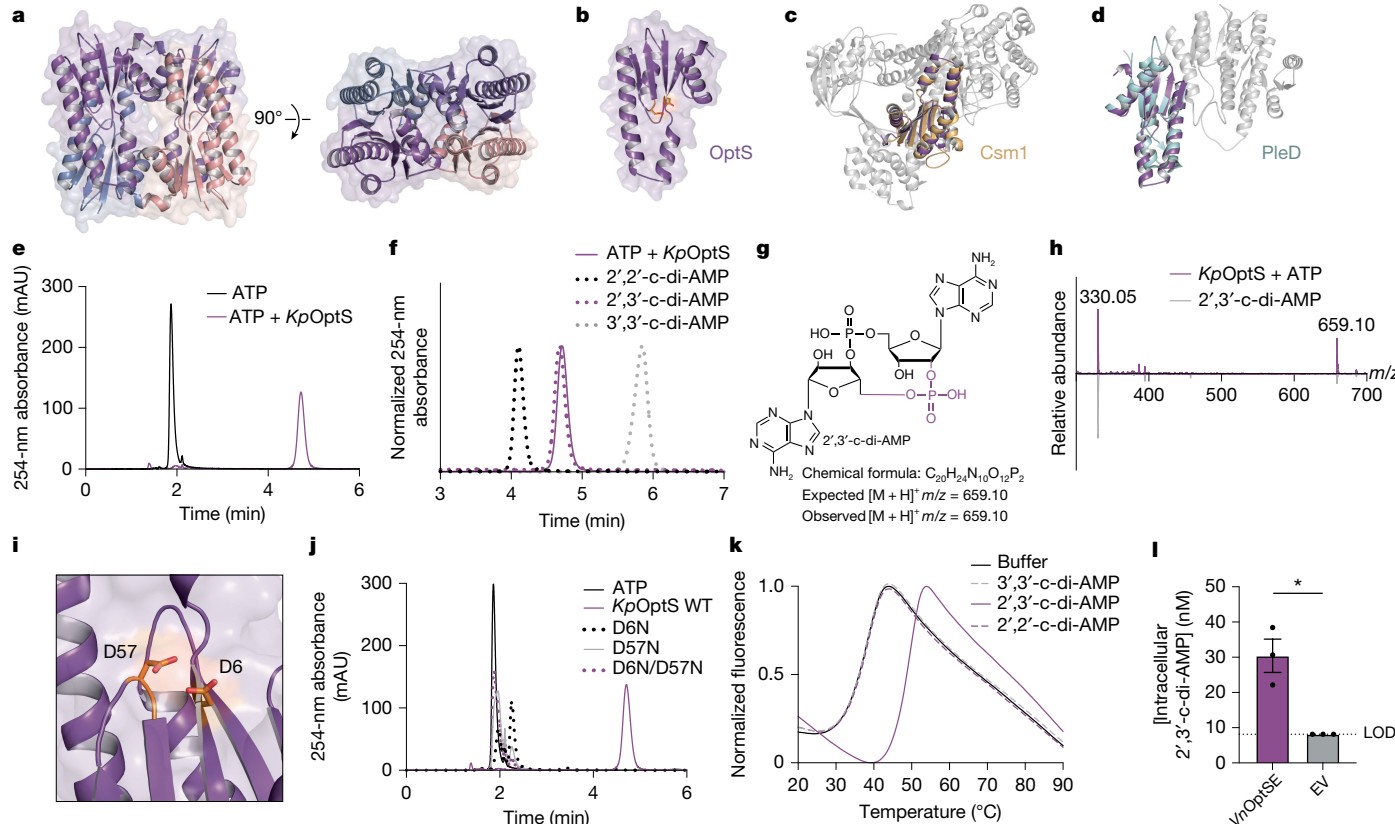

**Fig. 2 | OptS is a minimal CRISPR polymerase. a**, The crystal structure of the *Kp*OptS apoprotein shows a putative tetrameric architecture. Two rotated views are depicted with each protomer shown in a different colour. **b**, Structure of the isolated protomer (monomer) highlighting overall domain fold and putative active site (orange sticks for Asp6 and Asp57). **c**, Superposition of type III Cas10 CRISPR polymerase palm domain (Protein Data Bank (PDB) 6O75, light orange) with *Kp*OptS (purple). The remainder of the Cas10 structure depicted in grey highlights the minimal nature of the mCpol. **d**, Superposition of GGDEF-containing enzyme PleD (PDB 1W25, light blue) with *Kp*OptS, with unaligned regions shown in grey. **e**, HPLC analysis of an ATP chemical standard compared with the product of *Kp*OptS when incubated with ATP alone. **f**, HPLC analysis of 2′,2′-c-di-AMP, 2′,3′-c-di-AMP and 3′,3′-c-di-AMP chemical standards, compared with the product of *Kp*OptS. **g**, Chemical structure of 2′,3′-c-di-AMP.

**h**, Bottom, mass spectrometry spectra of a 2′,3′-c-di-AMP standard. Top, the *Kp*OptS ATP-dependent product. A value of 330.05 $m/z$ indicates a substantial proportion of the doubly charged species. **i**, Magnified view of the *Kp*OptS putative active site highlighting catalytic Asp6 and Asp57 (orange sticks). **j**, HPLC analysis of the reaction products when wild-type (WT) or mutant *Kp*OptS is incubated with ATP, compared with an ATP chemical standard. **k**, Thermal shift data showing the melt profile for a soluble version of *Vn*OptE incubated with several c-di-AMP analogues. Buffer indicates protein alone with no extra cyclic dinucleotide added. For **e**,**f**,**j** and **k**, data are representative of $n = 3$ biological replicates. **l**, Quantification of intracellular 2′,3′-c-di-AMP from *Vn*OptSE or empty vector-expressing *E. coli*. Data represent the mean ± s.e.m. of $n = 3$ biological replicates, shown as individual points. A two-sided Student's *t*-test was used. *$P < 0.05$. LOD, limit of detection; mAU, milli-absorbance unit.

We expressed the operon from its endogenous promoter in *Escherichia coli* MG1655 and challenged these bacteria with a panel of diverse phages. The *Vn*OptSE operon specifically defended against phages from the *Straboviridae* family (Fig. 1b). We focused on the *Tequatrovirus* genus that includes the T-even phages and found *Vn*OptSE provided more than 100-fold protection against phage T4, Bas36, 37, 42, 43 and 44, and more than 1,000-fold protection for Bas38, 39, 40, 41 and 45 (Fig. 1b,c). We selected the best characterized T-even phage, phage T4, for further analysis. We confirmed that *Vn*OptSE provided defence against phage T4 in both soft agar and liquid cultures (Fig. 1c and Extended Data Fig. 1a,b). A surprising outlier was phage T2, which was not restricted by *Vn*OptSE. These findings were unexpected as phage T2 is generally much more sensitive to phage defence systems than phage T4 (refs. 28,29).

## OptS is a minimal CRISPR polymerase

To better understand the possible enzymatic function of the OptS protein, we expressed and purified an orthologue from *Klebsiella pneumoniae* KP67 (*Kp*OptS) for in vitro biochemical and structural characterization (Extended Data Fig. 1c). *Kp*OptS and *Vn*OptS share

about 45% sequence identity and are predicted structural homologues (Extended Data Fig. 2a). We confirmed that the *Kp*OptSE Panoptes system defends against phages and found similar phage specificity as observed for *Vn*OptSE (Extended Data Fig. 1d–f). We determined the crystal structure of *Kp*OptS, which was refined at a nominal resolution of 1.75 Å (Extended Data Table 1). The crystallographic asymmetric unit contained four copies of *Kp*OptS, forming an apparent tetrameric complex or dimer of dimers (Fig. 2a and Extended Data Fig. 2b–e). Minimal conformational and B-factor variation was observed across the protomers and superposition showed root mean square deviation (r.m.s.d.) values of roughly 0.3–0.4 Å, on average, between pairs. Two symmetrically related dimer interfaces are observable in the crystal (OptS1–OptS2 and OptS3–OptS4; Extended Data Fig. 2c), each with about 1,100 Å² of buried surface area, as determined by Proteins, Interfaces, Structures and Assemblies (PISA) analysis (Supplementary Table 1). A second interface connecting the two dimers has around 500 Å² of buried surface area. It is not clear if this second interface is sufficient to determine whether the tetrameric complex represents a true species or whether it is a feature of crystallization. Although *Kp*OptS is a small protein (about 14 kDa) on the basis of primary amino acid sequence alone, size-exclusion chromatography supported the

existence of a larger molecular weight species in solution (greater than a dimer) (Extended Data Fig. 2f). Mass photometry experiments showed that *Kp*OptS can exist as a stable tetramer in solution at low protein concentration (around 58 kDa monodisperse population observed at a concentration of roughly 50 nM; Extended Data Fig. 2g,h). These data show that the tetrameric complex in the crystal structure probably is observable and biologically relevant, but we cannot conclude what the ratio of dimer:tetramer is in solution.

The protomer of *Kp*OptS contains a polymerase palm-domain fold composed of four primary α-helices surrounding a five-stranded β-sheet and a loop region containing a putative catalytic site centred on a core RNA-recognition motif fold[26] (Fig. 2b). Although different in quaternary structure and domain architecture, the protomeric unit of *Kp*OptS aligns well with the palm domains of Cas10 (Csm1) from *Thermococcus onnurineus* and the GGDEF diguanylate cyclase PleD from *Caulobacter crescentus* (r.m.s.d. values of 3.86 Å and 2.39 Å, respectively)[30,31] (Fig. 2c,d). These polymerase domains were also earlier unified with the histidinyl transfer RNA repair polymerase Thg1 (ref. 32). Indeed, the OptS protomer observed in the crystal structure was previously predicted by Burroughs et al. as encoding an mCpol domain owing to homology established with the Cas10 polymerase domain from type III CRISPR immune systems[26]. GGDEF diguanylate cyclases synthesize cyclic diguanosine monophosphate (c-di-GMP) from guanosine triphosphate (GTP) precursors and Cas10 enzymes synthesize cyclic oligoadenylate molecules (between two and six AMPs) from adenosine triphosphate (ATP) precursors[33–35]. GGDEF proteins are defined by a conserved linear sequence motif (glycine–glycine–[aspartate/glutamate]–[aspartate/glutamate]–phenylalanine) that is nearly absent from the mCpol proteins *Kp* and *Vn*OptS, yet they still retain the characteristic β-hairpin structure and positioning of an acidic residue into the active site (Extended Data Fig. 2a). The degree of structural homology to GGDEF, Cas10 and other nucleotide polymerases implicates Panoptes OptS proteins in the enzymatic generation of signalling nucleotides.

## *Kp*OptS synthesizes 2′,3′-c-di-AMP

Purified *Kp*OptS was incubated with ribonucleotide triphosphates (ATP, GTP, cytidine triphosphate (CTP), uridine triphosphate (UTP)) and the reactions were monitored by high-performance liquid chromatography (HPLC) to detect any synthesis activity (Extended Data Fig. 2i). The largest abundance product observed was derived from ATP, with minor products requiring the co-incubation of both ATP and GTP (Fig. 2e and Extended Data Fig. 2j–l). We sought to determine the identity of the ATP-derived *Kp*OptS product and found that it had a similar retention time to an isomer of cyclic diadenosine monophosphate with mixed phosphodiester linkages, 2′,3′-c-di-AMP (c[A(2′–5′)pA(3′–5′)p]) (Fig. 2f–h). Further biochemical evidence for both the cyclic and mixed phosphodiester linkages of the product came from nuclease treatment experiments (Extended Data Fig. 2m). The product was resistant to calf-intestinal phosphatase (CIP, cleaves terminal phosphates), but was partially susceptible to P1 nuclease (cleaves 3′–5′ phosphodiester bonds) shifting to a new retention time. The combination CIP and P1 treatment led to loss of signal at retention times observed in the untreated and P1-alone treatment conditions and the appearance of a new, late-eluting peak that was consistent with A(2′–5′)pA generated by CIP treatment of commercially available linear pppA(2′–5′)pA. Liquid chromatography–mass spectrometry analysis of the *Kp*OptS reaction with ATP confirmed that the main product is indeed 2′,3′-c-di-AMP (Fig. 2g,h). On the basis of conservation with diverse nucleotidyltransferases, we expected that *Kp*OptS residues D6 and D57 would be involved in catalysis as they probably coordinate Mg$^{2+}$ ions for triphosphate stabilization in the binding pocket (Fig. 2b,i). Accordingly, mutation of these amino acids to asparagine alone or in tandem led to complete loss of cyclic dinucleotide production (Fig. 2j).

The Panoptes OptE effector protein is predicted to bind cyclic nucleotide products generated by OptS. We purified a soluble β-barrel-domain-only variant of the *Vn*OptE effector protein and characterized its thermal stability in response to a panel of c-di-AMP linkage isomers using differential scanning fluorimetry (Fig. 2k and Extended Data Fig. 3). Incubation with 2′,3′-c-di-AMP was able to cause an appreciable shift in stability with a change in melting temperature ($\Delta T_M$) of about 11 °C, whereas other isomers of c-di-AMP resulted in no observable shifts.

We next measured OptS synthesis of 2′,3′-c-di-AMP in vivo by interrogating bacterial cell lysates using a biochemical reporter assay (Fig. 2l and Extended Data Fig. 4; for gel source data, see Supplementary Fig. 1). Recombinant *Ab*Cap5, a CBASS nuclease that was activated by 2′,3′-c-di-AMP, was mixed with DNA and a nucleotide sample. In the absence of nucleotide, no DNA degradation was detected, but in the presence of 2′,3′-c-di-AMP, robust DNA degradation could be observed. DNA integrity could be visualized qualitatively by agarose gel electrophoresis or quantitatively by using a fluorescent probe for DNase activity and measuring the rate of hydrolysis. Experiments with synthetic nucleotide standards validated this assay and we next replaced the nucleotide standards with bacterial cell lysates. Nuclease activation could not be detected in lysates of *E. coli* expressing empty vector; however, the activity was robustly detected in OptSE-expressing cells (Extended Data Fig. 4). Our results estimate the intracellular concentration of *Ab*Cap5-activating nucleotide in OptSE-expressing bacteria to be about 30.4 nM and signal for bacteria expressing empty vector was below the limit of detection of 8.13 nM (Fig. 2l). Notably for this assay, *Ab*Cap5 would report the total concentration of 2′,3′-c-di-AMP and other nucleotides that activate the protein, such as 3′,2′-cyclic guanosine monophosphate-adenosine monophosphate (3′,2′-cGAMP). Further, we do not know the efficiency of recovery of intracellular nucleotide pools in this assay and cannot be sure we are not underestimating the true intracellular concentration. These factors may be important when considering these reported absolute values. These data support a model in which, under steady-state conditions, OptS constitutively produces 2′,3′-c-di-AMP.

## Panoptes systems use diverse nucleotides

We determined a 2.42 Å crystal structure of *Kp*OptS bound to the ATP substrate analogue α,β-methyleneadenosine-5′-triphosphate (ApCpp). Our data showed four independent ligand binding sites; however, inconsistent occupancy for the ribose and adenine nucleobase led us to confidently model only one complete ApCpp and three molecules of pCpp (Fig. 3a, Extended Data Fig. 5a,b and Extended Data Table 1). In the binding site, we find that the ribose from ApCpp and the α-phosphate of an adjacently bound ligand are poised for reaction within a distance of about 4–5 Å of each other (Fig. 3a). All ligands are stably bound to their respective protomers by means of divalent cations, here modelled as magnesium ions, which coordinate the triphosphate tail of the substrate. The metals are held in place as anticipated by active site aspartates D6 and D57, previously shown to be critical for complete catalytic activity. The binding pocket consists of amino acid contributions from two protomers, with the triphosphate bound to one protomer and the ribose and nucleobase positioned in an adjacent protomer, similar to what has been observed for ATP binding to Csm1 (Cas10)[30].

We investigated whether *Vn*OptS was also capable of producing 2′,3′-c-di-AMP in vitro. Surprisingly, when incubated with the four principal ribonucleotide triphosphates, *Vn*OptS generated a variety of cyclic nucleotide products in addition to 2′,3′-c-di-AMP (Fig. 3b and Extended Data Figs. 5c–l and 6). The most abundant product of the in vitro reactions was a linkage isomer of cyclic GMP-AMP, 3′,2′-cGAMP, and we also observed a significant amount of 3′,2′-cyclic uridine monophosphate-adenosine monophosphate (3′,2′-cUAMP) production. Our results support the idea that *Vn*OptS probably has

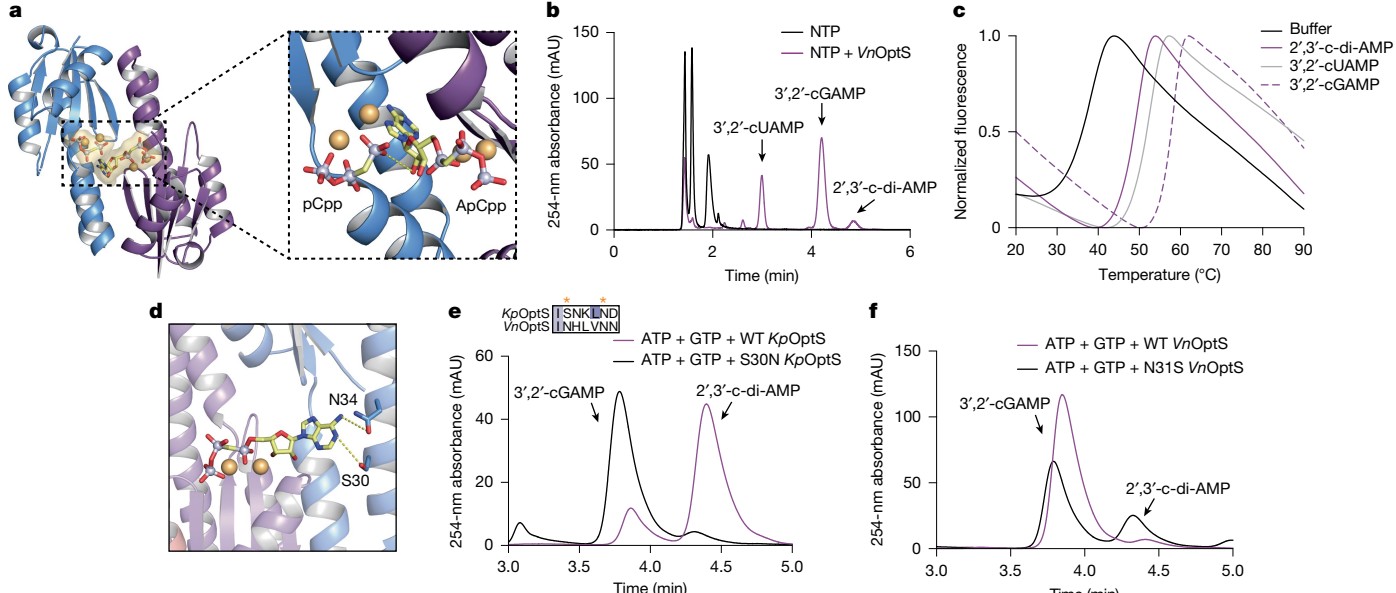

**Fig. 3 | Panoptes orthologues have altered cyclic nucleotide products.** **a**, Crystal structure of *Kp*OptS co-crystallized with non-hydrolysable ATP analogue ApCpp. Only two protomers are shown to highlight the orientation of analogues between two binding pockets (insufficient density to model the ribose and nucleobase for the second molecule of ApCpp). Inset, dashed line used to visualize the angle and close distance between reactive groups in the two substrate molecules. Divalent cation (probably Mg²⁺) shown as orange spheres. **b**, HPLC analysis of nucleoside triphosphate (NTP) (ATP, GTP, UTP, CTP) standards compared with the products of *Vn*OptS incubated with NTP. Arrows indicate known products verified through further experiments (Extended Data Figs. 2, 5 and 6). **c**, Thermal shift data showing the melt profile for a soluble version of *Vn*OptE incubated with the main cyclic dinucleotides produced by *Vn*OptS in vitro. Buffer indicates protein alone with no extra cyclic dinucleotide added. **d**, Nucleobase recognition site of *Kp*OptS with ApCpp highlighting potential interactions with Ser30 and Asn34. The metal/phosphate coordinating residues are in one protomer and the base recognition residues are in the other. **e**, HPLC analysis of reactions of *Kp*OptS (wild type or S30N mutant) with ATP and GTP showing altered product ratio. Inset, alignment of *Kp*OptS beginning at Ile29 and *Vn*OptS beginning at Ile30. Yellow asterisks indicate relevant positions of *Kp*OptS Ser30 and Asn34 from **d**. **f**, HPLC analysis of reactions of *Vn*OptS (wild type or N31S mutant) with ATP and GTP showing altered product ratio. For **b**, **c**, **e** and **f**, data are representative of *n* = 3 biological replicates.

reduced nucleobase selectivity, but that clearly at least one base of the final product must be adenine and the non-canonical bond forms between the 2′-OH of the adenosine and the 5′-α-phosphate of the second nucleotide (Extended Data Fig. 5k,l). As before, we tested the thermal stability of *Vn*OptE when incubated with these products as well as a larger panel of linkage isomers and a variety of other possible cyclic nucleotide ligands (Fig. 3c and Extended Data Fig. 3). In striking parallel with the proportion of cyclic dinucleotides produced by *Vn*OptS, shifts in melting temperature were most apparent for 3′,2′-cGAMP, 3′,2′-cUAMP and 2′,3′-c-di-AMP in order of decreasing average thermal shift. mCpol cyclases from Panoptes defence systems can therefore generate a variety of cyclic dinucleotide products and the cognate effector is capable of binding these signals.

We investigated the specific amino acid contacts that might be observed between the nucleobase of the ApCpp substrate analogue and *Kp*OptS (Fig. 3d). In the crystal structure, the adenine N-6 amino group is oriented towards an asparagine residue (N34) that is conserved between *Kp* and *Vn*OptS. The only observed contact that was predicted to differ in amino acid identity between the two proteins was S30 (N31 in *Vn*OptS). Although this position is S or N most frequently, it is highly variable across mCpol domains, suggesting a potential role in product nucleotide diversity. We generated mutant proteins for *Kp* and *Vn*OptS with active site swap substitutions at this location and found markedly altered product distributions for the two enzymes, relative to wild type (Fig. 3e,f). For the *Kp*OptS S30N mutant, there was a nearly complete inversion of the ratio of 2′,3′-c-di-AMP relative to 3′,2′-cGAMP produced in a reaction containing only ATP and GTP. Although more subtle, the corresponding mutant of *Vn*OptS (N31S) produced an increased proportion of 2′,3′-c-di-AMP and a decreased proportion of 3′,2′-cGAMP, relative to the wild type. These results agree

with prior studies on Cas10 and GGDEF proteins highlighting the role of serine and asparagine amino acid recognition of adenine and guanine bases, respectively[30,31].

## Acb2 is necessary to activate Panoptes

Phage T4 formed irregular plaques in soft agar overlays of OptSE-expressing bacteria (Extended Data Fig. 7a). At higher concentrations of phage, large clear plaques appeared and we isolated these escaper phages from three unrelated, clonal T4 lysates. Fifteen candidate escapers were plaque purified and genome sequenced along with their parent wild-type phages (Extended Data Fig. 7b). Out of 15 escaper phages, 14 encoded mutations exclusively in the anti-CBASS 2 gene (*acb2/vs.4*; Fig. 4a and Supplementary Table 2). Mutations included insertions and deletions that caused frame shifts, premature stop codons and large-scale deletions.

Acb2 was recently discovered to be an anti-defence protein that antagonizes CBASS immunity by acting as a nucleotide 'sponge', sequestering cyclic oligonucleotide signalling molecules[13,14,23]. Acb2 binding of CBASS-derived nucleotides interrupts downstream effector protein activation, allowing the phage to evade defence. We were surprised to find escaper phages with obvious loss-of-function mutations in *acb2* because this ruled out that our escapers had simply mutated Acb2 to increase affinity for the OptS-synthesized nucleotide. Instead, these data suggested that Acb2 was necessary for Panoptes-mediated defence.

We tested whether Acb2 was necessary for Panoptes-mediated defence by challenging empty vector or VnOptSE-expressing bacteria with either wild-type or Δ*acb2* T4 phages. Wild-type T4 efficiently replicated on empty-vector-expressing bacteria and was restricted

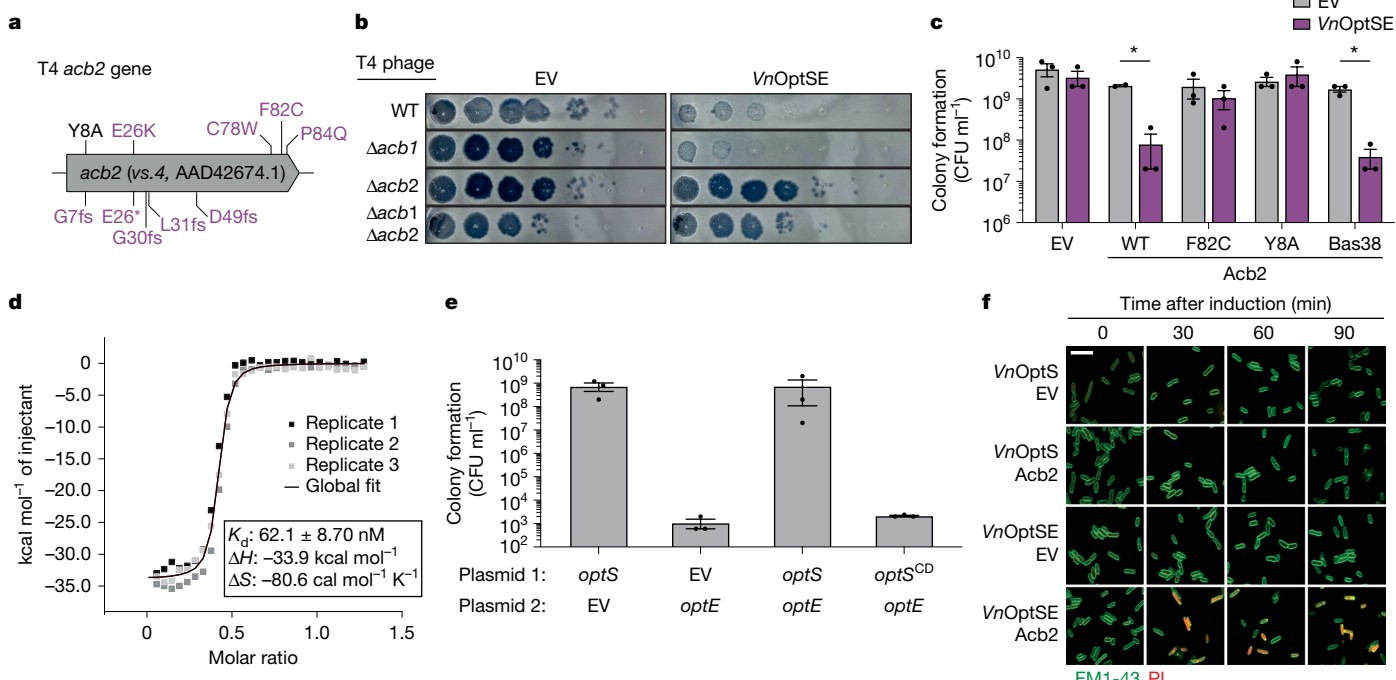

**Fig. 4 | Acb2 is necessary and sufficient to activate the Panoptes system.**
**a**, Schematic of phage T4 escape mutations from this paper (purple) and a previously investigated cyclic dinucleotide binding-site mutation (black[13,14]). **b**, Efficiency of plating of T4 phages with the indicated genotype on *E. coli* expressing an empty vector or the *Vn*OptSE operon from a plasmid. Images are representative of *n* = 3 biological replicates. **c**, Colony formation of *E. coli* expressing empty vector or *Vn*OptSE from a plasmid and indicated empty vector or *acb2* allele on an IPTG-inducible plasmid. Data are mean ± s.e.m. of *n* = 3 biological replicates, shown as individual points. A two-sided Student's *t*-test was used. *$P < 0.05$. **d**, ITC of 2′,3′-c-di-AMP interacting with Acb2. Data are *n* = 3 technical replicates. $K_d$, $\Delta H$ (change in enthalpy) and $\Delta S$ (change in entropy) were determined by generating a global fit for the three indicated replicates. Raw data are shown in Extended Data Fig. 8. **e**, Colony formation of *E. coli* co-expressing indicated *Vn*OptSE genes from IPTG-inducible (plasmid 1) and arabinose-inducible (plasmid 2) plasmids. CD, catalytically dead, OptS^D58A. Data represent the mean ± s.e.m. of *n* = 3 biological replicates, shown as individual points. **f**, Fluorescence microscopy of *E. coli* co-expressing the indicated *Vn*OptSE gene(s) from one plasmid and an empty vector or *acb2* allele on a second IPTG-inducible plasmid 0 min, 30 min, 60 min and 90 min after induction with IPTG. Bacteria were stained with membrane stain FM1-43 (green) and propidium iodide (PI; red). Images are representative of *n* = 2 biological replicates. Scale bar, 5 μm.

by VnOptSE. T4Δ*acb2*, however, was no longer restricted by OptSE and similarly replicated in both the empty-vector-expressing and OptSE-expressing bacteria (Fig. 4b). Phage T4 also encodes anti-CBASS 1 (*acb1*), a phosphodiesterase that degrades cyclic oligonucleotides to similarly antagonize CBASS signalling[12]. We constructed a T4Δ*acb1* phage and found no effect on OptSE-mediated defence (Fig. 4b and Extended Data Fig. 7c,d). Nevertheless, the T4Δ*acb1*Δ*acb2* double mutant seemed to be similar to T4Δ*acb2* alone. Acb2 was also important for *Kp*OptSE-dependent defence (Extended Data Fig. 7e). Taken together, these data suggest that the phage gene *acb2*, but not *acb1*, is necessary for OptSE-mediated defence.

## Acb2 is sufficient to activate Panoptes

We next explored whether *acb2* was sufficient to activate Panoptes in the absence of phage. OptE is a member of the S-2TMβ family of proteins that comprise a two-pass transmembrane element obligately fused to a soluble β-barrel nucleotide-binding domain[26]. Another member of this family, Cap15, is a CBASS effector that disrupts the bacterial inner membrane to initiate abortive infection[27]. We proposed that OptE might similarly disrupt bacterial host membranes and inhibit colony formation. Therefore, we co-expressed the *Vn*OptSE operon with *acb2* and assayed for colony formation. T4 *acb2* selectively inhibited colony formation only when co-expressed with the *Vn*OptSE operon (Fig. 4c). We observed the same phenotype when we expressed *acb2* from phage Bas38, which is restricted by the *optSE* operon by more than 1,000-fold. These data show that Acb2 is both necessary and sufficient to activate Panoptes defence.

Using the same co-expression assay, we found that the mutant *acb2* allele encoding Acb2^F82C identified in our escaper phage did not inhibit colony formation. The structure of Acb2 from phage T4 shows the Phe82 residue base-stacking with bound cyclic dinucleotides[14]. These data, along with other identified escaper mutations, suggested that Acb2 nucleotide binding was essential to activation of Panoptes. We tested this hypothesis using Acb2^Y8A, a mutation that has been previously shown to disrupt cyclic dinucleotide binding[13,14] (Fig. 4c). Acb2^Y8A co-expression with OptSE also failed to inhibit colony formation. These findings suggest that Acb2 binds nucleotides to activate Panoptes signalling.

Given that Acb2 activates Panoptes and is a phage 'sponge' that binds a wide array of nucleotide-derived products, we confirmed that Acb2 can bind the *Kp*OptS product, 2′,3′-c-di-AMP. HPLC experiments showed that incubation of purified T4 Acb2 with *Kp*OptS-derived 2′,3′-c-di-AMP depleted detectable signal (Extended Data Fig. 8a,b). When treated with proteinase K, Acb2 was degraded, and 2′,3′-c-di-AMP was once again detected by HPLC, indicating that Acb2 functions as a sponge of 2′,3′-c-di-AMP, rather than an active degrader. As expected, the escaper Acb2^F82C showed diminished binding (Extended Data Fig. 8c). Consistent with these results, isothermal titration calorimetry (ITC) demonstrated that Acb2 binds 2′,3′-c-di-AMP with an apparent dissociation constant ($K_d$) of around 62 nM (Fig. 4d and Extended Data Fig. 8d–f).

Our findings were initially paradoxical. Phage defence systems such as CBASS, Pycsar and Thoeris each generate nucleotide second messengers to activate defence. OptS and OptE are predicted to generate and to respond to cyclic oligonucleotides, respectively. Yet Acb2 nucleotide binding activated, rather than inhibited, Panoptes-mediated

defence (Fig. 4c). To explain these observations, we considered that, rather than OptS-synthesized nucleotides activating OptE, instead OptS-synthesized nucleotides inhibited OptE. If this were the case, OptE should inhibit growth unless an OptS-synthesized nucleotide is present.

We tested this hypothesis using plasmids that expressed *optS* and *optE* from isopropyl β-D-1-thiogalactopyranoside (IPTG)-inducible and arabinose-inducible promoters. Expression of OptS alone did not affect growth; however, expression of OptE alone potently inhibited colony formation (Fig. 4e). Growth of OptE-expressing strains could be restored by co-expressing wild-type OptS, but could not be restored using a catalytically dead OptS with mutated active site residues.

To explore the functional consequences of *Vn*OptSE activation in vivo, we co-expressed a plasmid containing either OptSE or OptS with a plasmid inducibly expressing *acb2* or an empty vector control. We monitored the cells using laser scanning confocal microscopy at a series of time points after induction with IPTG (Fig. 4f). These images showed that the co-expression of the full OptSE, but not OptS alone, with *acb2*, resulted in propidium iodide staining beginning 30 min after induction. The propidium iodide signal is indicative of bacterial membrane permeability, which is *acb2*- and OptE-dependent.

## Anti-defence proteins activate Panoptes

Phages encode a wide variety of anti-defence proteins that antagonize nucleotide-derived molecules in bacterial immune pathways. Thoeris anti-defence proteins 1 (Tad1) and 2 (Tad2) sequester both glycocyclic ADP-ribose (gcADPR) molecules synthesized by Thoeris systems and cyclic oligonucleotides synthesized by CBASS systems[15,18,25]. In addition, phages degrade nucleotide signalling molecules. Anti-Pycsar 1 (Apyc1), anti-CBASS 1 (Acb1) and type III anti-CRISPR (AcrIII-1) are phage-encoded phosphoesterases that degrade cyclic nucleotide molecules[12,19]. We proposed that these phage anti-defence proteins might also activate Panoptes through either sequestration or degradation of 2′,3′-c-di-AMP.

We co-expressed four Tad1 alleles with the *Vn*OptSE operon in *E. coli* and observed inhibition of colony formation when either *Cb*Tad1 or *Coli*Tad1 was expressed (Fig. 5a). We also tested four different Tad2 proteins and observed a slight reduction in colony formation when *Spt*Tad2 was expressed in the presence of OptSE. These data suggest that other phage 'sponge' proteins activate the Panoptes system, probably by binding and sequestering 2′,3′-c-di-AMP. On the other hand, we found the phage phosphoesterases Apyc1 or AcrIII-1 did not activate the Panoptes system (Fig. 5a). As Apyc1 and AcrIII-1 target cyclic mononucleotides and cyclic tetra-adenylates respectively[12,19], these data confirm Panoptes monitors only for cyclic dinucleotide-targeting proteins. Unexpectedly, expression of *acb1* reduced colony formation in this assay. However, deletion of *acb1* in phage T4 did not have an impact on *Vn*OptSE or *Kp*OptSE phage defence (Fig. 4b and Extended Data Fig. 7c–e). These conflicting findings may be due to higher than physiological expression of Acb1 in plasmid-based assays. These data indicate that a subset of phage immune evasion proteins can be detected by the Panoptes system, which may be dictated by their nucleotide preference.

## Distribution and gene linkage of mCpol

mCpol-containing systems are relatively rare but widely distributed across the two prokaryotic superkingdoms (Extended Data Fig. 9a and Supplementary Table 3). Like the GGDEF and CRISPR polymerase domains, they are strictly excluded from eukaryotes[32]. A total of 53% of all mCpol systems are encoded in genomes also coding for systems centred on SMODS-domain enzymes (cGAS/DncV-like nucleotidyltransferases (CD-NTases)), a highly significant linkage (P = 3.7 × 10⁻¹²) indicative of functional coupling between them (Fig. 5b and Extended Data Fig. 9b,c). Further, the mCpol genes show a similarly significant coupling with genes coding for members of the S-2TMβ family, such

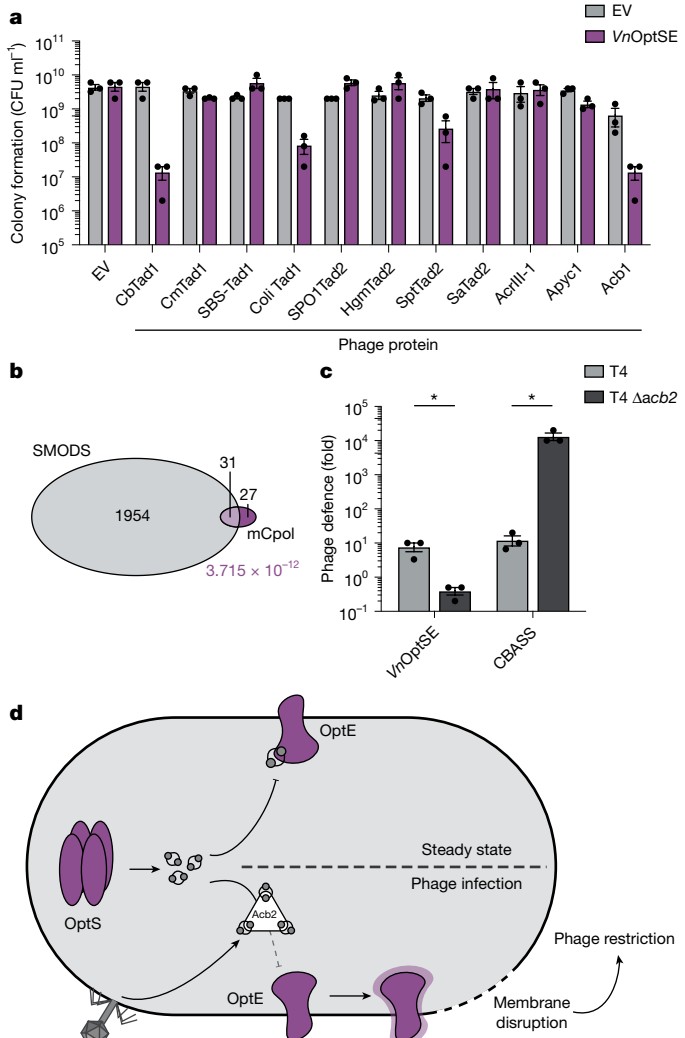

**Fig. 5 | Phage anti-defence proteins activate Panoptes. a**, Colony formation of *E. coli* expressing an empty vector or the *Vn*OptSE operon from one plasmid and empty vector or indicated phage protein on a second IPTG-inducible plasmid. Data represent the mean ± s.e.m. of *n* = 3 biological replicates, shown as individual points. **b**, Venn diagram of pairwise co-occurrence for mCpol and SMODS protein domains across prokaryotes. The *P* value (purple) represents the significance of pairwise co-occurrence from Fisher's exact test. **c**, Normalized phage defence provided by *Vn*OptSE or CBASS for wild-type T4 or T4 Δ*acb2* phage. *Escherichia coli* expressing *Vn*OptSE or CBASS were challenged with phages and fold defence was calculated for each phage by dividing the efficiency of plating (in PFU per millilitre) on empty vector by the efficiency of plating on *Vn*OptSE or CBASS-expressing bacteria. Data represent the mean ± s.e.m. of *n* = 3 biological replicates, shown as individual points. Efficiency of plating raw data are shown in Extended Data Fig. 7c (*Vn*OptSE) and Extended Data Fig. 7f (CBASS). A two-sided Student's *t*-test was used. *\*P < 0.05*. **d**, Model depicting the Panoptes system in steady state (top) and during infection by a phage that expresses Acb2 (bottom). OptS constitutively synthesizes cyclic nucleotides, which bind OptE and restrain activity. Upon infection, Acb2 sequesters the OptS-derived cyclic nucleotides from OptE. Activation of OptE ultimately leads to membrane disruption and restriction of phage replication.

as OptE (P = 6.9 × 10⁻²⁵). Notably, beyond co-occurring in the same genome as SMODS-domain proteins, in roughly half of these cases, Panoptes is also frequently coupled to CBASS systems in the same gene neighbourhood as immediately adjacent operons (Extended Data Fig. 9). In these instances, the coupled CBASS systems typically encode an effector with either the S-2TMβ or patatin-like phospholipase domain and a Ub-conjugation-like system implicated in the generation

of CD-NTase-target protein adducts[14,29,36,37]. These observations suggest that Panoptes systems are predominantly guardians of CBASS systems. To emphasize the selective benefit or disadvantage of phage T4 expressing *acb2*, we showed that, although expressing *acb2* is harmful for phage replication in the presence of the Panoptes defence system, it is required for circumventing a *Vibrio cholerae* CBASS system that synthesizes 3′,3′-cGAMP (Fig. 5c and Extended Data Fig. 7f).

Beyond coupling with S-2TMβ, 40% of the mCpol genes are in predicted operons with a gene coding for a protein with a 2TM segment coupled with a CRISPR-associated Rossmann fold (CARF) domain and a smaller set with a comparable gene encoding a similar 2TM protein with a SMODS-associated and fused to various effector domains (SAVED) domain in lieu of the CARF domain (Extended Data Fig. 9d). The CARF and SAVED domains are related Rossmannoid second-messenger sensor binding domains that take the place of the nucleotide-binding β-barrel domains of the S-2TMβ effectors[38]. Hence, these are predicted to be comparable effectors to the S-2TMβ proteins that are regulated by second-messenger nucleotides. Less frequently, the mCpol genes are coupled in operons or fused to higher eukaryotes and prokaryotes nucleotide-binding (HEPN) RNases and an array of predicted membrane perforating 2TM domains. This suggests that the mCpol-generated nucleotides are likely to regulate a wider range of effectors; however, it remains to be seen whether they are negative regulators as in the Panoptes system or positive regulators as in type III CRISPR and CBASS systems.

## Discussion

Here we discovered that the Panoptes antiphage system uses the vulnerability of a nucleotide-derived second messenger in immune signalling against phage by detecting the activity of counter-defence proteins. OptS constitutively synthesizes 2′,3′-c-di-AMP and other cyclic dinucleotides to hold OptE in an inactive state. During phage infection, Acb2 or similar immune evasion proteins are produced that sequester the OptS-derived signalling molecule, leading to a population of OptE that is no longer bound to cyclic dinucleotide and is free to become activated, in turn disrupting the bacterial membrane and resulting in phage defence (Fig. 5d). OptE is closely related to another member of the S-2TMβ family, Cap15, but each seems to use opposite signalling modalities. OptE inhibited growth in the absence of a cyclic nucleotide, whereas Cap15 inhibited growth in the presence of a cyclic nucleotide[27]. Cap15 effectors oligomerize when activated[27]. Therefore, we anticipate that the growth-inhibiting state of both proteins is the oligomer, which disrupts membrane integrity, but that subtle differences in the protein alter whether nucleotide binding stabilizes the monomer or oligomer. In support of that hypothesis, a contemporaneous report demonstrated that apo OptE forms an oligomer that is dispersed upon nucleotide binding[39].

The Panoptes system forces a dilemma for phage: should the phage lose Acb2 and be susceptible to CBASS or evade CBASS and risk agonizing Panoptes? Genome sequencing suggests that this evolutionary pressure is occurring in the wild. Although phage T4 has maintained a wild-type Acb2, related phage T2 encodes a mutation that probably selectively disrupts the Acb2 cyclic dinucleotide binding site (NC_054931, GenBank). Further, the related *Pseudomonas aeruginosa* phage PaMx41 encodes a premature stop codon in *acb2* (ref. 13). Phage can solve this dilemma by tuning the specificity of Acb2 away from the OptS-derived nucleotide, using an alternative CBASS-evasion protein such as Acb1 with different nucleotide specificity, or using CBASS antagonists that directly cripple the nucleotide synthase, a SMODS domain containing CD-NTase[6,7,26,40]. However, these responses by phage are limited by a key feature of the bacterial immune system: antiphage systems are distributed throughout that bacterial pangenome and are not necessarily co-resident in the same bacterium. Panoptes and CBASS can be encoded in the same genome so long as they use slightly different cyclic nucleotides to avoid cross-signalling. However, Panoptes and CBASS can also function in different genomes and use this separation to signal by means of an identical nucleotide. The latter circumstance would result in antiphage system incompatibility: Panoptes and CBASS that synthesize an identical nucleotide could not occur in the same bacterium. Antiphage system incompatibility may be an interesting, unappreciated layer to the ecology of phage defence gene distribution. Undoubtedly, Panoptes is partially responsible for the evolutionary pressures that have selected for a notable diversity of CD-NTase products found in CBASS systems.

Panoptes joins a growing list of antiphage systems that detect immune evasion, such as Phage Anti-Restriction-Induced System (PARIS), PrrC and Hailong[41–43]. A key distinction in these systems is how they detect immune evasion: either they directly bind an anti-defence protein or they sense anti-defence activity. PARIS and PrrC recognize anti-defence proteins, such as the restriction modification inhibitor Ocr[41,42]. Panoptes and Hailong recognize anti-defence activity using a decoy molecule, a cyclic nucleotide and single-stranded DNA, respectively. We speculate that bacteria may encode yet-to-be-discovered Panoptes-like systems that guard Pycsar- and Thoeris-derived second-messenger pools too. The discovery of these systems has the potential to identify further protein folds capable of synthesizing diverse nucleotide-derived second messengers.

Like bacteria, eukaryotes also guard immune pathways by surveying for immune evasion, often in a process called 'effector-triggered immunity'[44–46]. Mammals use nucleotide-derived second messengers for immune signalling and mammalian viruses degrade second messengers in the cGAS–STING and oligoadenylate synthetase (OAS)–RNaseL pathways using poxin and 2′,5′-phosphodiesterases, respectively[16,17]. Undiscovered layers of human immune pathways may therefore also exist that detect these viral evasion mechanisms in a manner similar to Panoptes defence systems. Intriguingly, there are many SMODS and Toll/interleukin-1 receptor domain enzymes of unknown function in several metazoans that could mediate these activities[47,48].

The Panoptes system represents a previously unreported reformulation of basic functional units that are used in other antiphage systems. The mCpol component seems to be a surviving remnant of an ancient antiviral system that was also elaborated and reformulated as the second-messenger signalling arm of a subset of the CRISPR systems[31,36]. For Panoptes, subtle molecular changes have enabled a new mechanism for sensing phage infection through detecting immune evasion. Although the phage seems to be powerless to escape the combination of CBASS and Panoptes, future research will surely uncover the inevitable phage-encoded anti-anti-anti-defence.

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

# Methods

## Bacterial strains and growth conditions

*Escherichia coli* strains that were used in this study are listed in Supplementary Table 4. All bacterial cultures were grown in 3.5 ml of media in 14-ml culture tubes shaking at 220 rpm at 37 °C, unless otherwise indicated. 'Overnight' cultures are defined as those that were grown for 16–20 h following inoculation from a single colony or glycerol stock. Where applicable, culture media was supplemented with carbenicillin (100 µg ml$^{-1}$) and/or chloramphenicol (20 µg ml$^{-1}$) for plasmid maintenance or strain selection. *Escherichia coli* OmniPir[49] was used for strain construction and storage of plasmids and *E. coli* MG1655 (CGSC6300) was used for all phage, colony formation and nucleotide extraction experiments.

All *E. coli* cultures used for cloning, strain construction, protein expression and indicated colony formation assays were grown in LB medium (1% tryptone, 0.5% yeast extract and 0.5% NaCl). All strains were frozen for long-term storage in LB plus 30% glycerol (v/v) at −70 °C. Strains used to perform phage amplification, phage infection assays, 2′,3′-c-diAMP extraction and measurement, and indicated colony formation assays were cultivated in 'MMCG' minimal medium (47.8 mM of Na$_2$HPO$_4$, 22 mM of KH$_2$PO$_4$, 18.7 mM of NH$_4$Cl, 8.6 mM of NaCl, 22.2 mM of glucose, 2 mM of MgSO$_4$, 100 mM of CaCl$_2$, 3 mM of thiamine and trace metals at 0.1× (Trace Metals Mixture T1001, Teknova; final concentration: 8.3 µM of FeCl$_3$, 2.7 µM of CaCl$_2$, 1.4 µM of MnCl$_2$, 1.8 µM of ZnSO$_4$, 370 nM of CoCl$_2$, 250 nM of CuCl$_2$, 350 nM of NiCl$_2$, 240 nM of Na$_2$MoO$_4$, 200 nM of Na$_2$SeO$_4$ and 200 nM of H$_3$BO$_3$)). When a strain with two plasmids was cultivated in MMCG medium, bacteria were grown in carbenicillin (20 µg ml$^{-1}$) and chloramphenicol (4 µg ml$^{-1}$). When growing strains that required induction, 500 µM of IPTG or 0.2% arabinose was used to induce, as appropriate. MMCG and LB agar plates contain 1.6% agar and media components described above.

## Plasmid construction

The plasmids used in this study are listed in Supplementary Table 4. Cloning and plasmid construction were performed as previously described[7]. In brief, genes of interest were amplified from phage genomic DNA or previously constructed plasmids using Q5 Hot Start High-Fidelity Master Mix (New England Biolabs (NEB)), or were synthesized as FragmentGENEs (Genewiz). Gene inserts were flanked by at least 18 base pairs of homology to the vector backbone outside the restriction digest sites. Ligation of genes into the digested, linearized backbone vector was done using modified Gibson Assembly[50] with HiFi DNA Assembly Master Mix (NEB). Gibson assemblies were transformed by electroporation into competent OmniPir and plated onto LB (1.6% agar) plates with appropriate antibiotics to select for successful transformations. Phage genes and *optS* with point mutations were generated by amplifying the gene of interest in two parts from a plasmid template, with the desired mutation occurring in the overlapping region between the two amplicons. Unless otherwise indicated, all enzymes were purchased from NEB.

For the *optSE* operon in a pLOCO3 backbone, complete vectors with the indicated operons were generated as ValueGENEs (Genewiz). The pLOCO3 vector (including superfolder green fluorescent protein (sfGFP)) was initially constructed using Gibson assembly to join and circularize two FragmentGENEs, one with the pLOCO3 backbone and one with the sfGFP gene (Genewiz; gene fragment sequences are listed in Supplementary Table 4).

For all vectors using the pTACxc backbone, pAW1608 was amplified and purified from OmniPir. Purified plasmid was then linearized using BamHI-HF and NotI-HF, or EcoRV-HF and PstI-HF. Gibson ligation was used to circularize the plasmid with the new insert.

For all vectors using the pBAD30 backbone, pAW1640 was amplified and purified from OmniPir. Purified plasmid was then linearized using EcoRI-HF and NotI-HF. Gibson ligation was used to circularize the plasmid with the new insert.

For all vectors using the pET16SUMO2 backbone, pAW1123 was amplified and purified from Sure1. Purified plasmid was then linearized using BamHI-HF and NotI-HF. Gibson ligation was used to circularize the plasmid with the new insert.

Plasmid sequences were verified with Sanger sequencing (Quintara Biosciences or Azenta) and/or Oxford Nanopore Sequencing (Plasmidsaurus). Reads were mapped to the predicted plasmid sequence using the Map to Reference feature of Geneious Prime (default settings).

## Phage amplification and storage

The phages used in this study are listed in Supplementary Table 4. Information on the Bacteriophage Selection for your Laboratory (BASEL) collection used in this study is reported in ref. 51. Phage lysates were generated through plate amplification using a modified double agar overlay[52]. For plate amplification, 400 ml of mid-log MG1655 were mixed with 3.5 ml of MMCG soft agar mix (MMCG with 0.35% agar and 10 mM of MgCl$_2$, 10 mM of CaCl$_2$ and 100 mM of MnCl$_2$) and 100–1,000 plaque-forming units (PFU). Plates were then incubated overnight at 37 °C. Phages were collected by adding 5 ml of SM buffer (100 mM of NaCl, 8 mM of MgSO$_4$, 50 mM of Tris-HCl (pH 7.5) and 0.01% gelatin) to the plate and incubating for 1 h at room temperature. To increase phage titre, the top agar overlay was scraped and collected along with the SM buffer. The SM buffer and top agar mixture was centrifuged at 4,000*g* for 10 min and the supernatant was transferred to a new tube. The resulting liquid was passed through a 0.2-mm filter or treated with two or three drops of chloroform, followed by vortexing, to remove viable bacteria. All amplified phages were stored at 4 °C in SM buffer.

## Phage infection assays

Phage infection assays and phage titre quantifications were performed using a modified double agar overlay technique[52]. Strains containing the indicated plasmids were cultivated overnight in MMCG or LB medium (including appropriate antibiotics) and were diluted 1:10 in fresh medium the following day. The bacteria were grown until they reached mid-logarithmic phase (OD$_{600}$ 0.1–0.8). A total of 400 µl of mid-log bacteria were mixed with 3.5 ml of 0.35% MMCG agar (plus 5 mM of MgCl$_2$ and 0.1 mM of MnCl$_2$) or LB agar (plus 10 mM of MgCl$_2$, 10 mM of CaCl$_2$ and 0.1 mM of MnCl$_2$) and poured on top of an MMCG (1.6% agar) or LB (1.6% agar) plate, respectively. The plate was allowed to cool for 15 min. Once cooled, 2 µl of a phage tenfold serial dilution series was spotted onto the soft agar overlay and allowed to dry, after which the plates were incubated at 37 °C overnight. Plates were imaged roughly 18–24 h after infection.

The resulting phage titre was quantified in PFU per millilitre for each phage lysate tested. PFU were enumerated on the basis of the lowest phage dilution spot with individual, quantifiable PFU. The dilution at that spot was used to calculate the PFU per millilitre appropriately. When there was a hazy zone of clearance, rather than identifiable plaques, the lowest phage concentration at which this was seen was counted as ten plaques. When no clearance was observed, the least dilute spot was counted as 0.9 plaques, and this was used as the limit of detection for the assay. Phage infection data were reported as PFU per millilitre ± standard error of the mean (s.e.m.) of *n* = 3 biological replicates.

## Phage infection time course in liquid culture

Bacterial strains containing the indicated plasmid were grown overnight in 25-ml MMCG or LB media plus appropriate antibiotics. Cultures were diluted to an OD$_{600}$ of 0.1 in fresh media without antibiotics (total volume, 30 ml) and were grown for 2 h at 37 °C with shaking at 220 rpm. After 2 h, the cultures were infected with phage at the indicated multiplicity of infection. The OD$_{600}$ of the cultures was measured at the indicated time points after infection. To enumerate PFU at each time

point, 250 μl of culture was collected and centrifuged at 20,000*g* for 5 min at 4 °C. The supernatant was transferred to a new tube and three to five drops of chloroform were added, followed by vortexing, to kill any bacteria that remained. The resulting lysates were titred using the phage infection assay protocol explained above.

### Escaper phage generation and amplification

T4 escaper phages were generated from three unrelated, clonal T4 lysates ('parents') that were separately plate amplified on wild-type *E. coli* MG1655. To make the T4 escaper phages ('daughters'), 400 μl of mid-log bacteria expressing the *optSE* operon (in MMCG plus 100 μg ml$^{-1}$ carbenicillin) was mixed with 100 μl of parent T4 lysate (about $4 \times 10^5$ PFU) and 3.5 ml of MMCG top agar and poured onto an MMCG agar plate. The plate was allowed to dry and was incubated overnight at 37 °C. The next day, five single escaper plaques were individually isolated from each parent T4 plate using a Pasteur pipette, soaked in 500 μl of SM buffer in an Eppendorf tube and filter sterilized using a Nanosep 0.2-μm spin filter (Pall Labs). A dilution series of each T4 escaper phage was spot-plated onto *E. coli* MG1655 expressing Panoptes to confirm replication in the presence of OptSE. From this plate, single plaques from each escaper were individually purified and plate amplified (as described in the phage amplification and storage protocol above) before storage.

### Phage genome sequencing and escaper analysis

The genomes of the parent and escaper T4 phages were purified as previously described[53]. To do this, 450 ml of phage lysate (more than $10^8$ PFU ml$^{-1}$) was mixed with 50 μl of 10× DNAse I buffer (100 mM of Tris-HCl (pH 7.6), 25 mM of MgCl$_2$ and 5 mM of CaCl$_2$) and treated with DNAse I (final concentration $2 \times 10^{-3}$ U μl$^{-1}$) and RNAse A (final concentration $2 \times 10^{-2}$ mg ml$^{-1}$). This mixture was incubated for 1.5 h at 37 °C to remove extracellular nucleic acids. After, EDTA was added to a final concentration 20 mM to stop the reaction. Each parent and escaper phage genome was then isolated and purified using the Qiagen DNeasy Cleanup Kit, starting at the proteinase K digestion step[53].

The purified phage genomes were prepared for Illumina sequencing using a modification of the Nextera kit protocol as previously described[54]. Illumina sequencing was performed using a MiSeq V2 Micro 300-cycle kit (SeqCenter). The resulting reads were mapped to Genome accession AF158101.6 using Geneious software's 'Map to Reference' feature. Each read was trimmed to remove the Nextera adapter sequences before mapping (sequence trimmed: AGATGTGTATAAGA GACAG) using the 'Trim sequences' option; otherwise, Geneious default settings were used. The trimmed sequences were mapped to the phage genome using default settings with the 'Map to Reference(s)' feature. The Geneious feature 'Find Variations/SNPs' was used to identify variants in daughter phage genomes. Called variants were identified as escaper mutations if they were present in 75% or more of reads and were not present in parent phage genomes.

### Construction of phage gene deletions

T4 knockout phages were generated as previously described[55]. In brief, 5 ml of *E. coli* MG1655 expressing a pET vector encoding the template for homologous repair were grown in LB to mid-log phase. Bacteria were then infected with about $4 \times 10^8$ PFU of T4 and grown for 2–3 h before collecting lysate. Phage lysates amplified in this way were then mixed with 400 μl of mid-log bacteria expressing eLbuCas13a constructs with spacers targeting the gene of interest and poured onto an MMCG agar plate as in the method described in solid plate amplification, detailed above. Individual plaques were isolated and spot-plated on *E. coli* MG1655 expressing the same spacer to confirm mutation and to plaque-purify each clone. Target gene deletion was validated using PCR. Homologous repair templates encoded 250 bp of homology on either side of the target gene. In-frame deletions retained the first and last six amino acids of the target gene, but deleted the intervening sequence.

Two 31-nt spacers were selected to target the beginning of each gene and induced as needed using anhydrotetracycline at 5 nM in the top agar of the soft agar overlay.

The *acb1* knockout was constructed with repair template pEK0220 and spacers pEK0223 and pEK0224. The knockout was PCR verified using the primers oEK0490 and oEK0491.

The *acb2* knockout was constructed with repair template pEK0221 and spacers pEK0225 and pEK0226. The knockout was PCR verified using the primers oEK0492 and oEK0493.

The double knockout was created by doing the Δ*acb2* knockout steps on a confirmed Δ*acb1* knockout phage.

### Colony formation assays for bacterial growth inhibition analysis

Bacterial growth inhibition was tested using colony formation assays. Bacterial strains with indicated plasmids were grown overnight in MMCG media plus appropriate antibiotics. The cultures were then tenfold serially diluted in fresh MMCG media (without antibiotics) and 5 μl of each dilution was spotted onto an MMCG agar plate containing the appropriate antibiotics, as well as IPTG (500 μM; induced condition) as indicated. Data in Fig. 4e were collected using LB media with appropriate antibiotics, with or without glucose (0.2% w/v; uninduced condition), IPTG (500 μM; induced condition) or arabinose (0.2% w/v; induced condition), as indicated. After the spotted bacteria were allowed to dry, plates were incubated at 37 °C for roughly 16–18 h for LB plates or around 24 h for MMCG plates. Growth inhibition was quantified the next day by counting the number of colony-forming units (CFU) of the lowest dilution that had individual colonies. When no individual colonies could be counted, the lowest bacterial concentration at which growth was observed was counted as 10 CFU. In instances where no growth was visible, the least dilute spot was counted as 0.9 CFU and used as the limit of detection. Colony formation data were reported as CFU per millilitre ± s.e.m. of $n = 3$ biological replicates.

### Acb2 protein expression and isothermal titration calorimetry

The vector expressing WT 6xHis-hSUMO2-Acb2 was transformed into Rosetta2 expressing the pRARE2 plasmid and plated onto 1.6% MMCG agar plates, plus 100 μg ml$^{-1}$ of carbenicillin and 20 μg ml$^{-1}$ of chloramphenicol. An individual colony was picked the following day and inoculated into 100 ml of M9ZB media (47.8 mM of Na$_2$HPO$_4$, 22 mM of KH$_2$PO$_4$, 18.7 mM of NH$_4$Cl, 85.6 mM of NaCl, 1% Casamino acids (VWR), 0.5% v/v glycerol, 2 mM of MgSO$_4$, trace metals at 0.5× (Trace Metals Mixture) plus 100 μg ml$^{-1}$ of carbenicillin and 20 μg ml$^{-1}$ of chloramphenicol. The culture was then grown overnight, shaking at 37 °C and 220 rpm. The following day, the culture was used to inoculate 2 l of the same, fresh media to an OD$_{600}$ of 0.05, then grown to an OD$_{600}$ of about 1.5. Cultures were crash-cooled on ice for 30 min before IPTG was added to 500 μM to induce protein expression. The culture was then moved to a 16-°C shaking incubator and allowed to grow overnight.

Cultures were collected by centrifugation for 30 min at 4,600*g* and 4 °C in an Avanti JXN-26 Floor Centrifuge using the JXN 12.500 rotor (Beckman Coulter). The resulting pellets were resuspended in 40 ml of lysis buffer (20 mM of HEPES (pH 7.5), 400 mM of NaCl, 10% v/v glycerol, 20 mM of imidazole, 0.1 mM of dithiothreitol (DTT)). After resuspension, cells were lysed by sonication at 80% amplitude, with 15-s-on, 45-s-off pulses for a total processing time of 10 min using a Q500 sonicator (Qsonica). Cellular debris was removed from sonicated lysates by centrifugation for 45 min at 4 °C and 16,000*g* in an Avanti JXN-26 Floor Centrifuge using the JA 25.50 rotor (Beckman Coulter). The soluble lysate was then decanted and protein was purified using immobilized metal affinity chromatography. Briefly, the soluble lysate was run over 2 ml of HisPure cobalt slurry (Fisher Scientific) equilibrated in lysis buffer. The resin was then washed with 2 × 25 ml of wash buffer (20 mM of HEPES (pH 7.5), 1 M of NaCl, 10% v/v glycerol, 20 mM of imidazole, 0.1 mM of DTT) and protein was eluted in 10 ml of elution buffer (20 mM of HEPES (pH 7.5), 400 mM of NaCl, 10% v/v glycerol, 300 mM

of imidazole, 0.1 mM of DTT). Proteins were then dialysed against 2 × 1 l of dialysis buffer (20 mM of HEPES (pH 7.5), 250 mM of KCl, 0.1 mM of DTT) overnight at 4 °C using 3.5-kDa molecular weight cut-off (MWCO) SnakeSkin Dialysis Tubing (VWR). The 6×His-SUMO-tag was cleaved using 6×His-hSENP2 (produced in-house; final concentration of 1:100 hSENP2:protein w/w) during the overnight dialysis step. After dialysis, proteins were run over 2 ml of HisPure cobalt slurry equilibrated in dialysis buffer to remove any 6×His-SUMO tagged proteins.

After dialysis, the protein was concentrated as needed using 3-kDa MWCO Nanosep spin concentration columns (Pall Labs) and stored as 200–500-μl aliquots in dialysis buffer at −70 °C. Protein concentrations were measured using $A_{280}$ on a Nanodrop One[C] (Thermo Fisher Scientific) and protein purity was visualized using SDS-PAGE followed by Coomassie staining.

ITC assays were adapted from the protocol described in ref. 13. In brief, the $K_d$, $\Delta H$ and $\Delta S$ for the binding of WT Acb2 with 2′,3′-c-di-AMP were determined using a MicroCal ITC200 calorimeter. Purified Acb2 and 2′,3′-c-di-AMP were dialysed into the ITC buffer (20 mM of HEPES (pH 7.5) and 200 mM of NaCl) at 4 °C overnight. The titration was carried out with 27 successive 1.5-μl injections of 150 μM of 2′,3′-c-di-AMP into the sample cell containing 50 μM of WT Acb2. Each injection was spaced by 180 s, and the cell was kept at 25 °C and stirred at 750 rpm. Origin software was used for integration and global curve fitting with a 'one set of sites' binding model. We selected the 'one set of sites' model, which is appropriate to use for binding between a macromolecule and a ligand that includes any number of binding sites in which all sites have the same association constant ($K_a$) and $\Delta H$.

## Laser scanning confocal microscopy

Strains were grown overnight in LB plus appropriate antibiotics and were diluted 1:10 into fresh medium the following day. Strains were grown until they reached mid-logarithmic phase ($OD_{600}$ 0.1–0.8) and were then normalized to an $OD_{600}$ of 0.55 before induction with 500 μM of IPTG. At each time point, culture samples were removed from the liquid culture, OD normalized to 0.55 in a volume of 100 μl with fresh MMCG medium and stained with 25 μg ml⁻¹ of DAPI, 5 μg ml⁻¹ of FM1-43 and 5 μM of propidium iodide by incubating samples with dyes for 2 min at room temperature. A total of 5 μl of each sample was pipetted onto an MMCG imaging pad containing 500 μM of IPTG and allowed to dry for about 5 min before imaging. Laser scanning confocal microscopy was performed on a Nikon A1R microscope. For FM1-43, a 488-nm laser and a Chroma ET525/50m emission filter were used. For propidium iodide, a 561-nm laser and a Chroma ET600/50m emission filter were used. All images were acquired using the same laser power and gain settings. For each strain condition and time point, a 3 × 3 image scan was collected; representative areas in these scans are shown in Fig. 4f.

## AbCap5 protein expression and purification

The vector for expressing 6×His-hSUMO2-AbCap5 was transformed into Rosetta2 expressing the pRARE2 plasmid and plated onto 1.6% LB agar plates (plus 100 μg ml⁻¹ of carbenicillin and 20 μg ml⁻¹ of chloramphenicol). The plate was allowed to dry and incubated at 37 °C overnight. The next day, an individual colony was picked to inoculate 25 ml of liquid LB media (plus 100 μg ml⁻¹ of carbenicillin and 20 μg ml⁻¹ of chloramphenicol), which was then grown overnight shaking at 220 rpm and 37 °C. The following day, the overnight culture was diluted 1:100 into fresh media plus antibiotics, then grown to an $OD_{600}$ of around 0.6 before IPTG addition to a final concentration of 500 μM to induce protein expression. Each culture flask was then moved to a 16-°C shaking incubator (220 rpm) and allowed to grow overnight.

The next day, cultures were collected by centrifugation for 30 min at 4,600g and 4 °C in an Avanti JXN-26 Centrifuge using the JXN 12.500 rotor (Beckman Coulter). Each bacterial pellet was resuspended to a total volume of 40 ml of lysis buffer (20 mM of HEPES (pH 7.5), 400 mM of NaCl, 10% v/v glycerol, 30 mM of imidazole and 1 mM of DTT) plus

1 μl of Pierce Universal Nuclease (Thermo Fisher Scientific). The resuspended bacterial pellets were kept chilled and were lysed by sonication at 80% amplitude, with 30 s on, followed by 30 s off, for a total processing time of 30 min using a Sonicator 4000 (Misonix). The lysed bacteria were centrifuged at 4 °C for 1 h at 14,000g in a 5910 R tabletop centrifuge (Eppendorf) to pellet cellular debris left over from sonication.

The resulting soluble lysate was transferred to a new conical tube and kept on ice. To purify protein, the entire soluble lysate was run over 1 ml of lysis buffer-equilibrated Ni-NTA resin (Thermo Fisher Scientific). The flow-through was collected and reapplied to the resin. The resin was then washed with 5 × 25 ml of wash buffer (20 mM of HEPES (pH 7.5), 1 M of NaCl, 10% v/v glycerol, 30 mM of imidazole and 1 mM of DTT). The protein was eluted in 10 ml of elution buffer (20 mM of HEPES (pH 7.5), 400 mM of NaCl, 10% v/v glycerol, 300 mM of imidazole and 1 mM of DTT). The eluted protein was added to 10-kDa MWCO tubing (VWR) and then dialysed in 1 l of dialysis buffer (20 mM of HEPES (pH 7.5), 250 mM of KCl and 1 mM of DTT) for 1 h at 4 °C. Afterwards, the dialysis tubing and protein were placed into 1 l of fresh dialysis buffer and allowed to dialyse overnight at 4 °C. The 6×His-SUMO-tag was cleaved from the N terminus of AbCap5 using 6×His-hSENP2 (purified in-house; final concentration of 1:100 hSENP2:protein w/w), which was added to the purified protein immediately before it was placed in the dialysis tubing. After dialysis and 6×His-SUMO cleavage, the purified proteins were applied to 1 ml of dialysis buffer-equilibrated Ni-NTA and the flow-through was collected and reapplied to the resin to remove any uncleaved 6×His-SUMO tagged proteins.

The purified AbCap5 was concentrated as needed using 30-kDa MWCO Macrosep spin concentration columns (Pall Labs) and stored in 1-ml aliquots in 50% glycerol v/v at −20 °C. A Nanodrop One[C] (Thermo Fisher Scientific) was used to measure $A_{280}$ to determine the protein concentration. Protein purity was determined by using SDS-PAGE followed by Coomassie staining. The AbCap5 protein purified in this way was used in nuclease assays for the measurement of intracellular 2′,3′-c-di-AMP.

## Determination of AbCap5 nucleotide specificity

For each 30-μl reaction, AbCap5 (final concentration 325 nM) was incubated with 500 ng of linear PCR-amplified DNA and 3 μl of the indicated nucleotide (final concentration 100 nM) in reaction buffer (final concentrations: 10 mM of Tris-HCl (pH 7.5), 25 mM of KCl and 10 mM of $MgCl_2$). The mixture was incubated for 1 h at 37 °C, then boiled for 5 min at 95 °C. Reactions were mixed with 6 μl of 6× Gel Loading Dye plus SDS (NEB) and 20 μl was loaded and visualized on a 2% (w/v) agarose gel stained with SYBR Safe (Thermo Fisher Scientific).

## Bacterial lysate preparation and nucleotide extraction

Bacterial strains expressing either the *optSE* operon or an empty vector were grown overnight in 25 ml of MMCG plus carbenicillin (100 μg ml⁻¹). The following day, the overnight culture was diluted 1:10 in 100 ml (total volume) of fresh media (without antibiotics) and was allowed to grow to an $OD_{600}$ of about 0.6–0.7 (noting the specific $OD_{600}$ at collection). Once the appropriate $OD_{600}$ was reached, the cultures were centrifuged at 4,000g for 10 min at 4 °C in a 5910 R tabletop centrifuge (Eppendorf) to pellet the bacteria. The resulting supernatant was discarded, and the bacterial pellet was resuspended in 500 μl of lysis buffer (10 mM of Tris-HCl (pH 7.5) and 25 mM of KCl) and transferred to a new Eppendorf tube. Hen-lysozyme (0.2 mg ml⁻¹ final concentration; VWR) and roughly 200 μl of zirconium beads were added before boiling at 95 °C for 2.5 min and then vortexing at max speed for 2 min. The samples were boiled and vortexed again and then treated with proteinase K (0.03 mg ml⁻¹ final concentration; Qiagen) and 1-μl Pierce Universal Nuclease (Thermo Fisher Scientific) for 30 min at 37 °C. After treatment, the samples were boiled and vortexed once again and Triton-X was added to a final concentration of 0.5% v/v. The samples underwent a final boil and vortex, and were centrifuged at 17,500g for 10 min at 4 °C.

The resulting lysate/supernatant was transferred to a new Eppendorf tube and kept on ice until ready for nucleotide extraction.

Nucleotides were extracted from bacterial lysates using a modified version of a previously described phenol-chloroform/chloroform extraction protocol[13]. In brief, 450 µl of phenol-chloroform (Thermo Fisher Scientific) was added to 450 µl of bacterial lysate (prepared above), vortexed for 30 s and then centrifuged at 17,500$g$ for 45 min at 4 °C. A total of 400 µl of the top aqueous layer was carefully removed and added to 400 µl of fresh phenol-chloroform in a new tube. This mixture was vortexed for 30 s and centrifuged at 17,500$g$ for 10 min at 4 °C. A total of 350 µl of the resulting top aqueous layer was carefully removed and added to 350 µl of chloroform (VWR) in a new tube. The mixture was vortexed again for 30 s and centrifuged one final time at 17,500$g$ for 10 min at 4 °C. The resulting top aqueous layer was removed and placed in a new tube for storage at −20 °C. Nucleotide extraction from indicated strains was carried out for a total of $n = 3$ biological replicates.

## AbCap5-based intracellular 2′,3′-c-di-AMP measurements

Commercially available 2′,3′-c-di-AMP (Enzo) was used to make a standard curve by twofold serially diluting the nucleotide in nuclease-free water. A dilution series of nucleotide extracts from the tested bacterial strains were made by twofold serially diluting the extracts in nuclease-free water. For each 25-µl reaction, AbCap5 (final concentration 50 nM) was incubated with 500 ng of linear PCR-amplified DNA and 5 µl of one dilution of nucleotide extract, 2′,3′-c-di-AMP or water in reaction buffer (final concentrations: 10 mM of Tris-HCl (pH 7.5), 25 mM of KCl and 10 mM of MgCl$_2$). The mixture was incubated for 2 h at 37 °C, and then boiled at 95 °C for 5 min. Reactions were mixed with 5 µl of 6× Gel Loading Dye plus SDS (NEB) and 20 µl were loaded and visualized on a 2% (w/v) agarose gel stained with SYBR Safe (Thermo Fisher Scientific).

The concentration of intracellular 2′,3′-c-di-AMP ([n.t.]) was calculated using the following equation:

$$[\text{n.t.}] = \frac{(\text{SC}) \times (\text{lysate volume})}{(\text{CV}) \times (\text{OD}_{600}) \times (\text{culture volume})}$$

SC is the upper or lower 2′,3′-c-di-AMP standard curve concentration (in [M]) that corresponds to a similar DNA degradation amount as the last visible nucleotide extract dilution reaction (from OptSE-expressing cells) that contains non-degraded or partially degraded DNA PCR product, respectively. Lysate volume is the volume of lysis buffer (in litres) that was used to resuspend the bacterial pellet before the lysis steps. CV is the OD$_{600}$-specific cell volume (in litres per OD per millilitre) of the bacteria based on the *E. coli* strain and growth conditions[56]. OD$_{600}$ is the optical density of the bacterial culture at the time of collection (before pelleting). Culture volume is the volume of bacterial culture (in millilitres) at the time of collection. Intracellular [2′,3′-c-di-AMP] is reported as an average for the upper and lower range of concentrations (in nanomolars) for $n = 3$ biological replicates of the nucleotide extraction.

## DNase alert-based intracellular 2′,3′-c-di-AMP measurements

Single-use DNase alert (Integrated DNA Technologies) was resuspended in 43 µl of 10× nuclease buffer per tube (1 M of Tris-HCl (pH 7.5), 2.5 M of KCl, 1 M of MgCl$_2$). The following was mixed in each well of a black 96-well, clear-bottom plate: 10 µl of 10× DNase alert, 35 µl of salmon sperm DNA (20 ng µl$^{-1}$), 5 µl of 2′3′-c-di-AMP nucleotide standard or bacterial lysate and 50 µl of AbCap5 (50 nM). The reaction was monitored at an excitation/emission of 536/556 nm in a TECAN Spark plate reader at 37 °C every 2 min for 30 min.

GraphPad Prism was used to plot the data and determine the velocity of the reaction (that is, slope measured in relative fluorescence units per minute) in the linear range. A chemical standard of 2′,3′-c-di-AMP

was used to determine a standard curve by means of linear regression ([agonist] versus response (three parameters); $Y = \text{bottom} + X \times (\text{top} - \text{bottom})/(\text{EC50} + X)$), which allowed for interpolation of nucleotide concentration in the well from bacterial lysate reaction velocities.

The concentration of intracellular 2′,3′-c-di-AMP ([n.t.]) was calculated using the following equation:

$$[\text{n.t.}] = \frac{([\text{lysate}]) \times (\text{lysate volume})}{(\text{CV}) \times (\text{OD}_{600}) \times (\text{culture volume})}$$

[Lysate] is the concentration of 2′,3′-c-di-AMP (in M) in the bacterial lysate, determined by multiplying the interpolated nucleotide concentration in the well by the appropriate dilution factor of the bacterial lysate in the reaction. Lysate volume is the volume of lysis buffer (in litres) that was used to resuspend the bacterial pellet before the lysis steps. CV is the OD$_{600}$-specific cell volume (in litres per OD per millilitre) of the bacteria based on the *E. coli* strain and growth conditions[56]. OD$_{600}$ is the optical density of the bacterial culture at the time of collection (before pelleting). Culture volume is the volume of bacterial culture (in millilitres) at the time of collection. Intracellular [2′,3′-c-di-AMP] is reported as an average (in nanomolars) for $n = 3$ biological replicates of the nucleotide extraction.

## Recombinant protein expression and purification

The genes for full-length *optS* from *K. pneumoniae* strain KP67, *optS* from *V. navarrensis*, *optE* from *V. navarrensis* (soluble β-barrel only, residues S73-END) and bacteriophage T4 *acb2* were codon optimized and synthesized as double-stranded DNA fragments (gBlocks; Integrated DNA Technologies). The genes were cloned using Gibson assembly into a linearized (restriction enzyme digested with BamHI and NotI) in-house pET16 expression vector modified to allow in-frame cloning of an N-terminal hexa-histidine tag with or without a hSUMO2 tag with a Gly-Ser linker. After Sanger sequencing confirmation, the resulting plasmids were subsequently transformed into BL21-RIL *E. coli* cells (BL21 derivative, DE3; Invitrogen, Life Technologies) for protein expression.

Transformant colonies were grown overnight at 37 °C on MDG media (0.5% glucose, 25 mM of Na$_2$HPO$_4$, 25 mM of KH$_2$PO$_4$, 50 mM of NH$_4$Cl, 5 mM of Na$_2$SO$_4$, 2 mM of MgSO$_4$, 0.25% aspartic acid and trace metals) plates supplemented with 100 µg ml$^{-1}$ of ampicillin and 34 µg ml$^{-1}$ of chloramphenicol for selection. Single colonies were used to inoculate liquid MDG media cultures (same components without agar), which were grown overnight for about 16–20 h at 37 °C and 230-rpm shaking. Overnight cultures were used to inoculate 1 l of liquid M9ZB expression media (0.5% glycerol, 1% Cas-amino acids, 47.8 mM of Na$_2$HPO$_4$, 22 mM of KH$_2$PO$_4$, 18.7 mM of NH$_4$Cl, 85.6 mM of NaCl, 2 mM of MgSO$_4$ and trace metals) contained in 2.5-l flasks. After cultures reached an OD$_{600}$ of greater than 2.5, flasks were placed on ice for 15 min; following this, 0.5 mM of IPTG (final) was added and the mixture was incubated at 16 °C with 230-rpm shaking overnight (about 16–20 h). Cells were collected by centrifugation, washed with PBS buffer and flash-frozen with liquid N$_2$ and stored at −80 °C until needed.

Conventional nickel-affinity chromatography was carried out using gravity flow at 4 °C. Briefly, *E. coli* cell pellets were resuspended with lysis buffer (20 mM of HEPES-KOH (pH 7.5), 400 mM of NaCl, 10% glycerol, 30 mM of imidazole (pH 7.5), 1 mM of DTT) and subjected to sonication to release cellular contents (10 s on, 20 s off, 70% amplitude, 5 min total on time). The lysate was clarified with centrifugation and the supernatant loaded onto 4–6 ml of packed Ni-NTA resin pre-equilibrated with lysis buffer. The column was then sequentially washed with 20 ml of lysis buffer, 70 ml of wash buffer (20 mM of HEPES-KOH (pH 7.5), 1 M of NaCl, 10% glycerol, 30 mM of imidazole (pH 7.5), 1 mM of DTT) and a final 35 ml of lysis buffer to remove high concentrations of salt. Protein was eluted with 20 ml of elution buffer (20 mM of HEPES-KOH (pH 7.5), 400 mM of NaCl, 10% glycerol, 300 mM of imidazole (pH 7.5), 1 mM of DTT). The eluent was dialysed overnight

in 20 mM of HEPES-KOH (pH 7.5), 250 mM of KCl and 1 mM of DTT at 4 °C with gentle stirring. In the case of Acb2, hSENP2 protease (D364–L589, M497A) was added to the dialysis tubing to cleave the N6xHis-SUMO2 tag. The dialysed eluent was concentrated and further purified through size-exclusion chromatography using a HiLoad 16/600 Superdex 200 pg column (Cytiva) equilibrated with 20 mM of HEPES (pH 7.5), 250 mM of KCl and 1 mM of TCEP-KOH. Size-exclusion chromatography fractions were analysed with SDS-PAGE; fractions that contained recombinant proteins of interest were pooled and concentrated to more than 10 mg ml⁻¹, were flash-frozen in liquid $N_2$ and then stored at −80 °C until needed.

### Protein crystallization and structure determination

Apoprotein crystals for OptS from *K. pneumoniae* strain KP67 were obtained at room temperature by mixing 1 µl of reservoir solution consisting of 8% (w/v) PEG 4000 with an equal volume of 7 mg ml⁻¹ of the protein solution using the hanging-drop vapour diffusion method. The crystals were soaked in mother liquor with 15% ethylene glycol as cryoprotectant and subsequently plunged into liquid $N_2$ and shipped for remote data collection.

Crystals for *Kp*OptS bound to ATP analogue ApCpp were grown at room temperature using the hanging-drop vapour diffusion method in 16% w/v PEG 4000 (reservoir solution). *Kp*OptS was diluted to 7 mg ml⁻¹ with buffer containing 100 mM of KCl, 20 mM of HEPES-KOH (pH 7.5), 1 mM of TCEP with the addition of 1 mM of ApCpp, 10 mM of $MgCl_2$ and 1 mM of $MnCl_2$ at their final concentration. The crystals were grown in 15-well trays (NeXtal) containing 350 µl of reservoir solution and 2-µl drops. Drops were mixed 1:1 with the purified protein-ligand mixture and reservoir solution. Crystals were cryoprotected with reservoir solution supplemented with 30% glycerol with the addition of 1 mM of ApCpp, 10 mM of $MgCl_2$ and 1 mM of $MnCl_2$ and collected by flash-freezing in liquid nitrogen.

X-ray diffraction data for apo *Kp*OptS were collected at experimental station 12-2 at the Stanford Synchrotron Radiation Lightsource (SSRL) using a Dectris Pilatus 16 M PAD detector. Beamline 12-2 was set at 0.97946 Å and the crystal data were collected at 100 K. The dataset was processed using the X-ray Detector Software package[57] that is incorporated into the automated ICEflow (autoPROC) processing pipeline used at SSRL. POINTLESS and AIMLESS were used for scaling and space group assignment, and TRUNCATE for conversion to structure factors. X-ray diffraction data for ApCpp-bound *Kp*OptS were collected at beamline 5.0.1 at the Advanced Light Source using a Dectris Pilatus 32 M detector. The beamline was set at 0.97741 Å wavelength and the crystal data were collected at 100 K. The structures were determined using molecular replacement conducted by the Phaser-MR program in the PHENIX suite (v.1.21-5207)[58] using a predicted structural model of *Kp*OptS generated by ColabFold v.1.5.5, which uses a homology search by MMseqs2 with AlphaFold2[59]. The structures of *Kp*OptS were iteratively refined using the phenix.refine program and the residue positions were manually adjusted in Coot (v.1.1.17)[60]. The ApCpp ligand was a predefined ligand searchable in the PDB under ligand identifier APC, whereas the pCpp fragment ligand restraints were generated using the Elbow tool in Phenix based on the SMILES code: O = P(O)(O)CP(=O)(O)OP(=O)(O)O. The final refined structures had the following Ramachandran statistics—apoprotein: 98.09% favoured, 1.91% allowed, 0.00% outlier; ApCpp-bound: 96.98% favoured, 3.02% allowed, 0% outlier. The full data collection and refinement statistics are summarized in Extended Data Table 1. The models were visualized and analysed using PyMOL (The PyMOL Molecular Graphics System, v.3.1.6.1, Schrödinger). The atomic coordinates for apo *Kp*OptS and *Kp*OptS bound to ApCpp have been deposited to the PDB using IDs 9MNR and 9PD0, respectively. Buried surface area analysis was conducted on the apo *Kp*OptS structure by uploading the coordinate file to the PDBePISA web-based program provided by the PDB in Europe[61].

### Mass photometry analysis of *Kp*OptS

Coverslips (24 mm × 50 mm, Thorlabs) and silicon gaskets (Grace Bio-Labs) were cleaned with several rinses of ultrapure water and HPLC-grade isopropanol and then dried with a clean nitrogen stream. A clean gasket was adhered to a coverslip, and the assembly was placed on the stage of a Refyn Two^MP mass photometer (Refeyn) and centred on a single well. A total of 15 µl of *Kp*OptS diluted to 50 nM in freshly prepared sample buffer (20 mM of HEPES (pH 7.5), 250 mM of KCl) was added to the well and a 60-s video was recorded at room temperature after auto-focusing using the Refeyn AcquireMP (v.2.3.0) software. Each measurement was done in triplicate on two different days. A total of 50 nM of β-amylase in sample buffer was used as a calibration standard (56 kDa, 112 kDa and 224 kDa). Data were processed using DiscoverMP (v.2.3.0; Refeyn).

### HPLC analysis of enzyme activity

Enzymatic reactions were performed in total volumes of 100–600 µl with, typically, 250 µM in total of equimolar ribonucleotide triphosphates (ATP, GTP, CTP, UTP), 1 mM of $MnCl_2$, 10 mM of $MgSO_4$, 20 mM of HEPES-KOH (pH 7.4) and 100 mM of NaCl, with 100 µM of OptS protein added last. Reactions were transferred to a 10-kDa cut-off filter and subjected to centrifugation for 10 min at 13,500g. When appropriate, purified Acb2 was added at 250 µM and incubated for a further 1 h at 37 °C. A total of 1 µl of about 20 mg ml⁻¹ of proteinase K (NEB, catalogue no. P8107S) was added to degrade Acb2 and release cyclic dinucleotide before filtering. Likewise, samples were at times treated with P1 nuclease and/or CIP as indicated. All samples were injected at 10 µl onto an Agilent 1200 Series HPLC equipped with a 4.6 × 150 mm and 5 µm particle-size Zorbax Bonus-RP C18 column using an isocratic elution method including 97% 50 mM $NaH_2PO_4$ (pH 6.8) and 3% acetonitrile buffer system held at 40 °C. Nucleotide separation was monitored at 254 nm using a multiwavelength detector. For mass spectrometry analysis, enzymatic reactions were prepared using 20 mM of ammonium acetate (pH 8.0) instead of HEPES-KOH and NaCl, but otherwise treated the same with or without an liquid chromatography fractionation step to isolate specific peaks. For fractionation on HPLC, the temperature was kept at 23 °C with a gradient elution method as follows: solvent A—20 mM of ammonium acetate (pH 8.0) and solvent B—methanol; 0–2 min, 100% A; 2–8 min, 0–100% B; 8–10 min, 100% B; 10–11 min, 0–100% A; 11–17 min, 100% A. Roughly 1-ml fractions were collected and concentrated through SpeedVac (1–3 h, 30 °C) before injection on a Waters Acquity ultra-performance liquid chromatography (UPLC)-QDa instrument equipped with a C18 50-mm UPLC column and single quadrupole mass spectrometry detector (QDa; *m/z* 50–1250) and photodiode array detector (200–500 nm). Standard gradient elution was used from 0–100% 0.1% formic acid:acetonitrile over 5 min. In ESI+ mode, analyte *m/z* ions (M + H)+ or (M + $NH_3$)+ were validated to less than ±5 ppm relative to the nearest sodiated polyethylene glycol (CAS: 25322-68-3, av. Mwt 400) or sodiated methoxypolyethyleneglycol (CAS: 990-74-4, av. Mwt 350) calibrant peak lockmass.

### Thermal shift analysis of effector nucleotide binding

The soluble β-barrel domain of the S-2TMβ *Vn*OptE effector (residues S73-END) was briefly incubated at a final concentration of 20 µM in reaction buffer containing 3× SYPRO Orange Dye, 100 mM of NaCl, 20 mM of HEPES-KOH (pH 7.5) and 500 µM of the respective cyclic nucleotide as indicated. Sample volumes of 25 µl each were heated from 20 °C to 95 °C over the course of about 1.2 h using a Bio-Rad CFX Duet Real-Time PCR System. The Förster resonance energy transfer channel with excitation at 450–490 nm and detection at 560–580 nm was used to monitor SYPRO fluorescence with every 0.5-°C increment. The first derivative of each fluorescence curve was calculated, and the melting temperature was identified as the peak of each derivative

curve. Data are presented as an average of nine technical replicates and representative of at least three biological replicates.

## Computational analysis of mCpol sequences and structures

PSI-BLAST (Research Resource Identification (RRID):SCR_001010)[62] and jackhmmer (RRID:SCR_005305)[63] were used to carry out iterative sequence profile searches against the non-redundant protein database (nr) from the National Center for Biotechnology Information (NCBI) clustered down to 90% identity (nr90) to eliminate redundancy. These searches were used to collect the complete complement of sequences belonging to the mCpol family. False-positive hits to Cas10 were eliminated using reverse profile searches with the RPS-BLAST program[64]. Clustering on the basis of percentage or bit-score similarity was performed using MMseqs (RRID: SCR_008184)[65], with parameters changed according to the clustering objective. Domains of mCpol-containing proteins and the proteins translated from surrounding genes on the genome with known homology were annotated using a database of domain sequence profiles including Pfam A models (RRID: SCR_004726)[66]. Profile–profile comparisons were performed with the HHpred program[67]. Structural models were constructed using the AlphaFold3 program[68]. Structure similarity searches were performed using the DALI[69] and FoldSeek programs[70].

## Phylogenetic distribution and significance analysis

Taxonomic lineages were obtained from the NCBI Taxonomy Database (RRID: SCR_003256). In total, 13,710 completely sequenced prokaryotic genomes encompassing the organisms from these lineages were downloaded from NCBI GenBank and checked for the presence or absence of representative conflict system components using the PSI-BLAST iterative search program. This was followed by clustering for homologue groups and reverse searches as described above. NCBI genome record (GCA) numbers were used as markers for unique organisms. The significance of pairwise system overlap within the sampled genome space was calculated using Fisher's exact test implemented in the R command fisher.test(). Data processing (knitr and dplyr libraries), statistical analysis (stats and Rmpfr libraries) and visualization (VennDiagram library) were performed using the R language.

## Synthetic nucleotide ligands

Synthetic cyclic dinucleotide ligands used for HPLC, thermofluor and mass spectrometry analysis were purchased from Biolog Life Science Institute: 3',3'-c-di-AMP (catalogue no. C 088), 2',3'-c-di-AMP (catalogue no. C 187), 2',2'-c-di-AMP (catalogue no. C 188), 3',3'-c-di-GMP (catalogue no. C 057), 2',3'-c-di-GMP (catalogue no. C 182), 2',2'-c-di-GMP (catalogue no. C 162), 3',3'-cGAMP (catalogue no. C 117), 2',3'-cGAMP (catalogue no. C 161), 3',2'-cGAMP (catalogue no. C 238), 2',2'-cGAMP (catalogue no. C 210), pppA(2',5')pA (catalogue no. T 073), 3',3'-cUAMP (catalogue no. C 357), 3',2'-cUAMP (catalogue no. C 398), 2',3'-cUAMP (catalogue no. C 399), c-tri-AMP (catalogue no. C 362), c-tetra-AMP (catalogue no. C 335), c-penta-AMP (catalogue no. C 394) and c-hexa-AMP (catalogue no. 332). ApCpp was purchased from Jena Bioscience (catalogue no. NU-421).

## Accession numbers

The crystal structure data for apo *Kp*OptS and *Kp*OptS bound to ApCpp have been deposited in the PDB (9MNR and 9PD0, respectively). All other relevant accession numbers can be found in Supplementary Table 4.

## Statistics and reproducibility

Each experiment presented was performed with independent biological replicates using bacterial cultures grown on separate days. The *n*, error bars and bar graph data are defined in the figure legends. Error bars were selected according to published recommendations[71–73].

Data were plotted using GraphPad Prism 9. Statistical analysis of specified comparisons was performed, and all results can be found in the corresponding source data. In all cases, a two-sided, unpaired, parametric *t*-test was used. To compare multi-log differences, data were log-transformed before statistical analysis. When four or fewer comparisons are made in a single graph, statistical comparisons are shown in the figure. In all other cases, statistics are presented in the source data to aid clarity. *$P < 0.05$, **$P < 0.001$. Illumina sequencing results were analysed using Geneious Prime Software (v.2024.0.1).

## Reporting summary

Further information on research design is available in the Nature Portfolio Reporting Summary linked to this article.

## Data availability

The crystal structure data for apo *Kp*OptS and *Kp*OptS bound to ApCpp have been deposited in the PDB (9MNR and 9PD0, respectively). All other data supporting the findings of this study are available in the paper and Supplementary Information. For gel source data, see Supplementary Fig. 1. Source data are provided with this paper.

## Code availability

All code and software used in this study are described in the paper and Supplementary Information. No custom code was developed for the purposes of any analyses in the paper.

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

**Acknowledgements** We thank the Shared Instruments Pool (RRID: SCR_018986) of the Department of Biochemistry at the University of Colorado Boulder for providing access to the Avanti JXN-26 Super Speed centrifuges and rotors, which are funded by National Institutes of Health (NIH) Grant R24OD033699-01, as well as the ITC 200, funded by the NIH Shared Instrumentation Grant S10RR026516, and members of the Whiteley and Morehouse laboratories for their advice and helpful discussion. Use of the Nikon A1R microscope in the BioFrontiers Institute's Advanced Light Microscopy Core (RRID: SCR_018302) was supported by NIST-CU Cooperative Agreement award number 70NANB15H226. We thank the UCI Mass Spectrometry Facility for assistance with small molecule analyses and accurate mass measurements. Use of the SSRL, SLAC National Accelerator Laboratory, is supported by the US Department of Energy, Office of Science, Office of Basic Energy Sciences under Contract No. DE-AC02-76SF00515. The SSRL Structural Molecular Biology Program is supported by the DOE Office of Biological and Environmental Research, and by the NIH National Institute of General Medical Sciences (P30GM133894). Beamline 5.0.1 of the Advanced Light Source, a DOE Office of Science User Facility under contract no. DE-AC02-05CH11231, is supported in part by the ALS-ENABLE programme funded by the NIH National Institute of General Medical Sciences, grant P30 GM124169-01. This work was funded by the National Institute of General Medical Sciences of the NIH under award number R35GM157311 (B.R.M.); the NIH through the NIH Director's New Innovator Award DP2AT012346 (A.T.W.); a PEW Charitable Trust Biomedical Scholars Award (A.T.W.); the Boettcher Foundation Webb-Waring Biomedical Research Award (A.T.W.); a Burroughs Wellcome Fund PATH Award 1186087 (A.T.W.); a University of Colorado ABNexus Grant (joint with A.T.W and K. Doran); and a Mallinckrodt Foundation Grant (A.T.W.). A.E.S. was supported in part by the NIH Interdisciplinary Predoctoral Training in Molecular Biophysics grant (T32GM145437) and a Ruth L. Kirschstein National Research Service Award Individual Predoctoral Fellowship (F31AI186492). K.S. was supported in part by the Undergraduate Research Opportunities Program Individual Grant funded by CU Boulder. C.R.K.H. was supported in part by the Boettcher Foundation Collaboration Grant and the Undergraduate Research Opportunities Program Individual Grant funded by CU Boulder. L.K.R. is supported in part by the Interdisciplinary Quantitative Biology (IQ Biology) PhD programme at the BioFrontiers Institute, University of Colorado Boulder, the National Science Foundation NRT Integrated Data Science Fellowship (2022138) and by the NIH Interdisciplinary Predoctoral Training in Molecular Biophysics grant (T32GM145437). H.E.L. was supported in part as a fellow of the Jane Coffin Childs Memorial Fund for Medical Research. U.T. was supported as a fellow of the Cancer Research Institute Irvington Postdoctoral Fellowship (CRI4043). E.M.K. was funded in part by the NIH T32 Signaling and Cellular Regulation training grant (T32GM008759 and T32GM142607). A.N. was supported by a Graduate Assistance in Areas of National Need (GAANN) Fellowship (#P200A240034) awarded to the University of California Irvine Department of Molecular Biology and Biochemistry. D.S.I. was supported in part by funding through a University of California Irvine Undergraduate Research Opportunities Program Fellowship. A.M.B. and L.A. are supported by the Intramural Research Program (ZIA LM594244) of the NIH. The contributions of the NIH authors were made as part of their official duties as NIH federal employees, are in compliance with agency policy requirements and are considered works of the US government.

**Author contributions** A.E.S., A.T.W. and B.R.M. conceptualized the study. A.E.S., A.T.W. and B.R.M. developed the methodology. Plate-based phage infection assays were performed by A.E.S., C.R.K.H. and H.E.L. Liquid culture phage infections were performed by A.E.S. and C.R.K.H. Escaper phage generation and analysis was performed by A.E.S. Construction of phage gene deletions was performed by E.M.K. Colony formation assays were performed by A.E.S. Protein expression and purification was performed by A.E.S., A.N., B.R.M., C.M.N., D.S.I., E.M.Q.E., K.S., M.L.D. and U.T. ITC was performed by M.L.D. Measurement of intracellular nucleotides was performed by K.S., T.A.N and A.E.S. HPLC was performed by D.S.I. and E.M.Q.E. Mass spectrometry was performed by E.M.Q.E. Thermofluor analysis was conducted by A.N. Protein crystallization was performed by C.M.N. and D.M.D. and data collection was carried out by D.M.D. Laser scanning confocal microscopy was performed by L.K.R. Mass photometry was performed by A.H.E. and A.E.S. Analysis of mCpol sequences, their phylogenetic distribution and gene linkage was performed by L.A. and A.M.B. A.E.S., A.T.W. and B.R.M. wrote the original draft. A.E.S., T.A.N, A.T.W. and B.R.M. reviewed and edited the manuscript. A.E.S., D.S.I. and B.R.M. were responsible for visualization. A.T.W. and B.R.M. supervised the project. A.T.W. and B.R.M. acquired funding.

**Competing interests** The University of Colorado Boulder and the University of California, Irvine, have patents pending for Panoptes and related technologies on which A.E.S., A.T.W. and B.R.M. are listed as inventors (provisional patent 63/772,257 filed 14 March 2025). The other authors declare no competing interests.

**Additional information**
**Correspondence and requests for materials** should be addressed to Aaron T. Whiteley or Benjamin R. Morehouse.

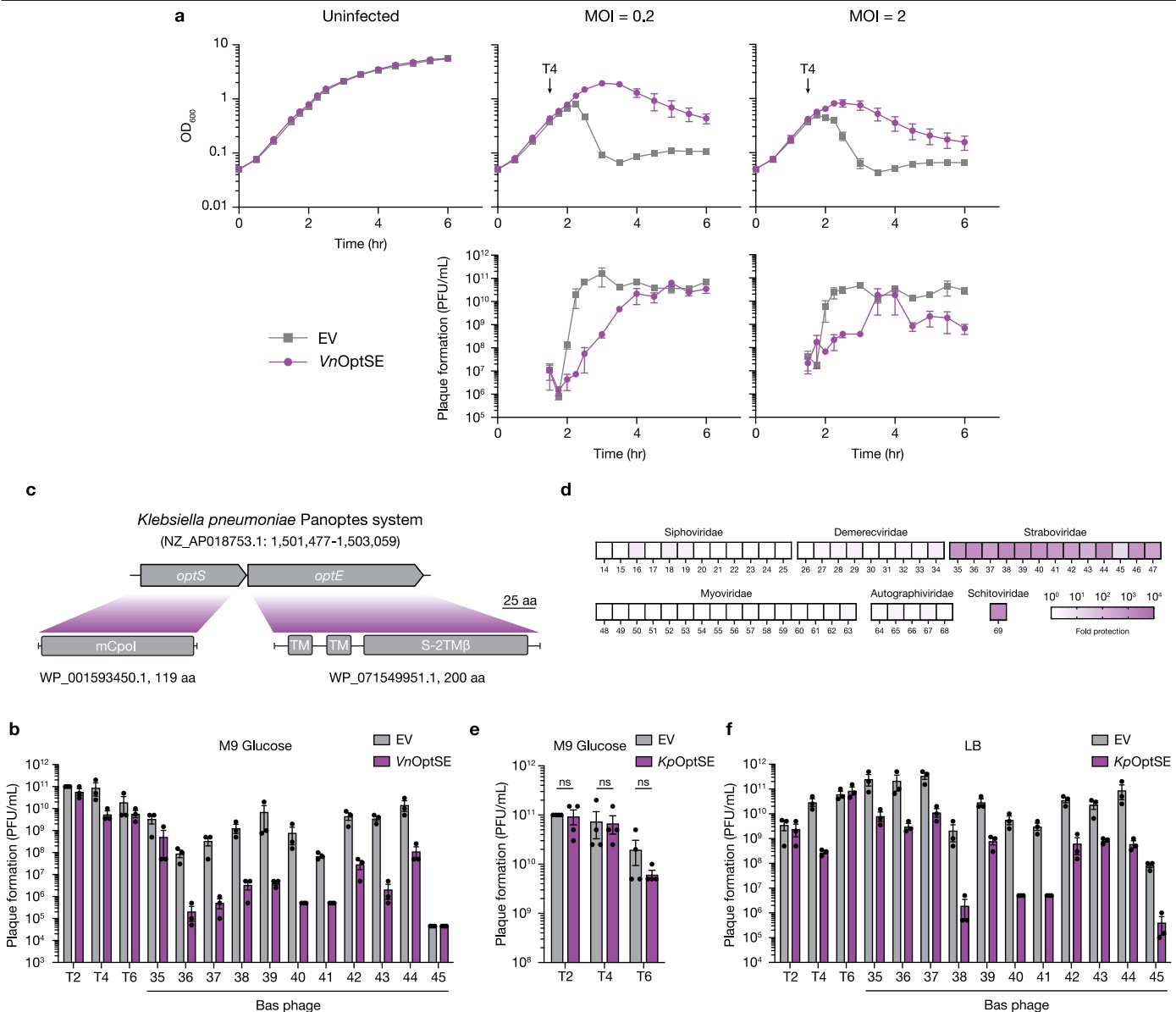

**Extended Data Fig. 1 | Panoptes protects against phages from the Straboviridae family. (a)** Above: growth curve of *E. coli* expressing the indicated plasmid. Arrows indicate the time each culture was infected with phage T4 at the indicated multiplicity of infection (MOI). Below: efficiency of plating of the phage present in each sample at the indicated time points. Data represent the mean ± SEM of n = 3 biological replicates. **(b)** Efficiency of plating of indicated phages infecting *E. coli* expressing a plasmid with either *Vn*OptSE or an empty vector (EV). Data represent the mean ± SEM of n = 3 biological replicates, shown as individual points. **(c)** Domain architectures of *Klebsiella pneumoniae* Panoptes operon genes, *optS* and *optE*. **(d)** Heatmap of fold defense provided by *Kp*OptSE for a panel of diverse phages from the BASEL

collection. *E. coli* expressing *Kp*OptSE were challenged with phages and fold defense was calculated for each phage by dividing the efficiency of plating (in PFU/mL) on EV by the efficiency of plating on *Kp*OptSE-expressing bacteria. Family names are above each indicated set of phages. **(e)** Efficiency of plating of indicated phages infecting *E. coli* expressing a plasmid with either *Kp*OptSE or an EV. Data represent the mean ± SEM of n = 4 biological replicates, shown as individual points. A two-sided Student's *t*-test was used. ns = not significant. **(f)** Efficiency of plating of indicated phages infecting *E. coli* expressing a plasmid with either *Kp*OptSE or an EV. Data represent the mean ± SEM of n = 3 biological replicates, shown as individual points.

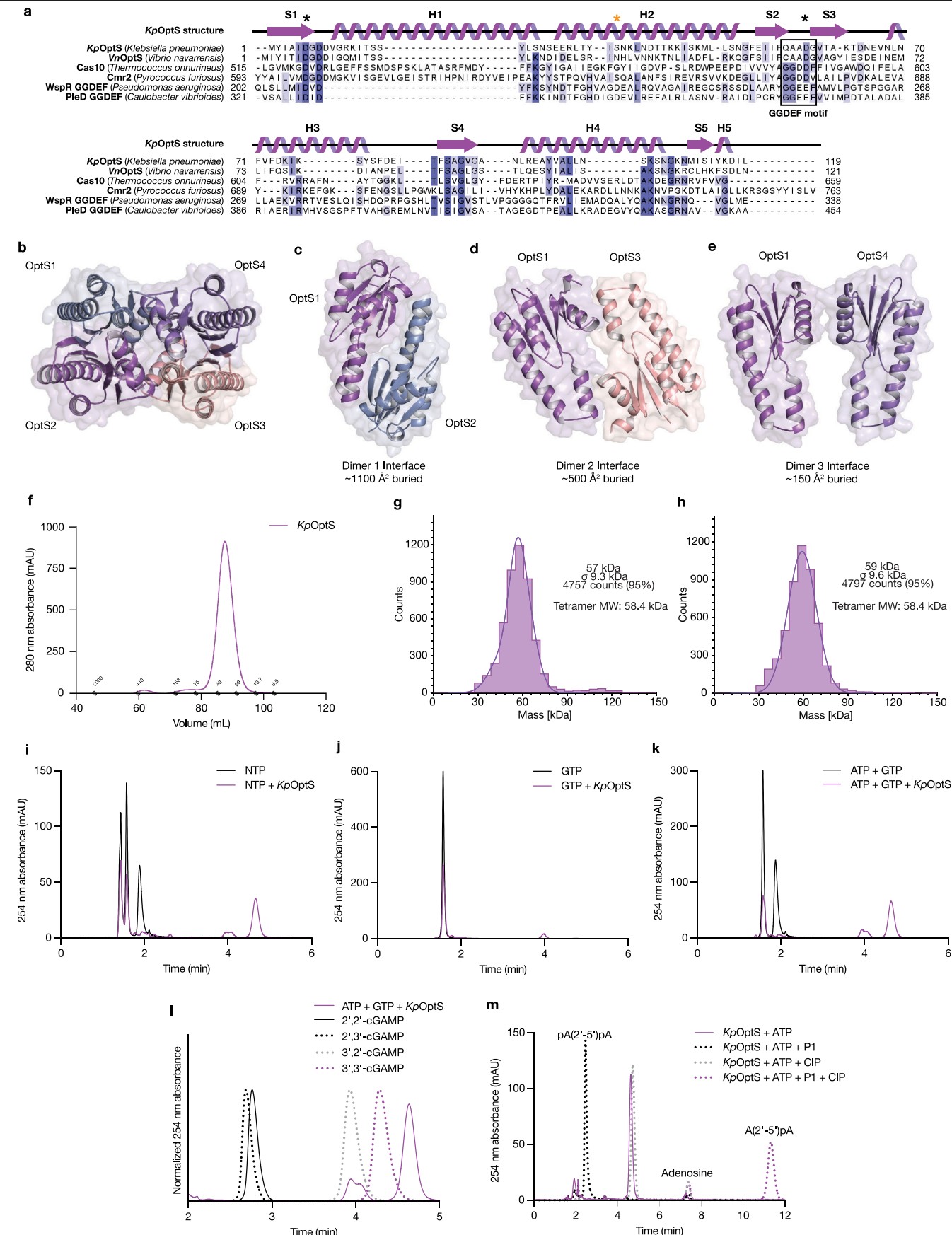

**Extended Data Fig. 2** | See next page for caption.

**Extended Data Fig. 2 | Oligomeric state analysis and biochemical profiling of *Kp*OptS. (a)** Multiple sequence alignment of representative Panoptes OptS, type-III CRISPR polymerase palm domains, and GGDEF palm domains. Residues important for catalysis are indicated with black asterisks. Orange asterisk indicated nucleobase recognition residues for *Kp* and *Vn*OptS. **(b–e)** The *Kp*OptS tetramer contains three possible dimer (pair of protomers) interfaces shown in order of decreasing buried surface area (determined with PISA analysis, see Supplementary Table 1). **(f)** Size exclusion chromatography reveals a monodisperse signal for *Kp*OptS with an elution volume consistent with an oligomer >29 kDa (monomer = 14 kDa). Grey dots indicate molecular weight standards: Blue dextran, 46.2 mL, 2000 kDa; Ferritin, 59.8 mL, 440 kDa; Aldolase, 71.0 mL, 158 kDa; Conalbumin, 78.9 mL, 75 kDa; Ovalbumin, 85.3 mL, 43 kDa; Carbonic anhydrase, 91.5 mL, 29 kDa; Ribonuclease A, 98.6 mL, 13.7 kDa; Aprotinin, 104.4 mL, 6.5 kDa. The trace is representative of at least two biological replicates. **(g, h)** Mass photometry results showing distribution of sizes detected for *Kp*OptS in solution at 50 nM. **(i)** HPLC analysis of mixture of 4×NTPs (ATP, GTP, UTP, CTP) compared with the product of *Kp*OptS when incubated with NTPs. Traces representative of at least three biological replicates. **(j)** HPLC analysis of GTP compared with the product of *Kp*OptS when incubated with GTP. Traces representative of at least three biological replicates. **(k)** HPLC analysis of ATP and GTP compared with the product of *Kp*OptS when incubated with ATP and GTP. Traces representative of at least three biological replicates. **(l)** HPLC analysis of 2′,2′-cGAMP, 2′,3′-cGAMP, 3′,2′-cGAMP, and 3′,3′-cGAMP chemical standards compared with the product of *Kp*OptS incubated with ATP and GTP. Traces representative of at least three biological replicates. **(m)** P1 nuclease and CIP treatment of *Kp*OptS ATP-only product. Traces representative of at least three biological replicates.

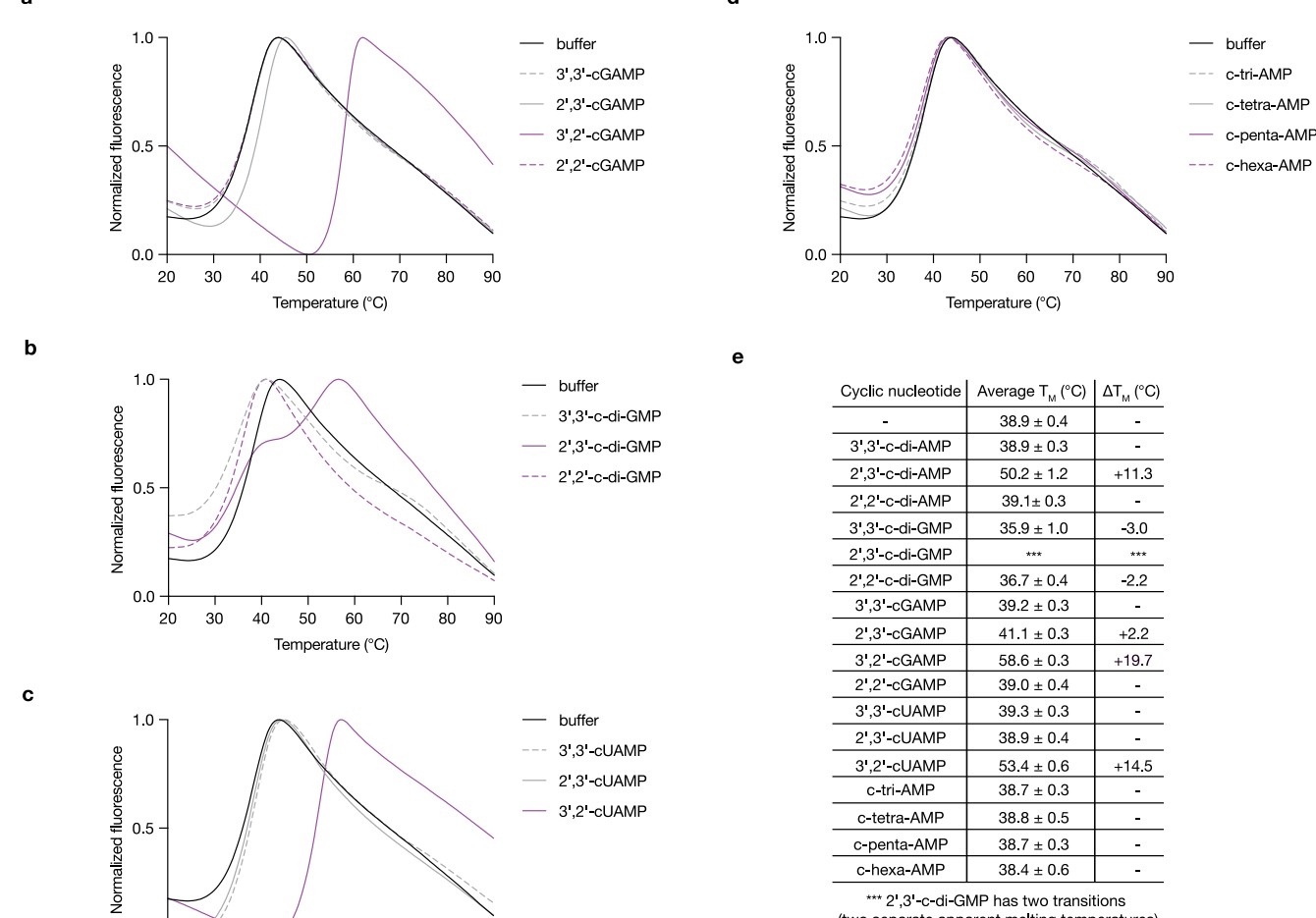

| Cyclic nucleotide | Average $T_M$ (°C) | $\Delta T_M$ (°C) |
|---|---|---|
| - | 38.9 ± 0.4 | - |
| 3',3'-c-di-AMP | 38.9 ± 0.3 | - |
| 2',3'-c-di-AMP | 50.2 ± 1.2 | +11.3 |
| 2',2'-c-di-AMP | 39.1± 0.3 | - |
| 3',3'-c-di-GMP | 35.9 ± 1.0 | -3.0 |
| 2',3'-c-di-GMP | *** | *** |
| 2',2'-c-di-GMP | 36.7 ± 0.4 | -2.2 |
| 3',3'-cGAMP | 39.2 ± 0.3 | - |
| 2',3'-cGAMP | 41.1 ± 0.3 | +2.2 |
| 3',2'-cGAMP | 58.6 ± 0.3 | +19.7 |
| 2',2'-cGAMP | 39.0 ± 0.4 | - |
| 3',3'-cUAMP | 39.3 ± 0.3 | - |
| 2',3'-cUAMP | 38.9 ± 0.4 | - |
| 3',2'-cUAMP | 53.4 ± 0.6 | +14.5 |
| c-tri-AMP | 38.7 ± 0.3 | - |
| c-tetra-AMP | 38.8 ± 0.5 | - |
| c-penta-AMP | 38.7 ± 0.3 | - |
| c-hexa-AMP | 38.4 ± 0.6 | - |

*** 2',3'-c-di-GMP has two transitions
(two separate apparent melting temperatures)
$T_{M1}$ = 34.9 °C ± 0.9 $T_{M2}$ = 52.4 °C ± 0.5
$\Delta T_{M1}$= -4.0 °C & $\Delta T_{M2}$= 13.5 °C

**Extended Data Fig. 3 | Thermal stability assays for *Vn*OptE. (a)** Thermal shift data showing the melt profile for a soluble version of *Vn*OptE incubated with several cGAMP linkage isomers. Data representative of nine independent reactions per tested condition (3 replicate plates, 3 technical replicates per plate). **(b)** Thermal shift data showing the melt profile for a soluble version of *Vn*OptE incubated with several c-di-GMP linkage isomers. Data representative of nine independent reactions per tested condition (3 replicate plates, 3 technical replicates per plate). **(c)** Thermal shift data showing the melt profile for a soluble version of *Vn*OptE incubated with several cUAMP linkage isomers.

Data representative of nine independent reactions per tested condition (3 replicate plates, 3 technical replicates per plate). **(d)** Thermal shift data showing the melt profile for a soluble version of *Vn*OptE incubated with several cyclic nucleotides which have been shown to be important in type III CRISPR signaling. **(e)** Table of melting temperatures and thermal shift values for all tested cyclic nucleotides with *Vn*OptE. Data representative of nine independent reactions per tested condition (3 replicate plates, 3 technical replicates per plate). Error in values is reported as standard deviation.

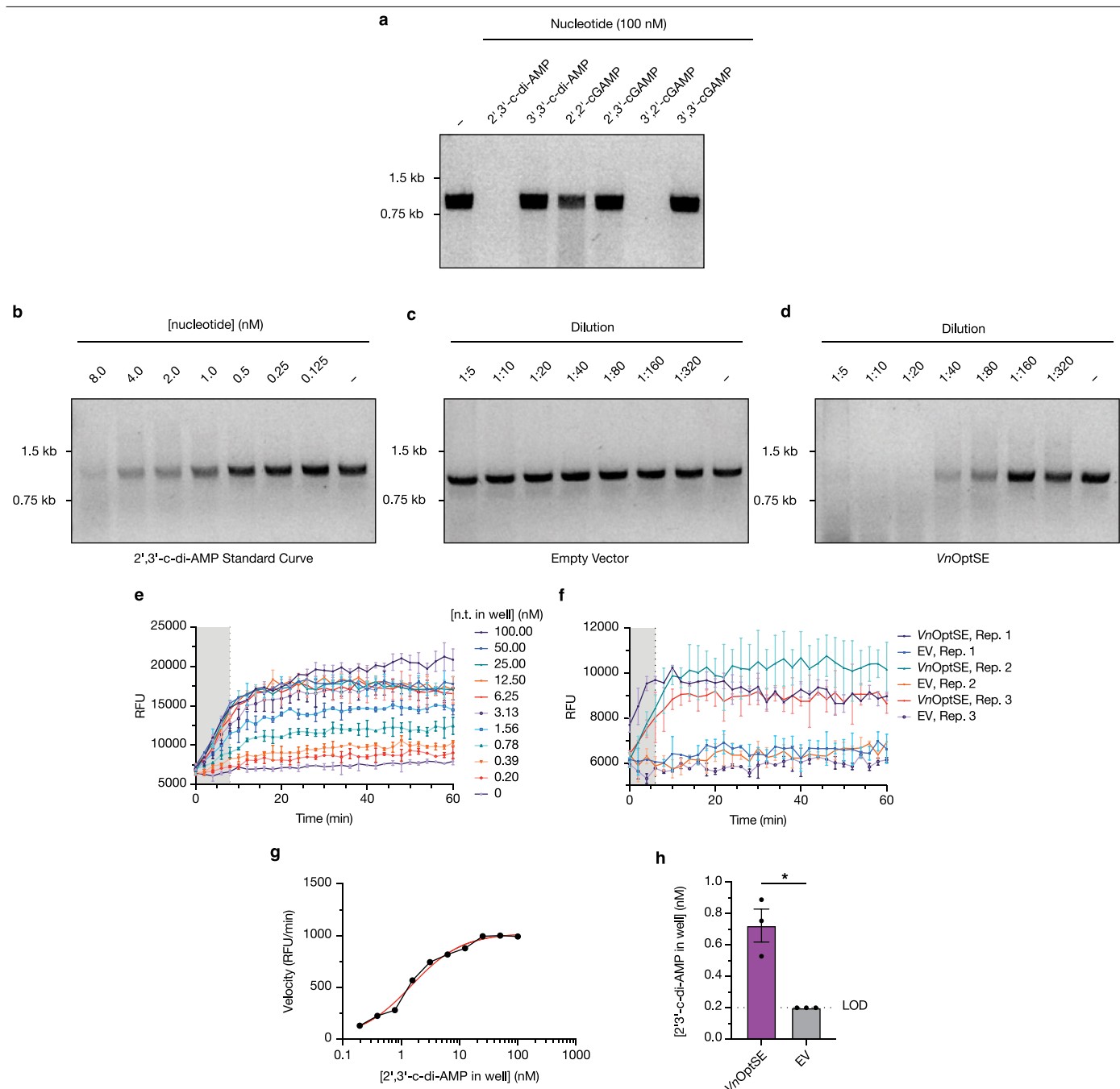

**Extended Data Fig. 4 | *Ab*Cap5 is a sensor of 2′,3′-c-di-AMP. (a)** Visualization of DNA degradation by *Ab*Cap5 when incubated with either 2′,3′-c-di-AMP, 3′,3′-c-di-AMP, 2′,2′-cGAMP, 2′,3′-cGAMP, 3′,2′-cGAMP, or 3′,3′-cGAMP. Data are representative images of n = 2 technical replicates. **(b–d)** Visualization of DNA degradation by *Ab*Cap5 when incubated with a dilution series of 2′,3′-c-di-AMP (**b**) or extracted nucleotides from *E. coli* expressing an empty vector (EV; **c**) or the *Vn*OptSE operon (**d**). Data are representative images of n = 3 biological replicates. **(e)** Fluorescence-based DNase activity assay of *Ab*Cap5 incubated with the indicated concentration of 2′,3′-c-di-AMP. Cleavage of the DNA substrate results in fluorescence and was measured as relative fluorescence units (RFU), which was monitored over time. The gray shaded area represents the linear range of the assay. Data are the mean ± SEM of n = 3 technical replicates and are representative of n = 2 biological replicates. **(f)** Fluorescence-based DNase

activity assay of *Ab*Cap5 incubated with nucleotide extracts from *E. coli* expressing either an EV or *Vn*OptSE. Cleavage of the DNA substrate results in fluorescence and was measured as relative fluorescence units (RFU), which was monitored over time. The gray shaded area represents the linear range of the assay. Data are the mean ± SEM of n = 3 technical replicates and are representative of n = 2 biological replicates. **(g)** Standard curve of velocity (RFU/min) versus concentration of 2′,3′-c-di-AMP in the reaction well. Velocities were determined for each 2′,3′-c-di-AMP concentration from the points within the linear range shown in (**e**). **(h)** Quantification of 2′,3′-c-di-AMP from *Vn*OptSE or EV-expressing *E. coli* in the reaction well. Data represents the mean ± SEM of n = 3 biological replicates, shown as individual points. A two-sided Student's *t*-test was used. *$P < 0.05$.

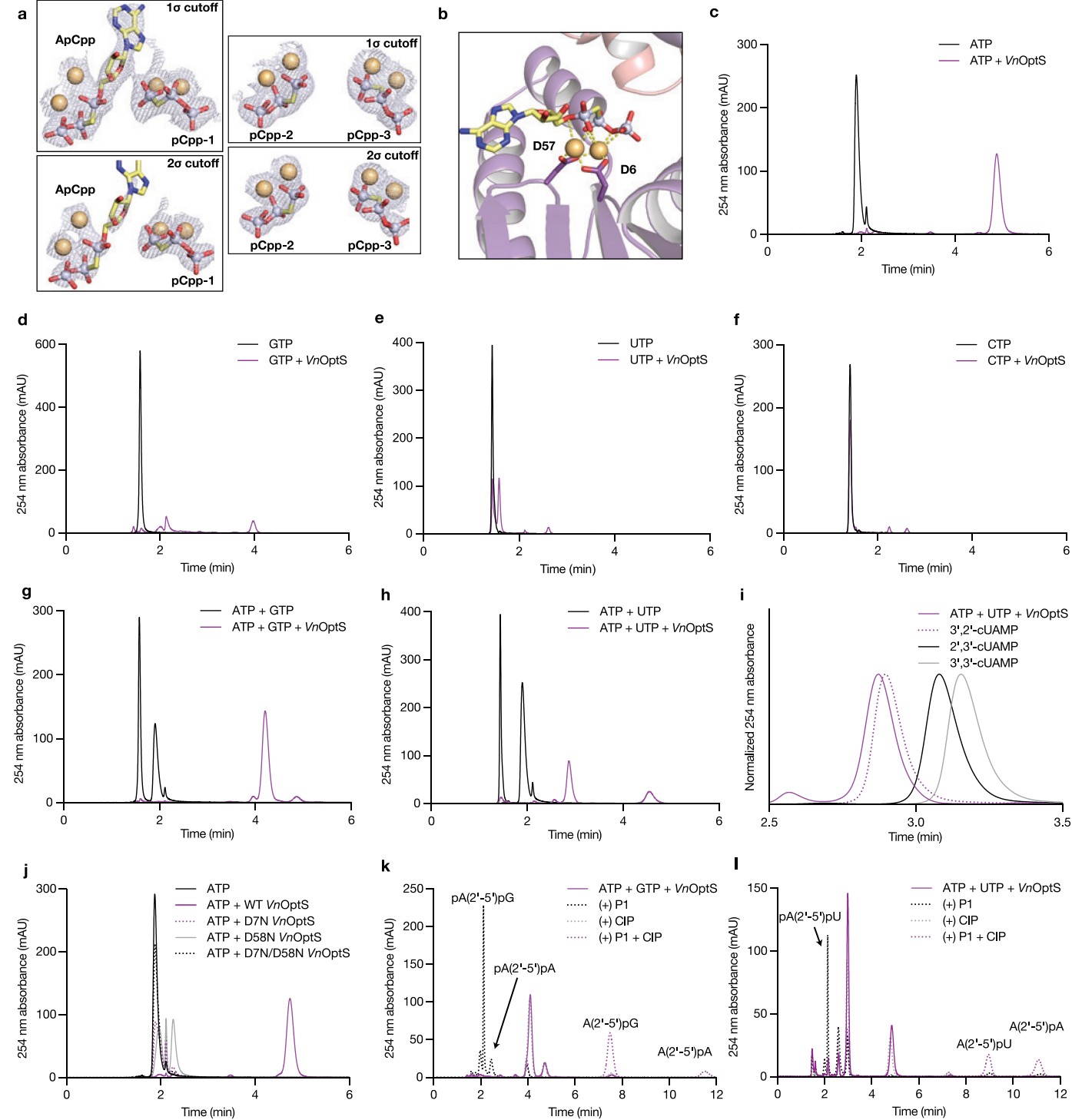

**Extended Data Fig. 5 | HPLC analysis of *Vn*OptS catalytic activity. (a)** Electron density map for the ligands in the *Kp*OptS structure at two sigma cutoff values (1 above, 2 below) showing the weak density quality for the ribose and adenine base in ApCpp but strong density supporting analog binding through phosphate-metal ion interactions at all four possible binding sites within the tetramer. **(b)** ApCpp interactions with divalent cation (likely Mg$^{2+}$) through catalytic residues Asp6 and Asp57. **(c)** HPLC analysis of the product of *Vn*OptS when incubated with ATP. Traces representative of at least three biological replicates. **(d)** HPLC analysis of the product of *Vn*OptS when incubated with GTP. Traces representative of at least three biological replicates. **(e)** HPLC analysis of the product of *Vn*OptS when incubated with UTP. Traces representative of at least three biological replicates. **(f)** HPLC analysis of the product of *Vn*OptS when incubated with CTP. Traces representative of at least three biological replicates.

**(g)** HPLC analysis of the product of *Vn*OptS when incubated with ATP and GTP. Traces representative of at least three biological replicates. **(h)** HPLC analysis of the product of *Vn*OptS when incubated with ATP and UTP. Traces representative of at least three biological replicates. **(i)** HPLC analysis of 3′,3′-cUAMP, 2′,3′-cUAMP, and 3′,2′-cUAMP chemical standards compared with the product of *Vn*OptS incubated with ATP and UTP. Traces representative of at least three biological replicates. **(j)** HPLC analysis of the reaction products when wild-type or mutant *Vn*OptS is incubated with ATP. Traces representative of at least three biological replicates. **(k)** HPLC analysis of P1 nuclease and CIP treatment of *Vn*OptS product of incubation with ATP and GTP (3′,2′-cGAMP). Traces representative of at least three biological replicates. **(l)** HPLC analysis of P1 nuclease and CIP treatment of *Vn*OptS product of incubation with ATP and UTP (3′,2′-cUAMP). Traces representative of at least three biological replicates.

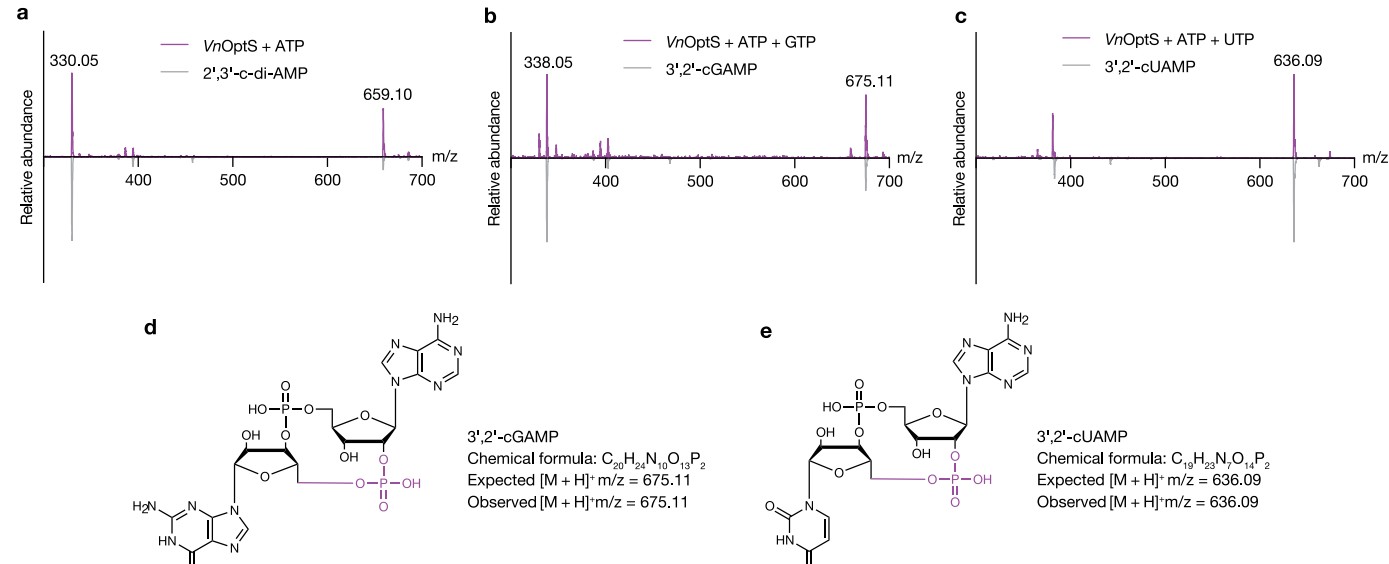

**a**
— VnOptS + ATP
— 2',3'-c-di-AMP

330.05
659.10

m/z

Relative abundance

**b**
— VnOptS + ATP + GTP
— 3',2'-cGAMP

338.05
675.11

m/z

Relative abundance

**c**
— VnOptS + ATP + UTP
— 3',2'-cUAMP

636.09

m/z

Relative abundance

**d**

3',2'-cGAMP
Chemical formula: $C_{20}H_{24}N_{10}O_{13}P_2$
Expected [M + H]$^+$ m/z = 675.11
Observed [M + H]$^+$ m/z = 675.11

**e**

3',2'-cUAMP
Chemical formula: $C_{19}H_{23}N_7O_{14}P_2$
Expected [M + H]$^+$ m/z = 636.09
Observed [M + H]$^+$ m/z = 636.09

**Extended Data Fig. 6 | Mass spectrometry data for *Vn*OptS cyclic dinucleotides. (a–c)** MS spectra of chemical standard (bottom) and the *Vn*OptS nucleotide-dependent product (top) for 2',3'-c-di-AMP, 3',2'-cGAMP, and 3',2'-cUAMP. **(d–e)** Chemical structure of 3',2'-cyclic guanosine monophosphate-adenosine monophosphate (3',2'-cGAMP) and 3',2'-cyclic uridine monophosphate-adenosine monophosphate (3',2'-cUAMP).

**Extended Data Fig. 7 | Phage T4 escapers evade Panoptes-mediated defense.**
**(a)** Efficiency of plating of T4 parent and escaper phages on *E. coli* expressing an empty vector (EV) or the *Vn*OptSE operon from a plasmid. Images are representative of n = 3 biological replicates. **(b)** Efficiency of plating of T4 parent phages and corresponding escaper mutant phages (Escaper, T4.X.Y, where X indicates the parent phage number and Y indicates the escaper phage number) on *E. coli* expressing an EV or the *Vn*OptSE operon from a plasmid. Data are the mean ± SEM of n = 3 biological replicates, shown as individual points. **(c)** Efficiency of plating of T4 phages with the indicated genotype on *E. coli* expressing an EV or the *Vn*OptSE operon from a plasmid. Data are the mean ± SEM of n = 3 biological replicates, shown as individual points. **(d)** Efficiency

of plating of T4 phages with the indicated genotype on *E. coli* expressing an EV or the *Vn*OptSE operon from a plasmid. Data are the mean ± SEM of n = 3 biological replicates, shown as individual points. **(e)** Efficiency of plating of T4 phages with the indicated genotype on *E. coli* expressing an EV or the *Kp*OptSE operon from a plasmid. Data are the mean ± SEM of n = 3 biological replicates, shown as individual points. **(f)** Efficiency of plating of T4 phages with the indicated genotype on *E. coli* expressing an EV or the CBASS operon from a plasmid. Data are the mean ± SEM of n = 3 biological replicates, shown as individual points. For **c-f**, a two-sided Student's *t*-test was used. *P < 0.05, **P < 0.001.

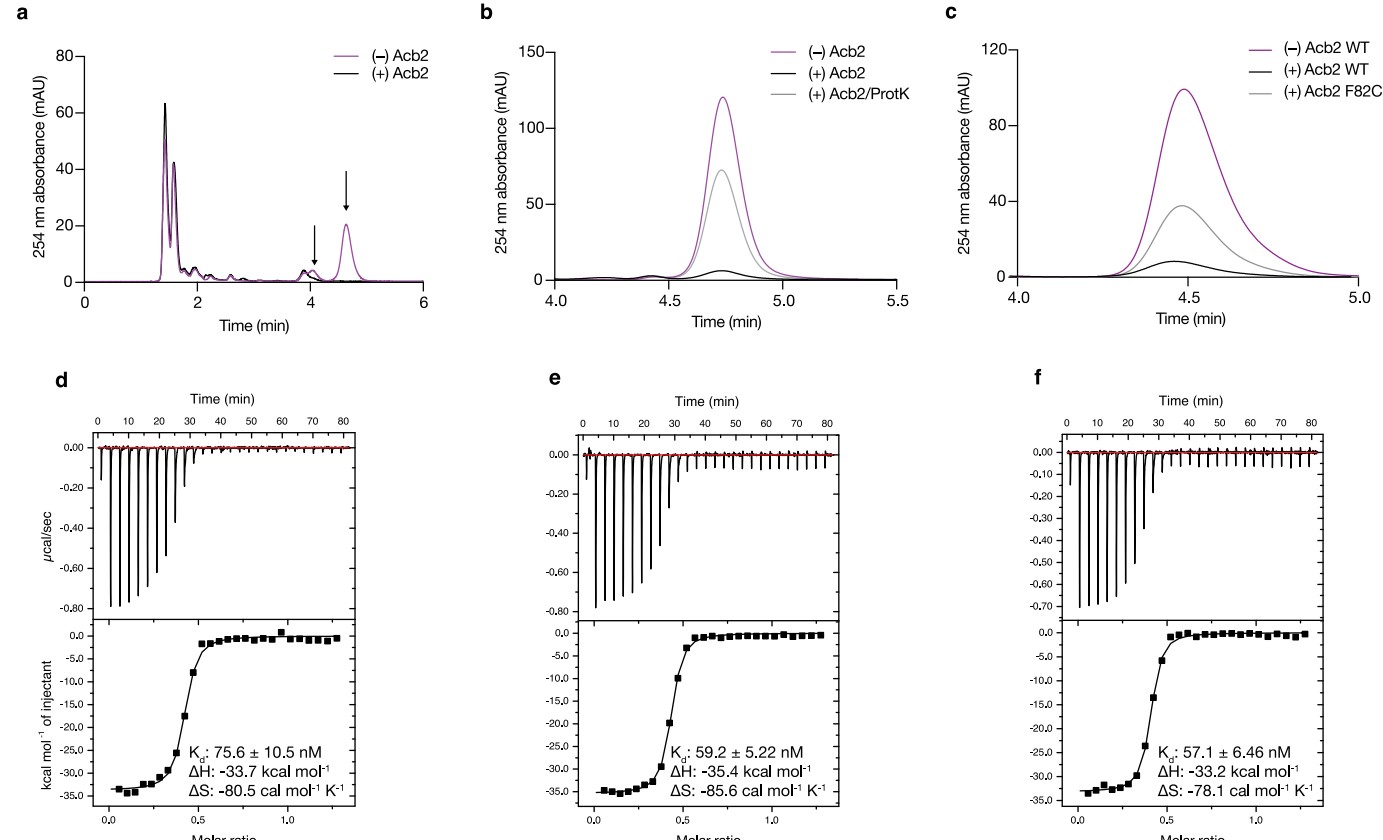

**Extended Data Fig. 8 | Acb2 binds the product of *Kp*OptS. (a)** HPLC analysis of the ability of Acb2 to bind the main products of *Kp*OptS (ATP and GTP reaction). Traces representative of at least three biological replicates. **(b)** HPLC analysis of the ability of Acb2 to bind and release the product of *Kp*OptS when treated with proteinase K (ATP only reaction). Traces representative of at least three biological replicates. **(c)** HPLC analysis of the ability of mutant Acb2 to bind the ATP-derived product of *Kp*OptS. Traces representative of at least three biological replicates. **(d–f)** Replicates of ITC assays to test binding of 2′,3′-c-di-AMP to Acb2.

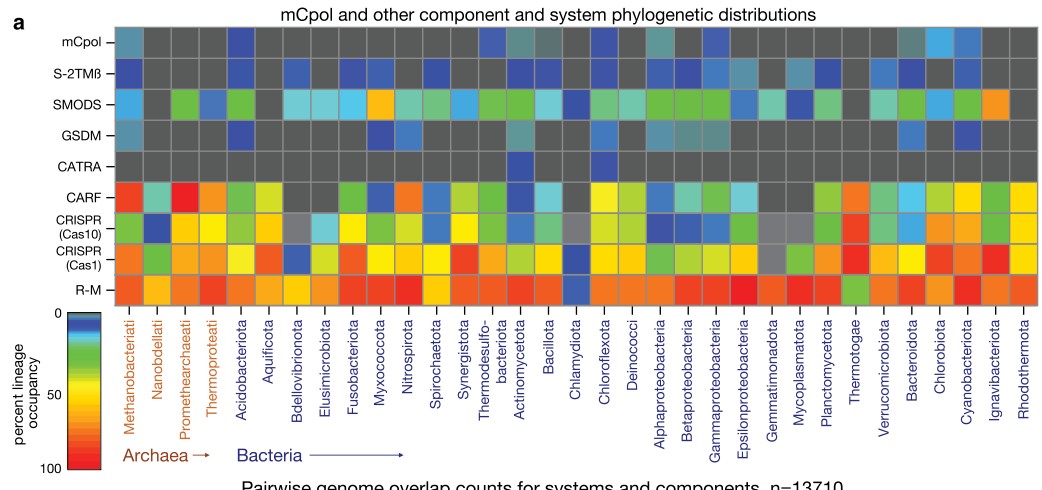

## a. mCpol and other component and system phylogenetic distributions

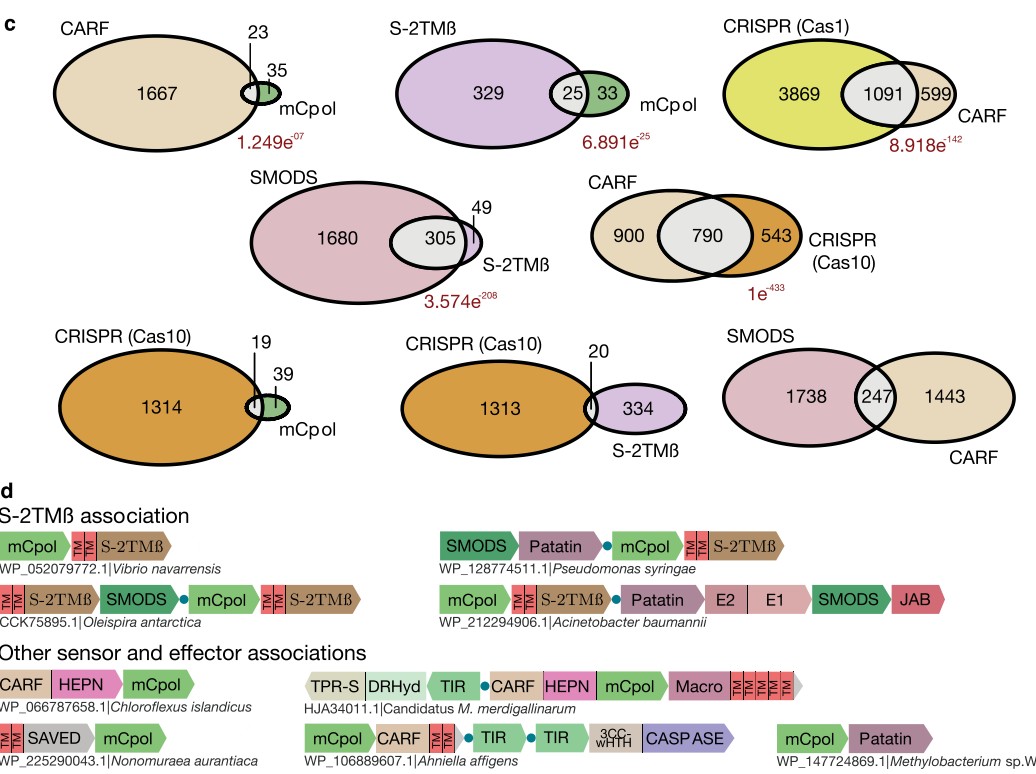

### b. Pairwise genome overlap counts for systems and components, n=13710

|  | R-M | CRISPR (Cas1) | CRISPR (Cas10) | CARF | CATRA | GSDM | SMODS | S-2TMß | mCpol |
|---|---|---|---|---|---|---|---|---|---|
| R-M | 79.30% | 31.12% | 8.21% | 10.07% | 0.34% | 0.53% | 12.76% | 2.23% | 0.39% |
| CRISPR (Cas1) | | 36.18% | 8.66% | 7.96% | 0.08% | 0.18% | 4.75% | 0.77% | 0.22% |
| CRISPR (Cas10) | | | 9.72% | 5.76% | 0.02% | 0.07% | 1.15% | 0.15% | 0.14% |
| CARF | | | | 12.33% | 0.04% | 0.07% | 1.80% | 0.34% | 0.17% |
| CATRA | | | | | 0.36% | 0% | 0.10% | 0% | 0.01% |
| GSDM | | | | | | 0.62% | 0.13% | 0.04% | 0% |
| SMODS | | | | | | | 14.48% | 2.22% | 0.23% |
| S-2TMß | | | | | | | | 2.58% | 0.18% |
| mCpol | | | | | | | | | 0.42% |

### p-value results for pairwise Fisher's exact test

|  | R-M | CRISPR (Cas1) | CRISPR (Cas10) | CARF | CATRA | GSDM | SMODS | S-2TMß | mCpol |
|---|---|---|---|---|---|---|---|---|---|
| R-M | 0 | 1.236e-50 | 5.031e-07 | 0.010 | 7.762e-03 | 0.282 | 2.656e-28 | 5.301e-04 | 8.335e-03 |
| CRISPR (Cas1) | | 0 | 1e-393 | 8.918e-142 | 0.039 | 0.141 | 7.109e-04 | 9.880e-03 | 0.019 |
| CRISPR (Cas10) | | | 0 | 1e-433 | 0.479 | 0.714 | 2.777e-03 | 8.071e-03 | 1.207e-06 |
| CARF | | | | 0 | 0.829 | 0.742 | 0.854 | 0.567 | 1.249e-07 |
| CATRA | | | | | 0 | 1 | 0.013 | 0.641 | 0.191 |
| GSDM | | | | | | 0 | 0.088 | 0.069 | 1 |
| SMODS | | | | | | | 0 | 3.574e-208 | 3.715e-12 |
| S-2TMß | | | | | | | | 0 | 6.891e-25 |
| mCpol | | | | | | | | | 0 |

### c.

CARF 23 / 35 / 1667 / mCpol — 1.249e⁻⁰⁷

S-2TMß 329 / 25 / 33 / mCpol — 6.891e²⁵

CRISPR (Cas1) 3869 / 1091 / 599 / CARF — 8.918e¹⁴²

SMODS 1680 / 305 / 49 / S-2TMß — 3.574e²⁰⁸

CARF 900 / 790 / 543 / CRISPR (Cas10) — 1e⁻⁴³³

CRISPR (Cas10) 1314 / 19 / 39 / mCpol

CRISPR (Cas10) 1313 / 20 / 334 / S-2TMß

SMODS 1738 / 247 / 1443 / CARF

### d.

#### S-2TMß association

mCpol — TM TM — S-2TMß
WP_052079772.1|*Vibrio navarrensis*

SMODS — Patatin — mCpol — TM TM — S-2TMß
WP_128774511.1|*Pseudomonas syringae*

TM TM — S-2TMß — SMODS — mCpol — TM TM — S-2TMß
CCK75895.1|*Oleispira antarctica*

mCpol — TM TM — S-2TMß — Patatin — E2 — E1 — SMODS — JAB
WP_212294906.1|*Acinetobacter baumannii*

#### Other sensor and effector associations

CARF — HEPN — mCpol
WP_066787658.1|*Chloroflexus islandicus*

TPR-S — DRHyd — TIR — CARF — HEPN — mCpol — Macro — TM TM TM TM
HJA34011.1|Candidatus *M. merdigallinarum*

TM TM — SAVED — mCpol
WP_225290043.1|*Nonomuraea aurantiaca*

mCpol — CARF — TM TM — TIR — TIR — 3CC wHTH — CASPASE
WP_106889607.1|*Ahniella affigens*

mCpol — Patatin
WP_147724869.1|*Methylobacterium* sp.WL116

**Extended Data Fig. 9** | See next page for caption.

**Extended Data Fig. 9 | mCpol distribution and gene co-occurrences.**
**(a)** Heat map depicting the phyletic distribution (degree of presence) of selected conflict systems in prokaryotic lineages. Rainbow coloring corresponds to the percentage of presence in a lineage as indicated by the key to the left of the figure: dark gray indicates the absence of a system in each lineage. The phyletically restricted CATRA (Caspase and TPR repeat-associated)- and GSDM (Gasdermin)-containing systems are included as comparisons to the similarly restricted mCpol system. **(b)** Tables depict: (top) percentages of degree of presence in all sampled prokaryotic lineages (self-cell/diagonal) and pairwise co-occurrence (off-diagonal) for selected systems across all prokaryotes and (bottom) $p$-values for the significance of pairwise co-occurrences. **(c)** Venn diagram representing selected system pairs from the above tables. For **b-c**, a two-sided exact Fisher's test was performed. Co-occurrences with $p$-values of less than 1e-06 (passing the stringent co-occurrence criterion) are shaded (red) or provided below the diagram (red), respectively. **(d)** Conserved mCpol gene neighborhood associations, with individual genes depicted as box arrows, colored by their discrete domains. GenBank accessions of mCpol genes are provided below the neighborhoods. "Core" mCpol-encoding operons are separated from discrete co-associating systems by blue circles.

**Extended Data Table 1 | Crystallographic data collection and refinement statistics**

| | *Kp*OptS- apo | *Kp*OptS- Apcpp & Mg/Mn$^{2+}$ |
|---|---|---|
| **Data collection** | | |
| Space group | C 1 2 1 | P 2$_1$ 2$_1$ 2$_1$ |
| Cell dimensions | | |
| $a$, $b$, $c$ (Å) | 114.17, 39.65, 109.20 | 54.42, 89.53, 104.22 |
| $\alpha$, $\beta$, $\gamma$ (°) | 90.00, 91.99, 90.00 | 90.00, 90.00, 90.00 |
| Resolution (Å)* | 37.45–1.75 (1.79–1.75) | 48.24–2.42 (2.52–2.42) |
| $R_{pim}$ | 0.066 (0.905) | 0.079 (0.404) |
| $I / \sigma I$ | 15.5 (1.7) | 11.3 (2.2) |
| Completeness (%) | 98.3 (96.5) | 100.0 (100.0) |
| Redundancy | 6.8 (6.2) | 13.2 (13.2) |
| | | |
| **Refinement** | | |
| Resolution (Å) | 1.75 | 2.42 |
| No. reflections | 48,188 (3414) | 20,062 (1411) |
| $R_{work}$ / $R_{free}$ | 0.2214 / 0.2597 | 0.2112 / 0.2694 |
| No. atoms | | |
| Protein | 3702 | 3713 |
| Ligand/ion | 0 | 78/8 |
| Water | 194 | 123 |
| *B*-factors | 33.14 overall | 35.63 overall |
| Protein | 32.82 | 35.38 |
| Ligand/ion | 0 | 46.49 |
| Water | 39.30 | 36.28 |
| R.m.s. deviations | | |
| Bond lengths (Å) | 0.005 | 0.010 |
| Bond angles (°) | 0.66 | 1.19 |
| **PDB Deposition ID** | 9MNR | 9PD0 |

*Values in parentheses are for highest-resolution shell.

# Reporting Summary

## Statistics

For all statistical analyses, confirm that the following items are present in the figure legend, table legend, main text, or Methods section.

| n/a | Confirmed | |
|---|---|---|
| ☐ | ☒ | The exact sample size (*n*) for each experimental group/condition, given as a discrete number and unit of measurement |
| ☐ | ☒ | A statement on whether measurements were taken from distinct samples or whether the same sample was measured repeatedly |
| ☐ | ☒ | The statistical test(s) used AND whether they are one- or two-sided<br>*Only common tests should be described solely by name; describe more complex techniques in the Methods section.* |
| ☒ | ☐ | A description of all covariates tested |
| ☒ | ☐ | A description of any assumptions or corrections, such as tests of normality and adjustment for multiple comparisons |
| ☐ | ☒ | A full description of the statistical parameters including central tendency (e.g. means) or other basic estimates (e.g. regression coefficient) AND variation (e.g. standard deviation) or associated estimates of uncertainty (e.g. confidence intervals) |
| ☐ | ☒ | For null hypothesis testing, the test statistic (e.g. *F*, *t*, *r*) with confidence intervals, effect sizes, degrees of freedom and *P* value noted<br>*Give P values as exact values whenever suitable.* |
| ☒ | ☐ | For Bayesian analysis, information on the choice of priors and Markov chain Monte Carlo settings |
| ☒ | ☐ | For hierarchical and complex designs, identification of the appropriate level for tests and full reporting of outcomes |
| ☒ | ☐ | Estimates of effect sizes (e.g. Cohen's *d*, Pearson's *r*), indicating how they were calculated |

*Our web collection on statistics for biologists contains articles on many of the points above.*

## Software and code

Policy information about availability of computer code

| Data collection | PSI-BLAST (RRID:SCR_001010), JACKHMMER (RRID:SCR_005305) |
|---|---|
| Data analysis | Coot (v1.1.17), Phenix (v1.21-5207), PyMol (3.1.6.1), ICEflow (autoPROC) pipeline (uses XDS for data indexing and reduction, POINTLESS and AIMLESS for scaling and space group assignment, and TRUNCATE for conversion to structure factors), Geneious Prime (v2024.0.1), GraphPad Prism 9, Alphafold2 and Alphafold3, HHPRED, MMSeqs (RRID:SCR_008184), DALI, Foldseek |

For manuscripts utilizing custom algorithms or software that are central to the research but not yet described in published literature, software must be made available to editors and reviewers. We strongly encourage code deposition in a community repository (e.g. GitHub). See the Nature Portfolio guidelines for submitting code & software for further information.

## Data

Policy information about availability of data

All manuscripts must include a data availability statement. This statement should provide the following information, where applicable:
- Accession codes, unique identifiers, or web links for publicly available datasets
- A description of any restrictions on data availability
- For clinical datasets or third party data, please ensure that the statement adheres to our policy

The crystal structure data for apo KpOptS and KpOptS bound to ApCpp have been deposited in the PDB (9MNR and 9PD0, respectively). All other data supporting the findings of this study are available within the paper and its Supplementary Information. Source data are provided.

# Research involving human participants, their data, or biological material

Policy information about studies with [human participants or human data](). See also policy information about [sex, gender (identity/presentation), and sexual orientation]() and [race, ethnicity and racism]().

| | |
|---|---|
| Reporting on sex and gender | N/A |
| Reporting on race, ethnicity, or other socially relevant groupings | N/A |
| Population characteristics | N/A |
| Recruitment | N/A |
| Ethics oversight | N/A |

Note that full information on the approval of the study protocol must also be provided in the manuscript.

# Field-specific reporting

Please select the one below that is the best fit for your research. If you are not sure, read the appropriate sections before making your selection.

☒ Life sciences ☐ Behavioural & social sciences ☐ Ecological, evolutionary & environmental sciences

For a reference copy of the document with all sections, see [nature.com/documents/nr-reporting-summary-flat.pdf]()

# Life sciences study design

All studies must disclose on these points even when the disclosure is negative.

| | |
|---|---|
| Sample size | No samples size predetermination estimation was conducted. Small sample sizes for phage experiments and in vitro biochemical studies are commonplace and accepted. When possible, n>3 was used. |
| Data exclusions | No data were excluded from the analysis. |
| Replication | All experiments were performed with independent replicates as described in the figure legends. |
| Randomization | Randomization is not relevant to the experimental design described due to small values of n, high degree of control, and limited expected variability across experiments. |
| Blinding | Blinding is not relevant to the experimental design described as no clinical information is relevant to this basic science study. |

# Reporting for specific materials, systems and methods

We require information from authors about some types of materials, experimental systems and methods used in many studies. Here, indicate whether each material, system or method listed is relevant to your study. If you are not sure if a list item applies to your research, read the appropriate section before selecting a response.

## Materials & experimental systems

| n/a | Involved in the study |
|---|---|
| ☒ | ☐ Antibodies |
| ☒ | ☐ Eukaryotic cell lines |
| ☒ | ☐ Palaeontology and archaeology |
| ☒ | ☐ Animals and other organisms |
| ☒ | ☐ Clinical data |
| ☒ | ☐ Dual use research of concern |
| ☒ | ☐ Plants |

## Methods

| n/a | Involved in the study |
|---|---|
| ☒ | ☐ ChIP-seq |
| ☒ | ☐ Flow cytometry |
| ☒ | ☐ MRI-based neuroimaging |

## Plants

Seed stocks

N/A

Novel plant genotypes

N/A

Authentication

N/A

