## [Peer Review File · Nature]

The Panoptes system uses decoy cyclic nucleotides to defend against phage

Corresponding Author: Dr Benjamin Morehouse

Version 0:

Reviewer comments:

Referee #1

(Remarks to the Author)

Sullivan et al. explore the antiphage defence system mCpol, predicted as ancestral to the type III CRISPR system. Here dubbed "Panoptes", the system is shown to provide defence against a subset of phage in the BASEL collection (Figure 1). OptS encodes a nucleotide cyclase which appears tetrameric in solution and in the crystal structure solved here (see point 1). In vitro, OptS synthesises 2',3'-c-di-AMP, which is confirmed by HPLC and standards (figure 3). Escape mutants of phage T4 reveal the key role of the nucleotide sponge protein Acb2 in activation of Panoptes defence (Figure 4) and expression of Acb2 in cells with Panoptes is sufficient to reduce cell viability significantly. A range of phage sponges/phosphodiesterases are then tested to assess Panoptes activation, revealing that Tad1 from some phage also activates the system (Figure 5a), and finally phage T4 with and without Acb2 is shown to be sensitive to Panoptes and a CBASS defence system, respectively (Figure 5b), hinting at the possibility of cooperation of these two defences as part of a bacterial immune system.

Overall this is an exciting finding that reveals novel and unexpected complexity in cyclic nucleotide based antiphage defence. There are significant weaknesses, in particular in the consideration of the likely mCpol cyclase catalytic mechanism wrt Cas10 and implications for quaternary structure. Secondly, there is no data on the effector protein and its behaviour in the presence and absence of cyclic nucleotides – leading to unsupported statements in the text that are really speculation. The Bioinformatic data needs substantive revision. Finally, the stats should all be reanalysed using SD instead of SEM.

This reviewer notes a preprint that has explored the same antiphage defence system, and they are largely complimentary and agree on the most important points. There are some differences in the findings that would be valuable to address.

1. If and when these papers are published, I would urge the authors to agree on a single nomenclature? The field would be grateful if this was agreed before publication. This reader prefers Panoptes, for what it's worth.
2. The papers differ on the effect of Acb1 on activation of the mCpol system. This could be due to the use of different isomers of c-di-AMP by the two enzymes of course.

Main points:

1. Is OptS really a tetramer? The functional unit is the dimer and it's not obvious why tetramers should be relevant. What is the PISA score for the structure – does it strongly support a tetrameric organisation? Size exclusion as shown in figure 2 is not strongly diagnostic for tetramers over dimers – can this be repeated using SEC-MALLS to give a more definitive conclusion? Extended data figure 2 shows three dimer interfaces. Which is most highly supported with PISA and which is likely to be the biologically-relevant unit? Overall, some further consideration of the likely mechanism of OptS is warranted and would strengthen the paper. The authors may wish to draw comparisons with the active site of Cas10 here.
2. Panoptes defence is described here as an Abi system. However, this is not supported by the data in extended data figure 1, where a high MOI results in protection of the culture that is expressing Panoptes. Abi is often "over diagnosed" so without further data the authors may wish to reconsider this statement.
3. Extended data figure 7 shows a bioinformatic analysis of the co-occurrence of various antiphage defence systems with Panoptes.

4. The section headed "Diverse phage proteins activate Panoptes" is a little misleading, as only Tad1 is shown to be an activator. The other proteins are really negative controls, as Apyc1 and AcrIII-1 are known to be specific for completely different signalling molecules, so this section could be refocussed to say that anti-defence proteins targeting cyclic dinucleotides activate Panoptes.
5. Line 329 asserts that c-di-AMP holds OptE in an inactive state. This is certainly a hypothesis but there are no experiments here that directly test this. Can the authors add any data showing that the effector binds the cyclic nucleotide, or visualise the consequences of cNT sequestration in cells expressing the effector?
6. The bioinformatic analysis presented in extended data figure 7 is quite hard to interpret. It is unclear why Cas1 has been used in preference to Cas10 for the analysis, since the latter should provide much stronger (and more relevant) stats. In particular, the linkage between Cas10 and mCpol might be statistically significant. The terms CATRA and GSDM are used here and are presumably other defence systems, but are not mentioned anywhere in the text. Perhaps remove them as they don't seem relevant anyway.

Specific points:

1. Abstract – "2'3'-cyclic adenosine monophosphate" is cAMP, not c-diAMP.
2. Line 59 – maybe just say 6 here rather than 5.8. Seems like this is heading towards 10 anyway.
3. Line 92 typo "which are synthesised"
4. Line 93 how about a reference to SMODS / CD-NTase here?
5. Figure 1,4, 5. For triplicate datasets the appropriate error is SD, not SEM.
6. Acb2 is a trimeric protein with multiple binding sites for cyclic di- and tri-nucleotides. Please explain equation chosen for fitting of ITC data (Ext data fig 5), which purports to be a single site binding model. What stoichiometry of binding is seen here, and does it make sense? Were other models tested?
7. Line 327 the phrase "flips the liability of a .. against the phage" could be written more clearly.
8. Cap15 is described as adopting monomeric and oligomeric states (Line 336). Where is the evidence for these different states?
9. Line 349. I don't understand the phrase "or as different genes" – they are always encoded by different genes.
10. Line 357 rephrase "the proteins of anti-defence".
11. Line 358 The speculation here is interesting but "must encode" is too strong in the absence of data.

Referee #2

(Remarks to the Author)

In this paper by Sullivan et al., they characterise a new antiphage system. This system, termed Panoptes, involves a minimal Cas10 palm domain protein (OptS) that forms a tetramer and synthesises cyclic-di-AMP. The c-di-AMP inhibits the activity of the protein (OptE) encoded by the second gene in the operon. This protein is a transmembrane protein that they demonstrate is toxic to the cell when it is not inhibited by c-di-AMP produced by OptS (akin to a toxin-antitoxin system). They show that the system provides defence against the Straboviridae, which includes well known phages such as T4. Mutations in T4 that enable escape from Panoptes had mutations in the gene encoding Acb2 (Anti-CBASS), which has previously been shown to act as a "sponge" to bind cyclic signals of the CBASS system to enable phages to escape CBASS defence. The Acb2 protein alone was sufficient to trigger the Panoptes system, whereas the escape mutant proteins were non-functional. Acb2 was shown to bind to the cyclic-di-AMP directly and the model proposed is that, by blocking the c-di-AMP signal, OptE is no longer inhibited and now forms pores in the membrane, that result in cell killing. Some other proteins that interfere with nucleotide-based signalling antiphage systems also triggered the defence (i.e. cell killing) properties of Panoptes. The work is interesting, clearly written, well performed and the conclusions are in line with the results presented.

Major Comments

1. Nature recently released an accelerated article preview on the Hailong defence system, which has a similar de-repression mechanism of antiphage activation. The author should include discussion and reference to this system that has a lot of conceptual similarities to the work reported in the current manuscript.
2. The mechanism of Acb1 would potentially be expected to also trigger this system. It is possible that the level of Acb1 expression in T4 under these lab conditions is not sufficient to trigger the system. For completeness in Figure 4, it would be beneficial to examine the effects of Acb1 overexpression, and in Extended Data Figure 5 to examine Acb1 binding to the signal by ITC.
3. There is no direct evidence presented that the OptE protein is membrane localised or that it directly responds to the c-di-AMP molecule. Given that these are key parts of the mechanism, stronger evidence in support of these steps would significantly strengthen the manuscript. The model also includes that the oligomeric state of this protein is altered by c-di-AMP – do the authors have any data supporting this? What are the models of this proteins and how its oligomeric state might alter?
4. No direct evidence of cell death is presented (except the measure of colony forming units). CFU cannot distinguish between inhibition of growth (e.g. dormancy or bacteriostasis) or cell killing. There are many standard methods used (suggest the author look at the toxin-antitoxin literature for examples – eg. cell viability stains).
5. The model suggests that cell death is occurring due to either cell lysis or membrane depolarisation due to membrane damage. No evidence in support of either of these models is provided. The authors could induce OptE and see if the OD of a cell culture decreases (lysis) or measure PMF or other membrane permeability metrics.

6. Although not necessary for the main conclusions, the paper would be improved by testing some of the prediction of many other related systems with different effectors identified in the final results section.
7. The paper would significantly benefit from a schematic model that summarises the proposed mechanism of the antiphage system.
8. The palm active site in CRISPR-Cas Cas10 contains the GGDD motif. Does OptS possess a similar motif that is critical for activity? The authors could perform multiple sequence alignments to investigate conserved motifs, identify these in the obtained structures and test activity of mutants. From the structural analysis, it would be interesting to compare sites across the protomer – are these sites equivalent?

Minor comments

1. The abstract could benefit from some reworking to make it as clear as possible. I found it a bit complex to read at first pass - particularly lines 39-42.
2. L33 – Perhaps “surveying” is not the best word here as it infers an active “surveillance” process, whereas it is merely sensing the concentration of a small molecule that is altered by certain phage proteins.
3. L102 – wording of this heading would be grammatically nicer if rephrased as “Panoptes is an antiphage system” – same point for the figure title.
4. L131 – Authors state that SEC support a tetrameric species in solution (Figure 2a), yet in the legend state it is at least a trimer. The authors should provide what the predicted size is and show how this was obtained.
5. L171 – Why were these residues expected to interact with Mg²⁺? If this was from multiple sequence alignments, then the authors should include this.
6. L257 – the use of a “biosensor” is referred to, which would typically indicate a biological (in vivo) type assay. Looking at the assay used (which is fine), it appears to be more of an in vitro biochemical activity assay. Suggest adding a line in the main text explaining the assay that is used and remove the term “bioassay”.
7. Colours of the lines in some of the figures is quite hard to tell the difference – suggest that more contrasting/different colours are used.
8. Some of the main figures were quite simple - could consider combining some figures or bringing some more of the supp data into the main – e.g. the MS spectra could have been in main.
9. Figure 1A. would be useful to include a nt scale bar and also perhaps a couple of genes either side to indicate the genomic context of this system.
10. Figure 2B. I thought it would be useful to have more information / zoom in on this tetramer, which is the active form of the protein. E.g. it is not really clear immediately where the active sites are positioned.
11. Figure 2D/E. These are useful, but could potentially be supplementary. I thought it might be a bit misleading for readers less familiar with Cas10(Csm1) that only a small portion of the protein is shown in this figure – perhaps having a greyed out portion of the rest of the protein would help people see how this is only a small sub-part that has similarity to the OptS protein.
12. Figure 4C. Does the dash above Acb2 refer to the wild-type version? This should be made clear. A dash typically means in the absence of, which I don't think is the intention.
13. Figure 4D. Include the error with the reported K_d measurement.
14. Figure 4E. Explain what the optSCD condition represents. This text was particularly difficult to read.
15. Figure 5B. The way this data is plotted was quite confusing. The legend states it is an EOP but it is presented as a fold-protection. Since none of the bars are set at fold = 1 it is not clear what this is normalised relative to. The authors should provide representative raw data that shows the differences in fold protection.
16. Not clear why SEM rather than SD was used in the figure plots.
17. There does not appear to be any statistical analysis applied to the differences reported in the graphs.
18. Extended data Fig 1. Why does T4 appear to keep going up in titre when defence is supposedly active?
19. Extended data Fig 2. It would be nice to add the tetramer to this figure and help relate these dimer interface images to the overall tetramer.

20. Extended data Fig 6. Did the authors need to plot this standard curve to determine the estimated concentration of c-di-AMP? If so, it would be good to include the standard curve and where the test samples fall on the curve (e.g. to support the estimate presented in the text).

21. Extended data Fig 7. D is too small and hard to see (but very interesting and important). In fact, that could be considered to be moved into the main text. I found the presentation of this figure (A-C) a little hard to follow. Can these be presented more clearly? E.g. heat map scale in A is too tiny to read. Are some of the squares in the heat map representing no hits? If so, what colour are they? This is not mentioned in the legend.

22. References, it may be good to included more recent reviews of anti-phage systems and phage anti-defence proteins.

23. In the discussion it might be nice to elaborate on the idea that Panoptes systems have the challenge of needing to respond to anti-defence proteins that recognise nucleotide signals similar (but different) to those that they are primarily “trying” to inhibit. If Panoptes makes the same signals as the co-encoded defence then the cell would die.

24. The concept that proteins can respond oppositely (activated or repressed) by small molecular inducers is a well-known concept in regulators of gene expression in bacteria and it might be beneficial to draw that parallel here – you can either activate killing or de-repress to result in killing! Same overall outcome.

Referee #3

(Remarks to the Author)

The authors present a compelling study on a newly identified antiphage defense system, termed Panoptes, which operates via a mechanism distinct from other known systems that rely on the production of nucleotide-derived second messengers to activate effector proteins and block phage replication. In this system, a two-gene operon (optSE) encodes the enzymes responsible for defense. OptS synthesizes 2',3'-cyclic adenosine monophosphate (2',3'-cAMP), which binds to OptE, a transmembrane effector protein, and inhibits its dimerization. In the absence of this cyclic nucleotide, OptE forms dimers that trigger host cell death. This defense mechanism is effective against phages from the Straboviridae family, including T4 phage. Notably, the authors also quantify intracellular levels of 2',3'-c-di-AMP in cells expressing optSE, providing strong evidence for its physiological relevance. Overall, the manuscript is well written, the experiments are carefully designed, and the data are presented with clarity and rigor, and will be of interest to a broad audience. It is suitable for publication in Nature provided the authors address some minor suggestions.

Comments:

Methods

- Line 113: Missing the word “family” after Straboviridae?
- In all instances where centrifugation speeds are given in RPM, please give the speed in x g so that readers can more easily reproduce the methods without having to look up the conversion factor for the particular rotor used.
- The methods state that Sanger sequencing was performed by plasmidsaurus. To the best of my knowledge, plasmidsaurus performs oxford nanopore sequencing not Sanger sequencing.

David Taylor and Tyler Dangerfield

Referee #4

(Remarks to the Author)

I co-reviewed this manuscript with one of the reviewers who provided the listed reports.

Referee #5

(Remarks to the Author)

I co-reviewed this manuscript with one of the reviewers who provided the listed reports.

Version 1:

Reviewer comments:

Referee #1

(Remarks to the Author)

The authors have addressed all my concerns, added significant new data and adjusted the text. I am happy to see this manuscript proceed to publication.

Referee #2

(Remarks to the Author)

The authors have done an excellent job at addressing the reviewers comments adequately. I have no further comments.

Referee #3

(Remarks to the Author)

These reviewers are satisfied with their response. The authors did quite a bit more than expected, showing that the enzyme can synthesize a variety of cyclic nucleotide products. Nice results! We recommend publication in Nature.

David Taylor and Tyler Dangerfield

Referee #4

(Remarks to the Author)

I co-reviewed this manuscript with one of the reviewers who provided the listed reports.

Referee #5

(Remarks to the Author)

I co-reviewed this manuscript with one of the reviewers who provided the listed reports.

Response to Reviewers

We thank the reviewers for their comments and suggestions. Below, we address each critique in **blue** text. In addition, we have included revised copies of the manuscript with and without changes highlighted from the previous draft.

The submitted manuscript has been revised to address each of the points raised by the reviewers. The major changes to the document include:

- Addition of data and analysis to characterize the oligomeric state of *KpOptS* (Fig. 2a, ED Fig. 2b-h and Supplementary Table 1)
- Addition of a crystal structure of *KpOptS* in complex with nonhydrolyzable ATP analog that enabled explanation of nucleotide preference of *Kp* and *VnOptS* (Fig. 3a, ED Fig. 5a and b).
- New biochemical characterization of *VnOptS* showcasing a more promiscuous production of a variety of cyclic dinucleotide products relative to *KpOptS* (Fig. 3b, ED Fig. 5c-l, and ED Fig. 6).
- New data supporting OptE effector binding to cyclic dinucleotides (Fig. 3c, ED Fig. 3).
- New microscopic analysis of bacteria expressing OptSE ± Acb2 that shows membrane disruption that is both Acb2 and OptE dependent (Fig. 4f).
- Inclusion of additional statistics throughout the manuscript
- Substantial revisions of the text

During the review process, we also significantly improved other aspects of the manuscript as part of reconciling some of the differences between our work and the manuscript by Doherty and Adler et al. (PMID: 40196485). Specifically, we originally were unable to observe phage defense for *KpOptSE* whereas a highly similar sequence provided robust phage defense in Doherty and Adler et al. We found that this was due to differences in the growth medium used in our experiments. The majority of our initial characterization was performed in M9 Glucose medium while Doherty and Adler et al. predominantly used LB. Surprisingly, *KpOptSE* only provides phage defense in LB, not M9 Glucose. The significance and mechanism are unknown; however, we have included these details and repeated much of our work in both media.

Reviewer Comments:

Referee #1 (Remarks to the Author):

Sullivan et al. explore the antiphage defence system mCpol, predicted as ancestral to the type III CRISPR system. Here dubbed “Panoptes”, the system is shown to provide defence against a subset of phage in the BASEL collection (Figure 1). OptS encodes a nucleotide cyclase which appears tetrameric in solution and in the crystal structure solved here (see point 1). In vitro, OptS synthesises 2',3'-c-di-AMP, which is confirmed by HPLC and standards (figure 3). Escape mutants of phage T4 reveal the key role of the nucleotide sponge protein Acb2 in activation of Panoptes defence (Figure 4) and expression of Acb2 in cells with Panoptes is sufficient to reduce cell viability significantly. A range of phage sponges/phosphodiesterases are then tested to assess Panoptes activation, revealing that Tad1 from some phage also activates the system (Figure 5a), and finally phage T4 with and without Acb2 is shown to be sensitive to Panoptes and a CBASS defence system, respectively (Figure 5b), hinting at the possibility of cooperation of these two defences as part of a bacterial immune system.

Overall this is an exciting finding that reveals novel and unexpected complexity in cyclic nucleotide based antiphage defence. There are significant weaknesses, in particular in the consideration of the likely mCpol cyclase catalytic mechanism wrt Cas10 and implications for quaternary structure. Secondly, there is no data on the effector protein and its behaviour in the presence and absence of cyclic nucleotides – leading to unsupported statements in the text that are really speculation. The Bioinformatic data needs substantive revision. Finally, the stats should all be reanalysed using SD instead of SEM.

This reviewer notes a preprint that has explored the same antiphage defence system, and they are largely complimentary and agree on the most important points. There are some differences in the findings that would be valuable to address.

1. If and when these papers are published, I would urge the authors to agree on a single nomenclature? The field would be grateful if this was agreed before publication. This reader prefers Panoptes, for what it's worth.
2. The papers differ on the effect of Acb1 on activation of the mCpol system. This could be due to the use of different isomers of c-di-AMP by the two enzymes of course.

We thank the reviewer for their careful consideration of our manuscript. Below, we address each of the main points individually. For (1) in the general comments, we have communicated with the authors of Doherty and Adler et al. (PMID: 40196485) and together adopted a unified name for this defense system of *Panoptes*. For (2), we now include additional data that helps explain some differences observed for T4 *acb1*. *KpOptSE* proteins are identical to ECOR31 OptSE reported in Doherty and Adler et al. at the amino acid level.

Main points:

1. Is OptS really a tetramer? The functional unit is the dimer and it's not obvious why tetramers should be relevant. What is the PISA score for the structure – does it strongly support a tetrameric organisation? Size exclusion as shown in figure 2 is not strongly diagnostic for tetramers over dimers – can this be repeated using SEC-MALLS to give a more definitive conclusion? Extended data figure 2 shows three dimer interfaces. Which is most highly supported with PISA and which is likely to be the biologically-relevant unit? Overall, some further consideration of the likely mechanism of OptS is warranted and would strengthen the paper. The authors may wish to draw comparisons with the active site of Cas10 here.

The first method we used to address this question was PISA score analysis, as suggested by the reviewer. These data showed a major interaction at the monomer-monomer interface, supporting the dimer, with buried surface area of $\sim 1180 \text{ \AA}^2$ and a minor interaction at the dimer-dimer interface, supporting the tetramer, with buried surface area of $\sim 500 \text{ \AA}^2$. These data have been included in the revised submission as Supplementary Table 1 and partly included in Extended Data Fig. 2c-e along with structures of the interfaces.

The second method we used was SEC-MALS of *KpOptS*. Unfortunately, we were unable to obtain usable data on the equipment available, and these experiments failed for technical reasons. While the protein behaved well during the initial size exclusion chromatography at 4 °C, *KpOptS* aggregated in SEC-MALS. This could be due to higher concentration of protein or analysis at room temperature, where the SEC-MALS instrument is maintained. We agree with the reviewer that these experiments would be a valuable addition; however, obtaining these results would require an indeterminant amount of time to troubleshoot.

The third method we used was mass photometry of *KpOptS*. Here, we succeeded in observing a ~ 59 kDa species (expected mass of the monomer = 14.6 kDa, tetramer = 58.4 kDa) when the protein was at a concentration of 50 nM in solution (Extended Data Fig. 2g and h). These data demonstrate that the tetramer is stable and detectable. However, the caveat of these data is that mass photometry is unable to detect species of ≤ 30 kDa (Refeyn Ltd., manufacturer description). For this reason, we cannot comment on the relative ratio of monomer:dimer:tetramer in these conditions.

We agree with the reviewer that our current size exclusion chromatography is not conclusive. The PISA analysis also does not provide unambiguous support for biologically relevant tetramer interactions. However, our mass photometry supports that the *KpOptS* tetrameric species is observable and stable at low protein concentrations. We have synthesized these observations and data in the results section but made changes in the text to ensure we do not overstate our conclusions. In addition, we have compared the active sites of *Kp* and *VnOptS*, along with Cas10 in the revised manuscript.

2. Panoptes defence is described here as an Abi system. However, this is not supported by the data in extended data figure 1, where a high MOI results in protection of the culture that is expressing Panoptes. Abi is often “over diagnosed” so without further data the authors may wish to reconsider this statement.

We agree with the reviewer and have updated the text of our manuscript to not make this claim. We have repeated our growth curve of bacteria infected with multiple MOIs of phage in conditions that resulted in a higher level of phage defense and revised Extended Data Fig. 1a. These data are not conclusive in supporting abortive infection. We have also now included microscopy that shows membrane disruption upon co-expression of *Acb2* with *VnOptSE*, these also do not clearly show abortive infection.

3. Extended data figure 7 shows a bioinformatic analysis of the co-occurrence of various antiphage defence systems with Panoptes.

Please find our complete response in point #6, below.

4. The section headed “Diverse phage proteins activate Panoptes” is a little misleading, as only Tad1 is shown to be an activator. The other proteins are really negative controls, as *Apyc1* and *AcrIII-1* are known to be specific for completely different signalling molecules, so this section

could be refocussed to say that anti-defence proteins targeting cyclic dinucleotides activate Panoptes.

We agree with the reviewer and have made the relevant change to the section heading.

5. Line 329 asserts that c-di-AMP holds OptE in an inactive state. This is certainly a hypothesis but there are no experiments here that directly test this. Can the authors add any data showing that the effector binds the cyclic nucleotide, or visualise the consequences of cNT sequestration in cells expressing the effector?

We have now included data in the revised manuscript to show that the soluble domain of VnOptE binds 2',3'-c-di-AMP as well as the other major products of VnOptS including 3',2'-cGAMP and 3',2'-cUAMP (Fig. 3b and c, Extended Data Figs. 5 and 6). To understand the functional implications *in vivo*, we have visualized membrane integrity by microscopy (Fig. 4f). Co-expression of Acb2 with VnOptSE resulted in cell permeability to propidium iodide, which indicates a loss of membrane integrity. This phenotype was OptE and Acb2 dependent. We have included these data now to support the hypothesis that 2',3'-c-di-AMP holds OptE in an inactive state and have made sure to modify the text to reflect our experimental results.

We were unable to produce recombinant, full-length Kp or VnOptE and therefore have not been able to evaluate changes in oligomerization resulting from nucleotide binding. However, a contemporaneous report by Doherty and Adler et al. (PMID: 40196485) shows that *apo* OptE (Δ 2TM) oligomerizes while nucleotide binding disrupts large oligomer formation. We have included these details and a citation in the manuscript.

6. The bioinformatic analysis presented in extended data figure 7 is quite hard to interpret. It is unclear why Cas1 has been used in preference to Cas10 for the analysis, since the latter should provide much stronger (and more relevant) stats. In particular, the linkage between Cas10 and mCpol might be statistically significant. The terms CATRA and GSDM are used here and are presumably other defence systems, but are not mentioned anywhere in the text. Perhaps remove them as they don't seem relevant anyway.

We have now modified Extended Data Fig. 9 to include Cas10. We have increased the font and added additional explanations to the legend to help with its interpretation. It should be noted that Cas1 is the single most diagnostic component of all known CRISPR/Cas systems. Hence, we believe having it is relevant as it allows us to assess the overlap with CRISPR/Cas systems as a whole. Cas10 and mCpol, being evolutionarily related, are both signaling nucleotide-generating enzymes; however, none of the conserved gene-neighborhood contexts indicate any kind of functional coupling between them, of the kind seen between mCpol and the S-2TM β and SMODS proteins. Consistent with this, we only have borderline significance for the co-occurrence of Cas10 and mCpol across bacterial genomes (revised Extended Data Fig. 7c). Likewise, the occurrence between S-2TM β and Cas10 does not cross the stringent significance threshold. In contrast, we see a significant co-occurrence between Cas10 and CARF, which is concordant with their previously confirmed functional coupling. As for CATRA and GSDM (we have now provided explanations for them: CATRA: A caspase-based 3-component prokaryotic immune system, and GSDM is the Gasdermin-centric system), we have included them as they are important "negative controls". They represent rarer systems comparable to mCpol systems but do not operate in a nucleotide-dependent manner. Hence, the insignificant statistics of overlap with them show that we are capturing a genuine signal in the cases where we have a significant phyletic overlap rather than an artifact of distribution.

Specific points:

1. Abstract – “2’3’-cyclic adenosine monophosphate” is cAMP, not c-diAMP.

Thank you for catching this mistake, the text has been corrected.

2. Line 59 – maybe just say 6 here rather than 5.8. Seems like this is heading towards 10 anyway.

We have made the suggested edit.

3. Line 92 typo “which are synthesised”

Typo fixed.

4. Line 93 how about a reference to SMOGS / CD-NTase here?

We have added references as suggested.

5. Figure 1,4, 5. For triplicate datasets the appropriate error is SD, not SEM.

We thank the reviewer for their question, which led to some (overdue) reading on the subject of error bar selection.

In the revised submission of the manuscript, we now include all source data for each graph and statistical comparisons of data. Note that the statistical tests to determine significance are not influenced by choice of values used for error bars.

In choosing between standard error of the mean (SEM) and standard deviation (SD), we consulted three publications: the *Nature* recommended Krzywinski & Altman (PMID: 24161969), Tang et al. 2021 (PMID: 34012702), and Cumming et al. 2007 (PMID: 17420288). In our view, the primary purpose of the error bar is to (1) estimate how far our measured mean differs from the true mean and (2) estimate the statistical significance of the difference. These measures are best captured by SEM and not SD. An excerpt from Tang et al. put this best:

“In most scientific data presentations with error bars, the goal is often to compare two or more population means. Although the population means are unknown, for the purpose of making a reliable inference, it is of more interest how far the estimated mean (not an individual observation) is from the true population mean. Therefore, the variability of the estimated means (i.e., SEM) suits the situation better than the SD.

The use of SEM also may enable one to make simple conclusions by visual inspection, because SEM is closely related to the confidence interval and p -value. For example, when comparing means, consider the popular 2-sample Student’s t -test. If the SEM bars of two groups touch when plotted as box plots side-by-side, it usually implies that the test statistic t is 1.41 or less, corresponding to a p -value greater than 0.15. For a visual display, if the sample size is 10 or more and both groups have similar SEMs, a gap of $1 \times \text{SEM}$ corresponds to $p \approx 0.05$ and $2 \times \text{SEM}$ corresponds to $p \approx 0.01$. For smaller sample sizes, larger gaps are needed to get the same p -values. In contrast, error bars using SD cannot easily suggest these conclusions visually.”

Cumming et al. describes SEM as an inferential error bar and SD as a descriptive error bar. Here, we hope to allow the reader to infer the statistical significance of our data.

We have revised our Methods subsection on “Statistics and reproducibility” to include the citations relevant to our choice of error bar statistics.

6. Acb2 is a trimeric protein with multiple binding sites for cyclic di- and tri-nucleotides. Please explain equation chosen for fitting of ITC data (Ext data fig 5), which purports to be a single site binding model. What stoichiometry of binding is seen here, and does it make sense? Were other models tested?

We selected the “One Set of Sites” model, which is appropriate to use for binding between a macromolecule and a ligand that includes any number of binding sites (e.g. three cyclic dinucleotide sites on Acb2) in which all sites have the same K (association constant) and ΔH . The alternative option is the “Two Set of Sites” model, which must be used if the binding sites on a macromolecule have different values of K and/or ΔH . Based on previous studies in which ITC was used to determine the dissociation constants between Acb2 and various cyclic dinucleotides (Huiting et al., 2023, PMID: 36750095, & Jenson et al., 2023, PMID: 36848932) there are no data supporting that the binding sites possess different thermodynamic values, and thus it is more appropriate to use the “One Set of Sites” model. We have expanded our ITC methods section to include this information.

We analyzed the data based on the concentration of 2',3'-c-di-AMP binding sites (i.e., 3 sites per Acb2 hexamer), as was done in Jenson et al., 2023. The values of “n” we observe are similar to those in Huiting et al., 2023, where ITC was used to measure binding of various cyclic dinucleotides to Acb2.

7. Line 327 the phrase “flips the liability of a .. against the phage” could be written more clearly.

We have revised this sentence to make our point more intuitive. This sentence now reads: “Here, we discovered that the Panoptes antiphage system uses the vulnerability of a nucleotide-derived second messenger in immune signaling against phage by detecting the activity of counter-defense proteins.”

8. Cap15 is described as adopting monomeric and oligomeric states (Line 336). Where is the evidence for these different states?

We have added a reference to these findings (Duncan-Lowey et al., 2021, PMID: 34784509).

9. Line 349. I don't understand the phrase “or as different genes” – they are always encoded by different genes.

We apologize for the confusion as this sentence contained a typo. We have corrected this sentence to “or in different genomes.”

10. Line 357 rephrase “the proteins of anti-defense”.

We have revised this sentence to increase clarity.

11. Line 358 The speculation here is interesting but “must encode” is too strong in the absence

of data.

We have, perhaps, let our exuberance get the better of us. The sentence has been revised to “may encode.”

Referee #2 (Remarks to the Author):

In this paper by Sullivan et al., they characterise a new antiphage system. This system, termed Panoptes, involves a minimal Cas10 palm domain protein (OptS) that forms a tetramer and synthesises cyclic-di-AMP. The c-di-AMP inhibits the activity of the protein (OptE) encoded by the second gene in the operon. This protein is a transmembrane protein that they demonstrate is toxic to the cell when it is not inhibited by c-di-AMP produced by OptS (akin to a toxin-antitoxin system). They show that the system provides defence against the Straboviridae, which includes well known phages such as T4. Mutations in T4 that enable escape from Panoptes had mutations in the gene encoding Acb2 (Anti-CBASS), which has previously been shown to act as a “sponge” to bind cyclic signals of the CBASS system to enables phages to escape CBASS defence. The Acb2 protein alone was sufficient to trigger the Panoptes system, whereas the escape mutant proteins were non-functional. Acb2 was shown to bind to the cyclic-di-AMP directly and the model proposed is that, by blocking the c-di-AMP signal, OptE is no longer inhibited and now forms pores in the membrane, that result in cell killing. Some other proteins that interfere with nucleotide-based signalling antiphage systems also triggered the defence (i.e. cell killing) properties of Panoptes. The work is interesting, clearly written, well performed and the conclusions are in line with the results presented.

We thank the reviewer for their kind words and thoughtful critique.

Major Comments

1. Nature recently released an accelerated article preview on the Hailong defence system, which has a similar de-repression mechanism of antiphage activation. The author should include discussion and reference to this system that has a lot of conceptual similarities to the work reported in the current manuscript.

Thank you for this suggestion. We have revised the discussion to include the Hailong system and now discuss the similarities in sensing anti-defense.

2. The mechanism of Acb1 would potentially be expected to also trigger this system. It is possible that the level of Acb1 expression in T4 under these lab conditions is not sufficient to trigger the system. For completeness in Figure 4, it would be beneficial to examine the effects of Acb1 overexpression, and in Extended Data Figure 5 to examine Acb1 binding to the signal by ITC.

We have now included new data for Acb1 when co-expressed with VnOptSE (Fig. 5a). In these conditions, we find that Acb1 is sufficient to activate OptSE-mediated growth inhibition. These results are surprising as deletion of *acb1* had minimal impact on OptSE-mediated defense against T4 (Fig. 4b). However, as the reviewer points out, these differences are likely due to Acb1 expression level, which may be higher in plasmid-based co-expression vs. phage infection.

We did not perform additional biochemistry on Acb1 protein in order to maintain the scope of our manuscript. However, we have provided a detailed description of our findings on Acb1, consideration for hypotheses that explain our findings, and explained observations made by a

contemporaneous report by Doherty and Adler et al. (PMID: 40196485) that found a modest role for Acb1 during phage infection.

3. There is no direct evidence presented that the OptE protein is membrane localised or that it directly responds to the c-di-AMP molecule. Given that these are key parts of the mechanism, stronger evidence in support of these steps would significantly strengthen the manuscript. The model also includes that the oligomeric state of this protein is altered by c-di-AMP – do the authors have any data supporting this? What are the models of this proteins and how its oligomeric state might alter?

We agree with the reviewer that additional data is required to support claims made in our initial manuscript submission. We have now included new experimental evidence for OptE nucleotide binding and made alterations to the text to ensure we have not overstated our conclusions. For a complete description of these changes, please see Reviewer 1, Main Point 5.

4. No direct evidence of cell death is presented (except the measure of colony forming units). CFU cannot distinguish between inhibition of growth (e.g. dormancy or bacteriostasis) or cell killing. There are many standard methods used (suggest the author look at the toxin-antitoxin literature for examples – eg. cell viability stains).

Note, this response is for both comments #4 and# 5:

We have now included microscopy analysis of bacteria while co-expressing OptSE and Acb2 (Fig. 4f). These data show that OptSE activation results in membrane disruption that is dependent on Acb2 and OptE. PI staining resulting from membrane permeability is typically a sign that bacteria are no longer viable. However, a growth curve of bacteria expressing OptSE and infected with phage T4 at multiple MOIs failed to show bacteriolysis and these data are not sufficient to label OptSE an abortive infection system. Accordingly, we have decided to modify our conclusions and not advocate for whether OptSE activation results in cell killing, dormancy, or bacteriostasis. Rather, we are simply describing the bacteria as having membrane damage and being unable to grow in the investigated conditions.

5. The model suggests that cell death is occurring due to either cell lysis or membrane depolarisation due to membrane damage. No evidence in support of either of these models is provided. The authors could induce OptE and see if the OD of a cell culture decreases (lysis) or measure PMF or other membrane permeability metrics.

We have responded to comments #4 and #5 together, following comment #4.

6. Although not necessary for the main conclusions, the paper would be improved by testing some of the prediction of many other related systems with different effectors identified in the final results section.

We agree that these related systems are very interesting but their investigation is outside the scope of what was possible during this revision timeline. These additional Panoptes variations will be the subject of future investigations.

7. The paper would significantly benefit from a schematic model that summarises the proposed mechanism of the antiphage system.

We thank the reviewer for their suggestion and have now included a model as Fig. 5d.

8. The palm active site in CRISPR-Cas Cas10 contains the GGDD motif. Does OptS possess a similar motif that is critical for activity? The authors could perform multiple sequence alignments to investigate conserved motifs, identify these in the obtained structures and test activity of mutants. From the structural analysis, it would be interesting to compare sites across the protomer – are these sites equivalent?

We have now included an alignment of OptS, c-di-GMP synthases, and Cas10 enzymes to investigate the conservation of the GGDEF motif in these related proteins (ED Fig. 2a). Further, we have included an additional crystal structure of *Kp*OptS with non-hydrolysable nucleotide to understand the active site architecture (Fig. 3a, ED Fig. 5a and b). These and other details have now been described in substantive text edits found mainly in the results subsections.

In answer to the reviewer's question, yes, OptS does possess a similar “DD/DE” portion of the GGDEF motif found in Cas10 and c-di-GMP synthases. However, these enzymes have lost the “GG” portion. In support of these “DD” residues being involved in catalysis, we previously showed that *Kp*OptS D57 is required for synthesis of 2',3'-c-di-AMP and provide new data that the same is true for D58 in *Vn*OptS, which also is required for cyclic nucleotide production (ED Fig. 5j).

In our new crystal structure of *Kp*OptS, we observe sufficient density to track all of the atoms of only one molecule of non-cyclizable ATP (ApCpp) and chose to model only the triphosphates for the other three ligand binding sites. Thus, we cannot say for sure if the sites are 100% identical as the nucleobase positioning could be very different across the protomers. A more detailed description of these findings and direct comparisons to Cas10 nucleotide binding sites is now provided in the main text.

Minor comments

1. The abstract could benefit from some reworking to make it as clear as possible. I found it a bit complex to read at first pass - particularly lines 39-42.

We have substantially revised the abstract to improve readability.

2. L33 – Perhaps “surveying” is not the best word here as it infers and active “surveillance” process, whereas it is merely sensing the concentration of a small molecule that is altered by certain phage proteins.

We selected the word “surveying” over “surveilling” for the abstract in order to emphasize the passive nature of the system. However, we think that the Panoptes system is providing a form of surveillance at the molecular level and provides a useful analogy. At any rate, the abstract has been edited and the offending word choice removed.

3. L102 – wording of this heading would be grammatically nicer if rephrased as “Panoptes is an antiphage system” – same point for the figure title.

We agree with the reviewer and have updated the title of this subsection.

4. L131 – Authors state that SEC support a tetrameric species in solution (Figure 2a), yet in the legend state it is at least a trimer. The authors should provide what the predicted size is and show how this was obtained.

We now include a detailed investigation and discussion of the oligomeric state of OptS in the manuscript. Briefly, our initial SEC analysis is not at a high enough resolution that we can conclusively determine dimer vs. tetramer (also pointed out by Reviewer 1, main point 1). We attempted to use SEC-MALS and detailed analysis of the OptS structures presented in the manuscript, however, received uninterpretable results. The most conclusive data we were able to gather is mass photometry, which demonstrated a stable *KpOptS* tetramer at low protein concentrations but could not determine the relative ratios of monomer:dimer:tetramer (ED Fig. 2g and h).

5. L171 – Why were these residues expected to interact with Mg²⁺? If this was from multiple sequence alignments, then the authors should include this.

Yes, this inference was made from sequence alignments. We now include an alignment of OptS with related nucleotidyltransferases and enhance our discussion of these findings (ED Fig. 2a).

6. L257 – the use of a “biosensor” is referred to, which would typically indicate a biological (in vivo) type assay. Looking at the assay used (which is fine), it appears to be more of an in vitro biochemical activity assay. Suggest adding a line in the main text explaining the assay that is used and remove the term “bioassay”.

We have removed the terms “biosensor”/“bioassay” in the suggested location of the text.

7. Colours of the lines in some of the figures is quite hard to tell the difference – suggest that more contrasting/different colours are used.

We have selected alternative colors or added symbols/dashed lines to ease comparisons where possible.

8. Some of the main figures were quite simple - could consider combining some figures or bringing some more of the supp data into the main – e.g. the MS spectra could have been in main.

We have added new data and thus rearranged the figures throughout the manuscript.

9. Figure 1A. would be useful to include a nt scale bar and also perhaps a couple of genes either side to indicate the genomic context of this system.

We have added a scale bar to Fig. 1a and Extended Data Fig. 1c.

10. Figure 2B. I thought it would be useful to have more information / zoom in on this tetramer, which is the active form of the protein. E.g. it is not really clear immediately where the active sites are positioned.

We have now modified Fig. 2 and added new figure panels to provide additional active site views and added a crystal structure of *KpOptS* bound to non-cyclizable ATP analog ApCpp which improves our ability to analyze the active site.

11. Figure 2D/E. These are useful, but could potentially be supplementary. I thought it might be a bit misleading for readers less familiar with Cas10(Csm1) that only a small portion of the protein is shown in this figure – perhaps having a greyed out portion of the rest of the protein

would help people see how this is only a small sub-part that has similarity to the OptS protein.

We thank the reviewer for the suggestion to add more of Cas10(Csm1) structure to provide visual context for the catalytic palm domain. We have now modified the superposition to reflect this suggestion.

12. Figure 4C. Does the dash above Acb2 refer to the wild-type version? This should be made clear. A dash typically means in the absence of, which I don't think is the intention.

We have replaced the dash mark with "WT" for clarity in Fig. 4c and ED Fig. 7c-f.

13. Figure 4D. Include the error with the reported K_d measurement.

We have added the error in the K_d measurement for all ITC data in the manuscript.

14. Figure 4E. Explain what the optSCD condition represents. This text was particularly difficult to read.

We have revised the text and figure legend to increase clarity.

15. Figure 5B. The way this data is plotted was quite confusing. The legend states it is an EOP but it is presented as a fold-protection. Since none of the bars are set at fold = 1 it is not clear what this is normalised relative to. The authors should provide representative raw data that shows the differences in fold protection.

We have revised the figure legend to explain how fold defense is calculated and provided a graph of the raw efficiency of plating as Extended Data Fig. 7f.

16. Not clear why SEM rather than SD was used in the figure plots.

Please see Reviewer 1, Minor Comment #5 for an explanation as to why we have chosen to use the inferential error bar of SEM instead of the descriptive error bar of SD in our figures.

17. There does not appear to be any statistical analysis applied to the differences reported in the graphs.

We have carried out statistical analyses of important comparisons throughout the manuscript and incorporated these data in the Figures, Extended Data Figures, and Source Data where relevant. When four or fewer comparisons are made in a single graph, statistical comparisons are displayed. In all other cases, statistics are presented in the Source Data tables to aid with clarity. These and other explanations have been incorporated into the *Statistics and reproducibility* subsection of the Methods.

18. Extended data Fig 1. Why does T4 appear to keep going up in titre when defence is supposedly active?

We have repeated our analysis presented in Extended Data Fig. 1 in alternative conditions and continue to observe this phenomenon. The results are, indeed, surprising. Nevertheless, the impact of OptSE on OD₆₀₀ and plaque formation is unambiguous. We hypothesize that the slow rate of increase in phage replication during liquid growth could be due to incomplete phage

defense across the population. One explanation could be that in the selected media, activation of OptSE only slows phage replication by disrupting processes that require an intact membrane (i.e. require proton motive force). Alternatively, stochasticity in OptSE expression level or function may allow for infrequent replication events that slowly grow in a large, well-mixed population.

19. Extended data Fig 2. It would be nice to add the tetramer to this figure and help relate these dimer interface images to the overall tetramer.

We have now added the tetramer to Extended Data Fig. 2b as suggested.

20. Extended data Fig 6. Did the authors need to plot this standard curve to determine the estimated concentration of c-di-AMP? If so, it would be good to include the standard curve and where the test samples fall on the curve (e.g. to support the estimate presented in the text).

We agree with the reviewer that a more quantitative analysis of a standard curve would improve our estimation of intracellular nucleotide concentration. In the revised manuscript, we now perform a fluorometric assay to measure rate of DNA degradation, which allowed us to construct a 2',3'-c-di-AMP standard curve and interpolate the intracellular concentration of 2',3'-c-di-AMP from bacterial lysates (ED Fig. 4). These data have been added to the manuscript and we have replaced our initial estimates.

21. Extended data Fig 7. D is too small and hard to see (but very interesting and important). In fact, that could be considered to be moved into the main text. I found the presentation of this figure (A-C) a little hard to follow. Can these be presented more clearly? E.g. heat map scale in A is too tiny to read. Are some of the squares in the heat map representing no hits? If so, what colour are they? This is not mentioned in the legend.

We have increased the font size of the panels of the ED Figure 9 as indicated by the referee. We have also added additional explanation to the legend to clarify these data. We were able to move a portion of these data to the main text, however, we did not have sufficient space to move any other figures. We have also now included a supplementary table of accession numbers for mCppl domain containing systems.

22. References, it may be good to included more recent reviews of anti-phage systems and phage anti-defence proteins.

We have now added more recent review citations.

23. In the discussion it might be nice to elaborate on the idea that Panoptes systems have the challenge of needing to respond to anti-defence proteins that recognise nucleotide signals similar (but different) to those that they are primarily "trying" to inhibit. If Panoptes makes the same signals as the co-encoded defence then the cell would die.

We agree and have tried to incorporate/clarify these ideas in the Discussion text.

24. The concept that proteins can respond oppositely (activated or repressed) by small molecular inducers is a well-known concept in regulators of gene expression in bacteria and it might be beneficial to draw that parallel here – you can either activate killing or de-repress to result in killing! Same overall outcome.

We have tried to incorporate these ideas in the context of immune signaling into our discussion section.

Referee #3 (Remarks to the Author):

The authors present a compelling study on a newly identified antiphage defense system, termed Panoptes, which operates via a mechanism distinct from other known systems that rely on the production of nucleotide-derived second messengers to activate effector proteins and block phage replication. In this system, a two-gene operon (*optSE*) encodes the enzymes responsible for defense. *OptS* synthesizes 2',3'-cyclic adenosine monophosphate (2',3'-cAMP), which binds to *OptE*, a transmembrane effector protein, and inhibits its dimerization. In the absence of this cyclic nucleotide, *OptE* forms dimers that trigger host cell death. This defense mechanism is effective against phages from the *Straboviridae* family, including T4 phage. Notably, the authors also quantify intracellular levels of 2',3'-c-di-AMP in cells expressing *optSE*, providing strong evidence for its physiological relevance. Overall, the manuscript is well written, the experiments are carefully designed, and the data are presented with clarity and rigor, and will be of interest to a broad audience. It is suitable for publication in *Nature* provided the authors address some minor suggestions.

We thank the reviewers for their critique of our manuscript and kind words.

Comments:

Methods

- Line 113: Missing the word “family” after *Straboviridae*?

We have corrected this and all other instances.

- In all instances where centrifugation speeds are given in RPM, please give the speed in $\times g$ so that readers can more easily reproduce the methods without having to look up the conversion factor for the particular rotor used.

Thank you for this suggestion, all centrifugation speeds have been updated to $\times g$.

- The methods state that Sanger sequencing was performed by *plasmidsaurus*. To the best of my knowledge, *plasmidsaurus* performs oxford nanopore sequencing not Sanger sequencing.

Thank you for catching this mistake, the methods text has been corrected.

David Taylor and Tyler Dangerfield